# The C5a/C5a receptor 1 axis controls tissue neovascularization through CXCL4 release from platelets

Henry Nording[1,2], Lasse Baron[1], David Haberthür [3], Frederic Emschermann[4], Matthias Mezger[1], Manuela Sauter[1], Reinhard Sauter[1], Johannes Patzelt[5], Kai Knoepp[6], Anne Nording[7], Moritz Meusel [5], Roza Meyer-Saraei[2,5], Ruslan Hlushchuk[3], Daniel Sedding[6], Oliver Borst[4], Ingo Eitel[2,5], Christian M. Karsten[8], Robert Feil [9], Bernd Pichler[10], Jeanette Erdmann[2,11], Admar Verschoor[8], Emmanouil Chavakis[12], Triantafyllos Chavakis [13], Philipp von Hundelshausen[14], Jörg Köhl [8,15], Meinrad Gawaz [4] & Harald F. Langer [1,2,5✉]

Platelets contribute to the regulation of tissue neovascularization, although the specific factors underlying this function are unknown. Here, we identified the complement anaphylatoxin C5a-mediated activation of C5a receptor 1 (C5aR) on platelets as a negative regulatory mechanism of vessel formation. We showed that platelets expressing C5aR1 exert an inhibitory effect on endothelial cell functions such as migration and 2D and 3D tube formation. Growth factor- and hypoxia-driven vascularization was markedly increased in $C5ar1^{-/-}$ mice. Platelet-specific deletion of C5aR1 resulted in a proangiogenic phenotype with increased collateralization, capillarization and improved pericyte coverage. Mechanistically, we found that C5a induced preferential release of CXC chemokine ligand 4 (CXCL4, PF4) from platelets as an important antiangiogenic paracrine effector molecule. Interfering with the C5aR1-CXCL4 axis reversed the antiangiogenic effect of platelets both in vitro and in vivo. In conclusion, we identified a mechanism for the control of tissue neovascularization through C5a/C5aR1 axis activation in platelets and subsequent induction of the antiangiogenic factor CXCL4.

[1] Cardioimmunology Group, Medical Clinic II, University Heart Center Lübeck, Lübeck, Germany. [2] DZHK (German Centre for Cardiovascular Research), partner site Hamburg/Lübeck/Kiel, Lübeck, Germany. [3] Institute of Anatomy, University of Bern, Bern, Switzerland. [4] University Hospital, Department of Cardiovascular Medicine, Eberhard Karls University, Tübingen, Germany. [5] University Hospital, Medical Clinic II, University Heart Center Lübeck, Lübeck, Germany. [6] Department of Internal Medicine III, Cardiology, Angiology and Intensive Care Medicine, Martin-Luther-University Halle (Saale), Halle (Saale), Germany. [7] Institute of Medical Genetics and Applied Genomics, Eberhard Karls University, Tübingen, Germany. [8] Institute for Systemic Inflammation Research, University of Lübeck, Lübeck, Germany. [9] Interfaculty Institute of Biochemistry, University of Tübingen, Tübingen, Germany. [10] Institute for Preclinical Imaging, Eberhard Karls University, Tübingen, Germany. [11] Institute for Cardiogenetics, University of Lübeck, Lübeck, Germany. [12] Department for Internal Medicine III/Cardiology, University Hospital of the Johann-Wolfgang Goethe University, Frankfurt am Main, Germany. [13] Department of Clinical Pathobiochemistry, Institute of Clinical Chemistry and Laboratory Medicine, Medical Faculty, Technische Universität Dresden, Dresden, Germany. [14] Institute for Cardiovascular Prevention, Ludwig Maximilians University Munich, Munich, Germany. [15] Division of Immunobiology, Cincinnati Children's Hospital Medical Center, Cincinnati, OH, USA. ✉email: harald.langer@uksh.de

Tissue homeostasis and healing processes are of fundamental importance in living organisms. The complement system, a central player of the innate immune response, contributes to both tissue homeostasis and dysfunction[1]. Recent clinical and preclinical trials have shown that the complement system may be a promising therapeutic target for the restoration of tissue functionality in different pathological conditions, such as stroke[2], asthma[3], and myocardial infarction[4,5].

The complement system comprises a group of >30 plasma and cell-bound proteins that are activated canonically by one of the three pathways or noncanonically through plasma or cell-bound proteases[6]. The complement system exerts functions in innate immune defense as an important sentinel system[7]. It is tightly integrated and involved in crosstalk with other humoral and cellular arms of innate immunity, such as the contact system, pattern-recognition receptors, and immunoglobulin G (IgG) Fc receptors. Functions of the complement system include the opsonization of microbial intruders, the clearance of immune complexes, and the induction and regulation of the inflammatory response through the small cleavage fragments of C3 and C5, known as anaphylatoxins[8]. Activation of the terminal pathway of the cascade results in the assembly of C5b–C9, which can nucleate the membrane attack complex and mediate the osmotic lysis of target cells[1]. Previously, components of the complement cascade have been demonstrated to regulate angiogenesis in different pathophysiological settings[9–12], and the therapeutic value of targeting complement receptors is currently being explored in clinical trials[13]. Nevertheless, the exact mechanisms by which the complement system may interfere with proangiogenic or antiangiogenic cells or receptors remain to be determined.

Platelets play a decisive role in diseases that are featured by thrombus formation, and thus targeting platelet-associated functions is an established therapeutic principle for treating these diseases[14,15]. Beyond their classical function in blood hemostasis, platelets contribute to inflammation[16], immunomodulation[17,18], and atherosclerosis[19]. Increasing evidence points to a functional intersection between hemostasis and activation of the complement cascade[20]. Previous studies have reported that complement receptors may be expressed on platelets[21,22]. We recently showed that the complement C3a receptor expressed on platelets modulates platelet aggregation[23,24]. Interestingly, platelets store proangiogenic and antiangiogenic factors in distinct granules and can release them upon stimulation. For instance, platelet α-granules contain proangiogenic vascular endothelial growth factor (VEGF) and antiangiogenic CXC chemokine ligand 4 (CXCL4, PF4[25–27]). However, knowledge regarding the modulation of angiogenesis by platelets in vivo and the specific factors orchestrating this complex process is limited[18,27].

Here we addressed the crosstalk between the complement system and platelets in the context of ischemia-induced revascularization in vivo and find that C5a receptor 1 (C5aR1) expressed on platelets regulates the angiogenesis-related functions of platelets.

## Results

### Platelets express complement anaphylatoxin receptor C5aR1 in an environment of ischemia-induced revascularization. While it has been described that platelets contain proangiogenic and antiangiogenic factors[27], very little information exists regarding their regulatory function of hypoxia-driven revascularization and on how the release of these paracrine mediators is regulated. To address the role of platelets in the context of vessel formation and complement activation in vivo, we performed hindlimb ischemia experiments[28]. In this model, femoral artery ligation causes ischemia in the distal hindlimb tissue, which triggers revascularization.

Interestingly, we were able to detect complement activation in ischemic but not in nonischemic control hindlimb tissue, as demonstrated by the deposition of C3b, which colocalized with the formation of vascular structures (Fig. 1a). Further, we found significantly increased expression of complement anaphylatoxin receptor C5ar1 on the level of mRNA in ischemic hindlimb muscle tissue as compared with the nonischemic contralateral hindlimb (Fig. 1b). We also evaluated the presence of CD42b$^+$ platelets within the muscle tissue and found a significant increase in platelet abundance in the ischemic versus the nonischemic control hindlimb (Fig. 1c and Supplementary Fig. 1a, b). Accordingly, the platelet lineage-specific markers Cd42b mRNA and Cd41 mRNA were significantly elevated in the ischemic tissue (Fig. 1d). In vitro, we could demonstrate that adenosine diphosphate (ADP)-induced adherence of platelets to endothelial cells under static conditions was increased when the endothelium was hypoxic (Fig. 1e). Consistently, analysis of ischemic hindlimb muscles revealed colocalization of C5aR1 with the platelet marker CD42b at different stages of hindlimb revascularization (Fig. 1f). Thus, platelets spatiotemporally colocalize with complement activation and express C5aR1 in angiogenic tissue.

### C5a-mediated activation of C5aR1 on platelets inhibits endothelial functions that are important for angiogenesis. As both platelets and the complement system have been shown to regulate angiogenesis[12,29], we further questioned whether C5aR1 expressed on platelets modulates endothelial functions that are important for angiogenesis. We were able to detect C5aR1 in unstimulated washed platelets (Fig. 2a, b), which was markedly enhanced after collagen-related peptide (CRP) stimulation (Fig. 2c), slightly upregulated after ADP stimulation (Fig. 2d) and not altered after C5a stimulation (Fig. 2e).

Subsequently, we coincubated endothelial cells with platelets isolated from wild-type (WT) or C5ar1$^{-/-}$ mice. There was no difference in endothelial cell proliferation after the addition of platelets in the presence or absence of C5aR1 using the murine endothelial cell line MHEC-5T (Supplementary Fig. 2a, b). However, endothelial migration was increased by coincubation with C5ar1$^{-/-}$ platelets compared to coincubation with WT platelets using primary mouse lung endothelial cells (MLECs; Fig. 2f). The purity of primary cells was at least 90%, as verified by staining for CD102, CD144, and CD31 (Supplementary Fig. 3). Similarly, endothelial tube formation in an in vitro two-dimensional and three-dimensional (3D) tube-formation assay was increased after coincubation with C5ar1$^{-/-}$ platelets or C5ar1$^{-/-}$ platelet supernatant compared to coincubation with WT platelets or supernatant in MHEC-5T cells and in primary MLECs (Fig. 2g–k). Next, we preincubated human platelets with the C5aR1-specific antagonist PMX53 or control peptide before their addition to human umbilical vein endothelial cells (HUVECs) in the presence of C5a. The results show that endothelial tube formation was significantly enhanced when C5aR1 activity was inhibited by PMX53 (Fig. 2l). In order to better characterize platelet–endothelial interactions in vitro, we assessed whether WT or C5ar1$^{-/-}$ platelets differ with respect to static adhesion to endothelial cells. ADP induced a significant increase in adhesion but there was no difference between WT and C5aR1-deficient platelets (Fig. 2m, n).

### Increased ischemia-induced revascularization in C5aR1-deficient mice. Next, we explored the functional role of C5aR1 in vessel formation in vivo by assessing murine embryonic angiogenesis and hindlimb ischemia-induced revascularization. Whole-mount staining was performed on E11.5 WT or C5ar1$^{-/-}$

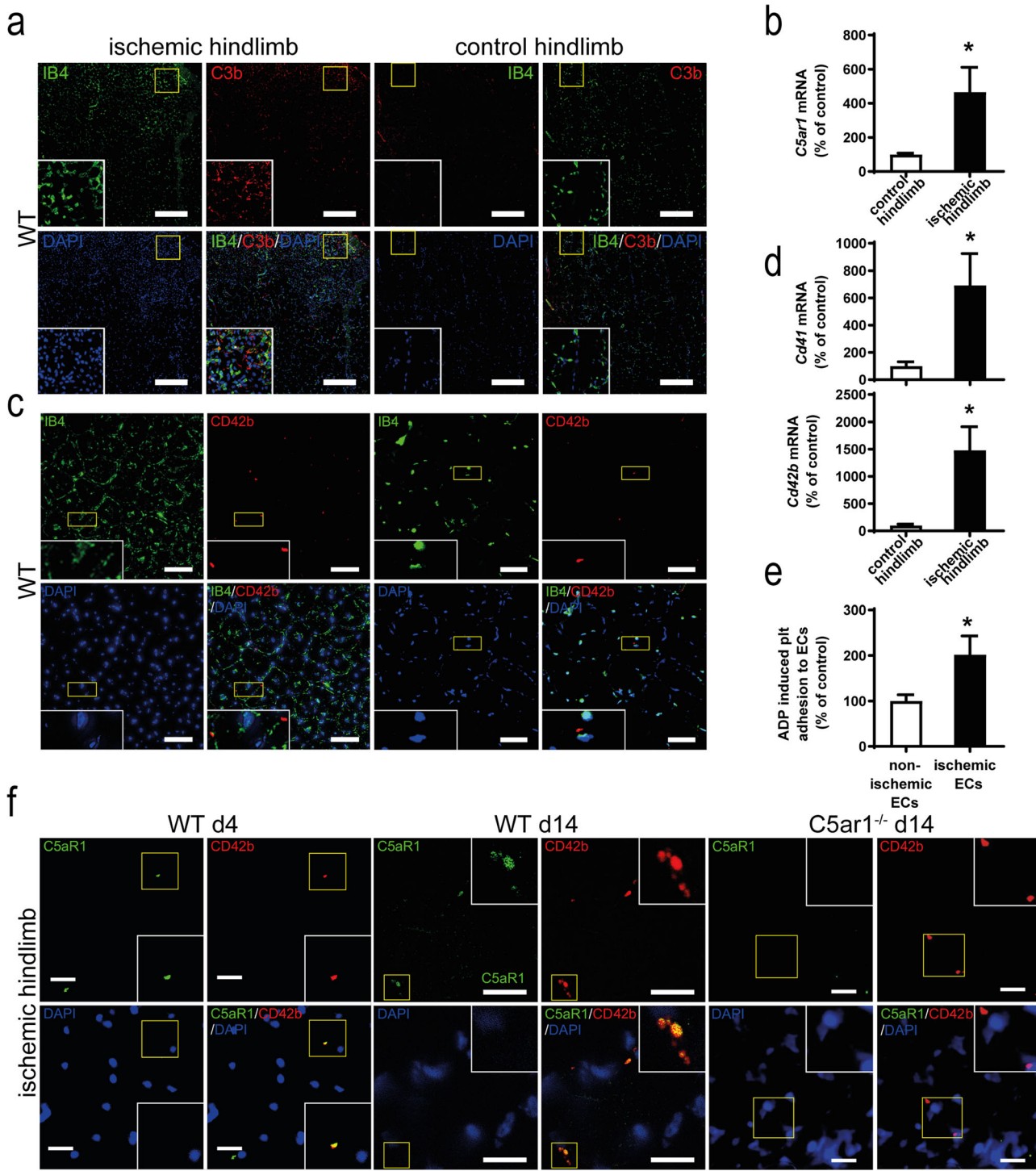

embryonic hindbrains. No impact of C5aR1 on developmental angiogenesis could be detected (Supplementary Fig. 4). In contrast, we observed an effect of C5aR1 deficiency on ischemia-driven vessel growth. When we monitored revascularization after femoral artery ligation by repeated laser Doppler imaging (LDI) measurements over 14 days, we found significantly enhanced perfusion in $C5ar1^{-/-}$ animals, indicating faster and more pronounced revascularization when compared to WT control animals (Fig. 3a, b). Next, gastrocnemius muscles were explanted from ischemic hindlimbs after 14 days, and vessel density was quantified by immunofluorescence staining for isolectin B4 (IB4; green) and 4,6-diamidino-2-phenylindole (DAPI; blue; Fig. 3c).

Vessel density of hindlimb muscles from $C5ar1^{-/-}$ mice was significantly higher as compared with that of WT controls after the induction of hindlimb ischemia (Fig. 3c, d). Subsequently, we administered platelet-depleting serum to both WT and $C5ar1^{-/-}$ mice after induction of hindlimb ischemia over 14 days. Platelet depletion was effective and showed a platelet-depletion efficiency >90% measured in peripheral blood of mice (Supplementary Fig. 5) Upon platelet depletion, we observed no significant difference in revascularization between both strains any more, suggesting a critical role of C5aR1 activation on platelets for ischemia-induced angiogenesis (Fig. 3e). Accordingly, vessel density was comparable in both the groups (Fig. 3f).

**Fig. 1 Complement anaphylatoxin receptor C5aR1 associated with platelets is abundant in ischemic tissue.** WT mice were subjected to hindlimb ischemia as described in the "Methods" section. **a** One week after the induction of hindlimb ischemia, ischemic muscles revealed abundant complement activation (left side), as demonstrated by the presence of the C3 cleavage product C3b (red). IB4 staining (green) depicts vascular structures; DAPI (blue) depicts nuclei. The nonischemic contralateral control muscle only displayed minimal complement C3b deposition (right side). ×200 magnification, scale bars represent 200 μm. Image is representative of at least four analyzed muscles. **b** Furthermore, ischemia resulted in significantly increased mRNA levels of the anaphylatoxin receptor *C5ar1* compared with nonischemic contralateral hindlimb muscles as internal controls. Data are the mean ± SEM ($n = 4$ independent experiments) and are shown as the percentage of control. mRNA levels in the contralateral control muscles of WT animals represent 100%. *$p < 0.05$. **c** In the ischemic hindlimb tissue, more platelets could be detected compared to the nonischemic control limb using CD42b (red). Displayed is a representative image, IB4 staining (green) depicts vascular structures; DAPI (blue) depicts nuclei. Scale bars represent 10 μm. Image is representative of 11–12 whole-muscle sections per group. **d** RT-PCR analysis detected significantly increased mRNA levels of the platelet marker *Cd42b* as well as the platelet marker *Cd41* compared with the contralateral unaffected muscles. Data are presented as the mean ± SEM ($n = 4$ independent experiments) and are shown as the percentage of control. mRNA levels in the contralateral control muscles of WT animals represent 100%. *$p < 0.05$. **e** In a static adhesion assay of platelets on endothelial cells (MHEC), the ADP-induced increase in adhesion was significantly greater under ischemic conditions with an ischemically preconditioned endothelial monolayer compared to normoxic endothelial cells. Data are presented as the mean ± SEM ($n = 4$ independent experiments) and are shown as the percentage of control. ADP-induced platelet adhesion to normoxic endothelial cells expressed as the area fraction of platelet-specific staining represents 100%. *$p < 0.05$. **f** Immunofluorescent costaining of murine hindlimb muscle sections 4 as well as 14 days after the induction of ischemia at ×630 magnification (left side) showing colocalization of the platelet markers CD42b (red) and C5aR1 (green). Representative images confirm that the cells are anuclear (DAPI negative, blue). Control staining of muscles from *C5ar1*[−/−] mice (right side) verifies specificity of C5aR1 staining. Scale bars represent 5 μm. Image is representative of at least four analyzed muscles. Two-sided Student's *t* test in **b**, **d**, **e**.

**The increased revascularization in C5aR1-deficient mice is mediated by platelet C5aR1 stimulation-driven collateralization as well as capillary and pericyte growth.** To further define the role of platelet C5aR1 in vivo, we crossed floxed GFP-*C5ar1*[fl/fl][30] mice with *Pf4-cre*[+] mice[31] and generated *Pf4-cre*[+] *C5ar1*[fl/fl] mice, lacking C5aR1 in platelets (Supplementary Fig. 6). When we induced hindlimb ischemia and performed repeated LDI measurements over 14 days in *Pf4-cre*[+] *C5ar1*[fl/fl] mice, we found stronger revascularization in platelet-specific *C5ar1*-knockout mice than in *Pf4-cre*[−] *C5ar1*[fl/fl] littermate controls (Fig. 4a, b). Similar to *C5ar1*[−/−] mice, we detected an increased vessel density in the ischemic muscle tissue of *Pf4-cre*[+] *C5ar1*[fl/fl] mice (Fig. 4c, d). To rule out that a difference in platelet deposition upon C5aR1 deficiency in ischemic tissues is responsible for the observed effects, we assessed platelet abundance and microthrombi size in whole-muscle sections of ischemic *Pf4-cre*[+] *C5ar1*[fl/fl] mice and *Pf4-cre*[−] *C5ar1*[fl/fl] controls (Fig. 4e). We found no significant differences between both the groups (Fig. 4f). As CXCL4 (PF4) is also known to be expressed by monocytic cells, which are important mediators of ischemia-induced revascularization[32], we next assessed whether *Pf4-cre*[+] *C5ar1*[fl/fl] mice showed altered C5aR1 expression on monocytes or macrophages compared to *Pf4-cre*[−] *C5ar1*[fl/fl] mice but found no differences (Supplementary Fig. 7).

To obtain insights into the mechanisms underlying the increased revascularization observed in *Pf4-cre*[+] *C5ar1*[fl/fl] mice in vivo, we performed micro-computed tomographic (microCT) analysis of platelet-specific *C5ar1*-knockout mice subjected to the hindlimb ischemia model at day 9. After 3D reconstruction, the occluded vessel could be identified by comparison with the contralateral control hindlimb (Fig. 5a, b, arrows). Furthermore, collateral artery formation could be observed (Fig. 5a, b and Supplementary Movies 1a–d). By virtual reconstruction, the vascular tree was extracted from the images and analyzed (Supplementary Fig. 8a, b). Vessel size distribution in both the groups (*Pf4-cre*[+] *C5ar1*[fl/fl] and *Pf4-cre*[−] *C5ar1*[fl/fl]) was assessed. The analysis yielded an increase in small- and medium-sized vessels, which comprise arterial collateral vessels, in the proximal section of the hindlimb (Fig. 5c). We also considered the possibility that platelet C5aR1 impacts on the formation of pre-existent collateral vessels, which are known to be important for the initial phase of ischemia-induced revascularization[33]. Therefore, the vascular tree in nonischemic hindlimbs of both genotypes was quantified. No difference in mean vessel size could be observed (Supplementary Fig. 8c).

In the distal hindlimb, we stained for smooth muscle actin (SMA) and NG2 expression, which denotes arterial vessels. We then quantified the size of the largest arteries and arteriolar density within the gastrocnemius muscle (Fig. 5d, e). However, no significant differences could be detected between *Pf4-cre*[+] *C5ar1*[fl/fl] mice and *Pf4-cre*[−] *C5ar1*[fl/fl] mice in this distal muscle. NG2 staining allowed us to determine pericyte abundance and pericyte coverage of vessels. Comparing ischemic hindlimb muscles from platelet-specific C5aR1-deficient mice with *Pf4-cre*[−] *C5ar1*[fl/fl] control mice at day 14 after induction of ischemia, we observed increased pericyte density and enhanced pericyte coverage in platelet-specific C5aR1-deficient mice (Fig. 5f, g). This effect may explain the amplified vessel density in *Pf4-cre*[+] *C5ar1*[fl/fl] mice (Fig. 4c, d).

Our observations that platelet C5aR1 inhibits endothelial cell functions, which are relevant for angiogenesis in vitro, and our findings that platelet-specific *C5ar1*[−/−] animals exhibit increased ischemia-induced revascularization prompted us to investigate the regulatory mechanisms underlying this process.

**Platelet C5aR1 mediates the release of antiangiogenic, platelet-derived CXCL4.** To elucidate the mechanisms underlying platelet C5aR1-mediated inhibition of vessel growth and endothelial functions, we stimulated isolated platelets with C5a or vehicle control. A membrane-based antibody array was performed using the harvested supernatants to obtain insights into the regulation of secreted antiangiogenic factors (Fig. 6a). Interestingly, the platelet α-granule component CXCL4 was strongly induced after C5a stimulation (Fig. 6b). Bonferroni's post hoc analysis showed significant differences compared to all factors except Angiopoietin-1, for which C5a induced a downregulation, as well as endostatin (Supplementary Fig. 9).

To further approach the mechanism of C5a-induced CXCL4 secretion, we stained stimulated platelets for the α-granule components P-selectin and CXCL4 (Fig. 6c). Low-dose ADP and C5a stimulation induced platelet activation, as indicated by P-selectin upregulation, and P-selectin mobilization toward the cell periphery (ring-like staining pattern, arrows; Fig. 6c, d). C5a induced an altered secretion response compared to ADP, as indicated by a decrease in CXCL4 intensity relative to P-selectin intensity after C5a stimulation compared with ADP (Fig. 6e).

Interestingly, using confocal microscopy (Supplementary Fig. 10) we found α-granules, which contain predominantly CXCL4 over P-selectin as well as granules containing both CXCL4 and P-selectin (Fig. 6f, g). When we stimulated platelets

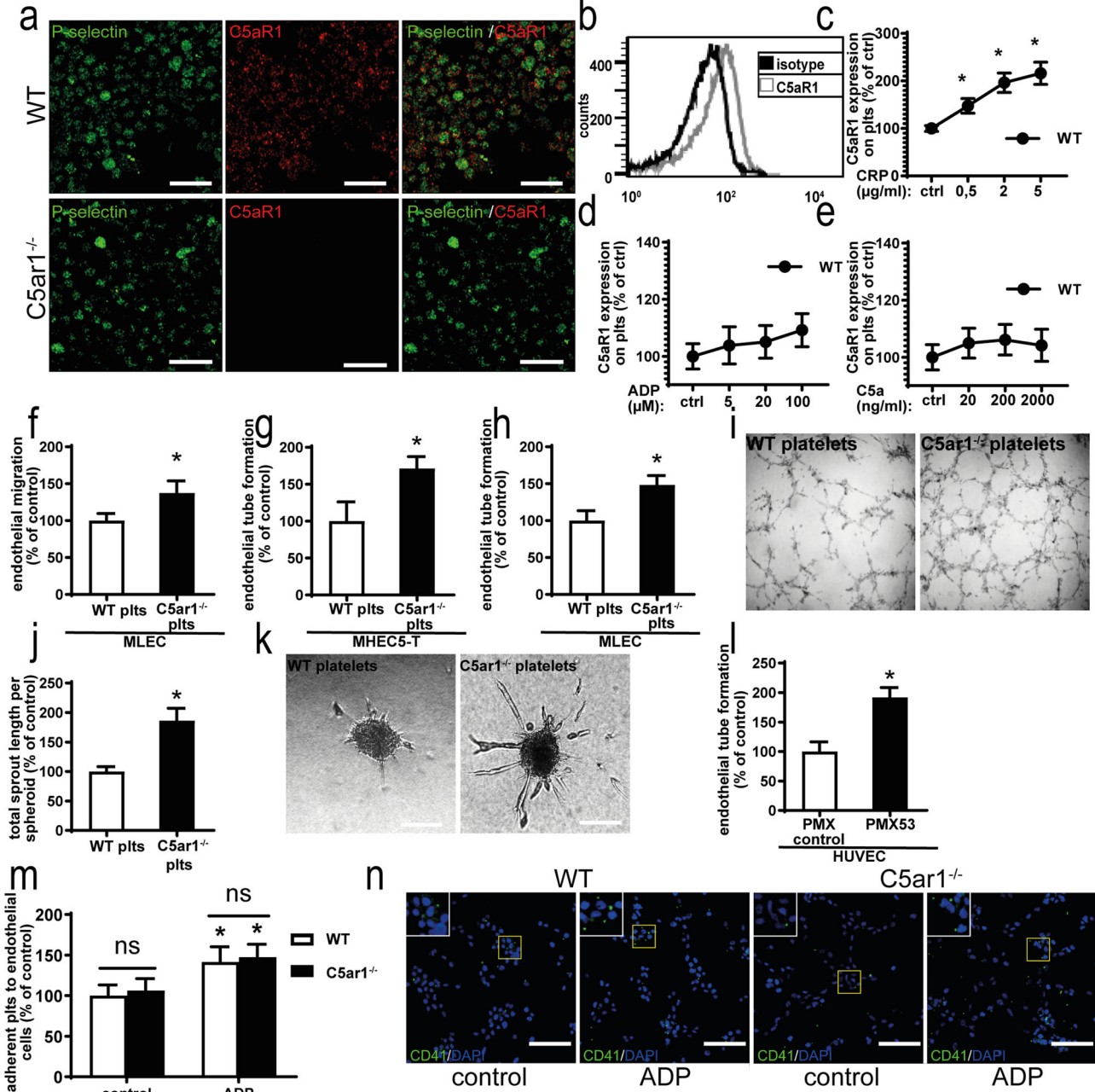

with C5a, we observed a release of these CXCL4-predominant granules from platelets measured as the reduction in area fraction (Fig. 6h) but not of granules containing both CXCL4 and P-selectin or granules containing P-selectin but not CXCL4. These findings strongly suggest that C5a induces the secretion of an α-granule subset predominantly holding CXCL4.

Using human cells, conventional enzyme-linked immunosorbent assay (ELISA) measurements of the CXCL4 concentration in the supernatant of C5a-stimulated platelets confirmed a dose-dependent and significant increase compared with vehicle control-treated platelets (Fig. 6i). Next, time and dose dependency in murine platelets were assessed to find the optimal stimulation conditions in murine platelets (Supplementary Fig. 11). Maximum CXCL4 release was observed at a C5a concentration of 200 ng/ml for mouse samples and 2000 ng/ml for human cells (Fig. 6i and Supplementary Fig. 12a). A dose-dependent CXCL4 secretion effect was achieved at stimulation conditions of 10 min at 25 °C and 10 min at 37 °C

(Supplementary Fig. 12). Analyzing megakaryocytes from $C5ar1^{-/-}$ mice, we observed no difference in CXCL4 granule area fraction (Fig. 6j, k) compared to WT cells. Furthermore, there was no difference in $Cxcl4$ mRNA between WT and $C5ar1^{-/-}$ megakaryocytes as analyzed by real-time PCR (Fig. 6l), indicating regulation of CXCL4 on the platelet rather than on the megakaryocyte level.

In order to compare C5a-induced CXCL4 release with CXCL4 secretion induced by classical platelet agonists, we performed platelet stimulation with different platelet activators and compared the secretion response of WT and $C5ar1^{-/-}$ mice (Supplementary Fig. 13). We found that, while WT and $C5ar1^{-/-}$ mice displayed no significant differences in response to classical agonist-induced CXCL4 release, no significant CXCL4 secretion occurred after C5a stimulation of $C5ar1^{-/-}$ platelets (Supplementary Fig. 13b). In WT mice, CXCL4 level after 200 ng/ml C5a stimulation was comparable to 25 μM ADP stimulation (Supplementary Fig. 13a). WT and $C5ar1^{-/-}$ platelets did not differ

**Fig. 2 Platelets express C5aR1 and platelet C5aR1 inhibits various endothelial functions. a** Isolated washed murine platelets express C5aR1 as assessed by immune fluorescence microscopy. There was only partial colocalization of C5aR1 (red) with the α-granule marker P-selectin (green). ×630 magnification, scale bars represent 5 µm. Images are representative of four independent experiments. **b** Histogram showing C5aR1 expression on platelets (gray curve); the black curves show the histogram obtained with an IgG isotype. The histogram is representative of the analysis of four independent platelet samples. **c** Furthermore, flow cytometry revealed that C5aR1 expression on platelets is dynamic. Upon stimulation with CRP, platelet C5aR1 expression increased in WT platelets. In **c**–**e**, data are displayed as the mean ± SEM ($n = 4$ independent experiments) and are shown as the percentage of control. The expression in the unstimulated WT group represents 100%. *$p < 0.05$. **d** Upon stimulation with ADP, WT platelets displayed C5aR1 upregulation at higher concentrations. **e** Upon treatment with C5a at the indicated concentrations, no significant changes in C5aR1 expression on platelets could be detected. **f** Primary lung endothelial cells (MLECs) were grown to a confluent monolayer, which was wounded by scratching with a plastic pipette and coincubated with $C5ar1^{-/-}$ or WT platelets. The absence of C5aR1 resulted in increased endothelial migration after 17 h. Data are displayed as the mean ± SEM ($n = 6$ independent experiments). The area not populated with cells at the start of the experiment minus the area not populated with cells after 17 h of coincubation with WT platelets represents 100%. *$p < 0.05$. **g** Endothelial tube formation by MHEC-5T cells in vitro was significantly increased after incubation with platelets isolated from $C5ar1^{-/-}$ mice compared platelets isolated from WT mice. Data are shown as the mean ± SEM ($n = 4$ independent experiments) and displayed as the total tube length after 6.5 h. The cumulative length of endothelial tubes after incubation with WT control platelets represents 100%. *$p < 0.05$. **h** Similarly, endothelial tube formation by MLECs was increased after exposure to $C5ar1^{-/-}$ platelets compared with WT platelets. Data are shown as the mean ± SEM ($n = 4$ independent experiments). Tube formation after coincubation with WT control platelets represents 100%. *$p < 0.05$. **i** Representative images of endothelial tube formation of MHEC-5T after coincubation with WT or $C5ar1^{-/-}$ platelets. **j** The supernatant of murine WT platelets significantly inhibited tube formation from endothelial spheroids compared with the supernatant of $C5ar1^{-/-}$ platelets. Data are presented as the mean ± SEM ($n = 3$ independent experiments) and are shown as the percentage of the total tube length after 24 h in controls. Tube formation after incubation with WT control platelet supernatant represents 100%. *$p < 0.05$. **k** Representative images of sprout formation from endothelial spheroids in both groups. Scale bar represents 100 µm. **l** Human platelets were preincubated with a C5aR1 antagonist (PMX53) or control peptide. Subsequently, platelets were coincubated with human umbilical vein endothelial cells (HUVECs) on Matrigel. Endothelial tube formation was significantly enhanced by preincubation with PMX53 compared with the control. Data are shown as the mean ± SEM ($n = 4$ independent experiments with separate donors) of the total tube length after 6.5 h. The cumulative length of endothelial tubes after incubation with platelets preincubated with control peptide represents 100%. *$p < 0.05$. **m** There was no significant difference in adhesion to endothelial cells under static conditions between WT and C5aR1-deficient platelets under normoxic conditions. Data are shown as the mean ± SEM ($n = 4$–5 independent experiments) and as the percentage of control. Adherent WT platelets expressed as the area fraction of platelet-specific staining divided by the number of endothelial cells expressed as the DAPI count per area represents 100%. *$p < 0.05$ compared to control-stimulated platelets. **n** Representative images of static platelet adhesion of WT and $C5ar1^{-/-}$ platelets to endothelial cells. Scale bars represent 200 µm. Two-sided Student's $t$ test in **b**, **f**–**h**, **j**, **l**. ANOVA in **c**–**e**, **m**.

significantly in their aggregation behavior (Supplementary Fig. 14a, b). To further rule out that defects in hemostasis account for the neovascularization phenotype of $C5ar1^{-/-}$ mice, we carried out additional in vivo experiments using ferric chloride-induced vascular injury. We observed no significant difference in time to thrombus formation in the absence of C5aR1 in comparison to WT mice (Supplementary Fig. 14c). Furthermore, C5aR1 deficiency had no significant effect on in vivo bleeding time (Supplementary Fig. 14d). Nevertheless, as the results in Fig. 6c, d already showed, C5a induced platelet activation. We quantified C5aR1-dependent platelet activation using flow cytometry and found that there was significantly increased but modest platelet activation in WT platelets following C5a stimulation but not in $C5ar1^{-/-}$ platelets (Fig. 6m). Furthermore, C5a induced a dose-dependent calcium signal (Supplementary Fig. 14e). However, C5a had no effect on other platelet activation markers like GPVI, CD40L, CD61, and platelet–leukocyte aggregate formation (Supplementary Fig. 14f). Finally, C5a had no impact on classical platelet functions such as adhesion to von Willebrand factor or aggregation (Supplementary Fig. 15). C5aR1 deficiency did not influence C3aR protein content of platelets (Supplementary Fig. 16). C3aR has recently been shown to modulate platelet aggregation[24]. Similarly, C3aR receptor expression profile of platelets was not significantly altered in the absence of C5aR1 (Supplementary Fig. 17).

Next, we assessed the impact of C5a on the secretion of other angiogenic factors, which are known to be stored in platelets, by conventional ELISA. Compared to other agonists and vehicle control, C5a did not induce secretion of thrombospondin 1 (Supplementary Fig. 18a), endostatin (Supplementary Fig. 18b), or platelet-derived growth factor (PDGF; Supplementary Fig. 18c) from human platelets. Also, C5a stimulation had no effect on the secretion of δ-granule cargo ATP from platelets (Supplementary

Fig. 19). The granule content of platelets from WT versus $C5ar1^{-/-}$ or $Pf4\text{-}cre^-$ $C5ar1^{fl/fl}$ versus $Pf4\text{-}cre^+$ $C5ar1^{fl/fl}$ mice showed no significant differences, respectively (Supplementary Fig. 20). Then we analyzed the effect of C5 deficiency on C5aR1-dependent CXCL4 secretion from platelets. Platelets isolated from C5-deficient mice expressed C5aR1 (Supplementary Fig. 21a) but displayed C5a-dependent CXCL4 secretion after stimulation with higher C5a concentrations (500 ng/ml; Supplementary Fig. 21b), while no major differences were detected in the activation response of $C5^{-/-}$ platelets compared to WT platelets (Supplementary Fig. 21c, d). C5aR2 is known to modulate C5aR1 function in a number of different cell types[34]. However, we could not detect relevant C5aR2 expression on platelets or an effect of C5aR2-specific stimulation on CXCL4 release from platelets (Supplementary Fig. 22).

Furthermore, we analyzed several potential pathways, which could link binding of C5a via C5aR1 to CXCL4 secretion. We tested phosphorylation of Akt, phosphoinositide-3 kinase (PI3K), glycogen synthase kinase 3β (GSK-3β), p44/42 mitogen-activated protein kinase (MAPK), PLCβ3, PLCγ2, protein kinase A (PKA), and RAP1. Interestingly, we found that the signal is conducted via the Gβγ subunit of C5aR1 and not by Gαi, as we found C5a-dependent phosphorylation of PI3K and Akt but not PKA (Fig. 6n–q and statistical analysis in Supplementary Fig. 23). PKC activation seems to be central for C5a-induced CXCL4 secretion as we found consistent C5a-dependent PKC phosphorylation and could also show that PKC activation induces CXCL4 secretion (Fig. 6n, o).

**Platelet C5aR1-driven CXCL4 confers the inhibitory effect on vessel formation.** To confirm the relevance of C5a-induced CXCL4 secretion, we treated endothelial cells with the supernatant of C5a-stimulated platelets. The addition of supernatant from C5a-treated WT platelets to endothelial cells resulted in an

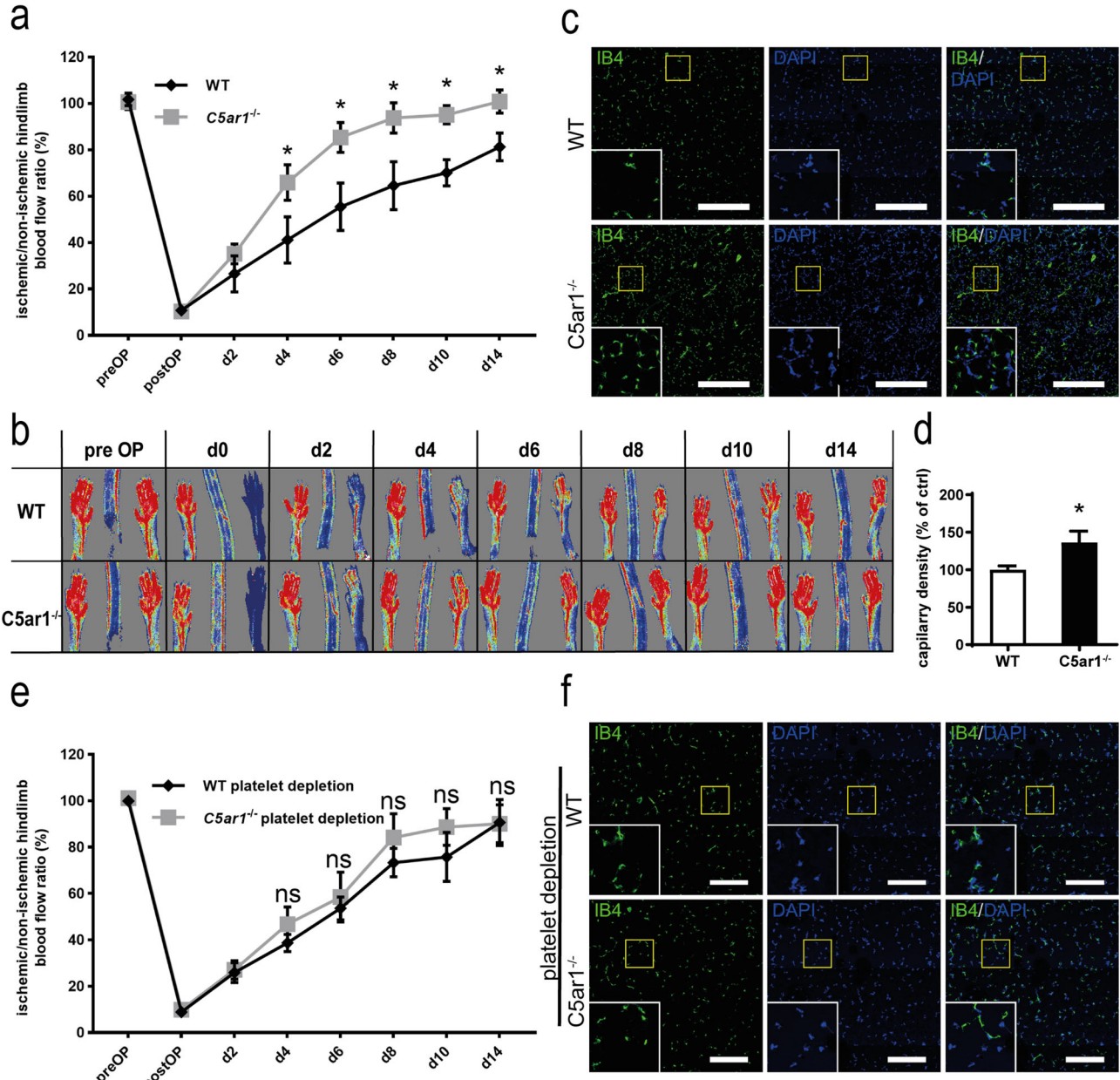

**Fig. 3 C5aR1 deficiency promotes ischemia-induced revascularization in vivo.** WT or *C5ar1*$^{-/-}$ mice were subjected to hindlimb ischemia and analyzed after 2 weeks. **a** Revascularization of the hindlimbs after femoral artery ligation was visualized by laser Doppler fluximetry (LDI). We found increased revascularization in *C5ar1*$^{-/-}$ animals. Data are shown as the mean ± SEM ($n = 7$ animals per group) and are displayed as the percentage of the perfusion in the contralateral control limb. *$p < 0.05$. **b** Representative LDI images of mouse hindlimbs after femoral artery ligation illustrate increased revascularization in *C5ar1*$^{-/-}$ animals compared with WT controls. **c** Vessel density in the gastrocnemius muscle of the ischemic limbs was quantified by immunofluorescence staining. Vessels were visualized by isolectin B4 (IB4 in green, nuclei in blue), and images of whole-muscle sections were acquired as tile scans and analyzed. At 14 days after the induction of ischemia, *C5ar1*$^{-/-}$ mice exhibited a significantly higher vessel density than WT controls. ×200 magnification, scale bars represent 200 μm. **d** Quantification of the IB4-positive area fraction in the muscle sections reveals higher vessel abundance in ischemic *C5ar1*$^{-/-}$ hindlimbs. Data are shown as the mean ± SEM ($n = 10$ whole-muscle sections per group) and are displayed as the percentage of control. The IB4-positive area fraction in WT control hindlimbs represents 100%. *p < 0.001. **e** Furthermore, WT or *C5ar1*$^{-/-}$ mice were subjected to hindlimb ischemia and platelets were depleted systemically by injection of platelet depleting serum starting on the first day post induction of ischemia. We could not detect significant differences ($p < 0.05$) in revascularization. Data are shown as the mean ± SEM ($n = 7$ animals per group) and are displayed as the percentage of the perfusion in the contralateral control limb. **f** Vessel density in the gastrocnemius muscle of the ischemic limbs was quantified by immunofluorescence staining as described in **c**. At 14 days after the induction of ischemia, WT and *C5ar1*$^{-/-}$ mice with platelet depletion did not exhibit significant differences in vessel density. ×200 magnification, scale bars represent 200 μm. Image is representative of 10 whole-muscle sections per group analyzed. Two-way ANOVA with Bonferroni's post hoc test in **a**, **e**. Two-sided Student's *t* test in **d**.

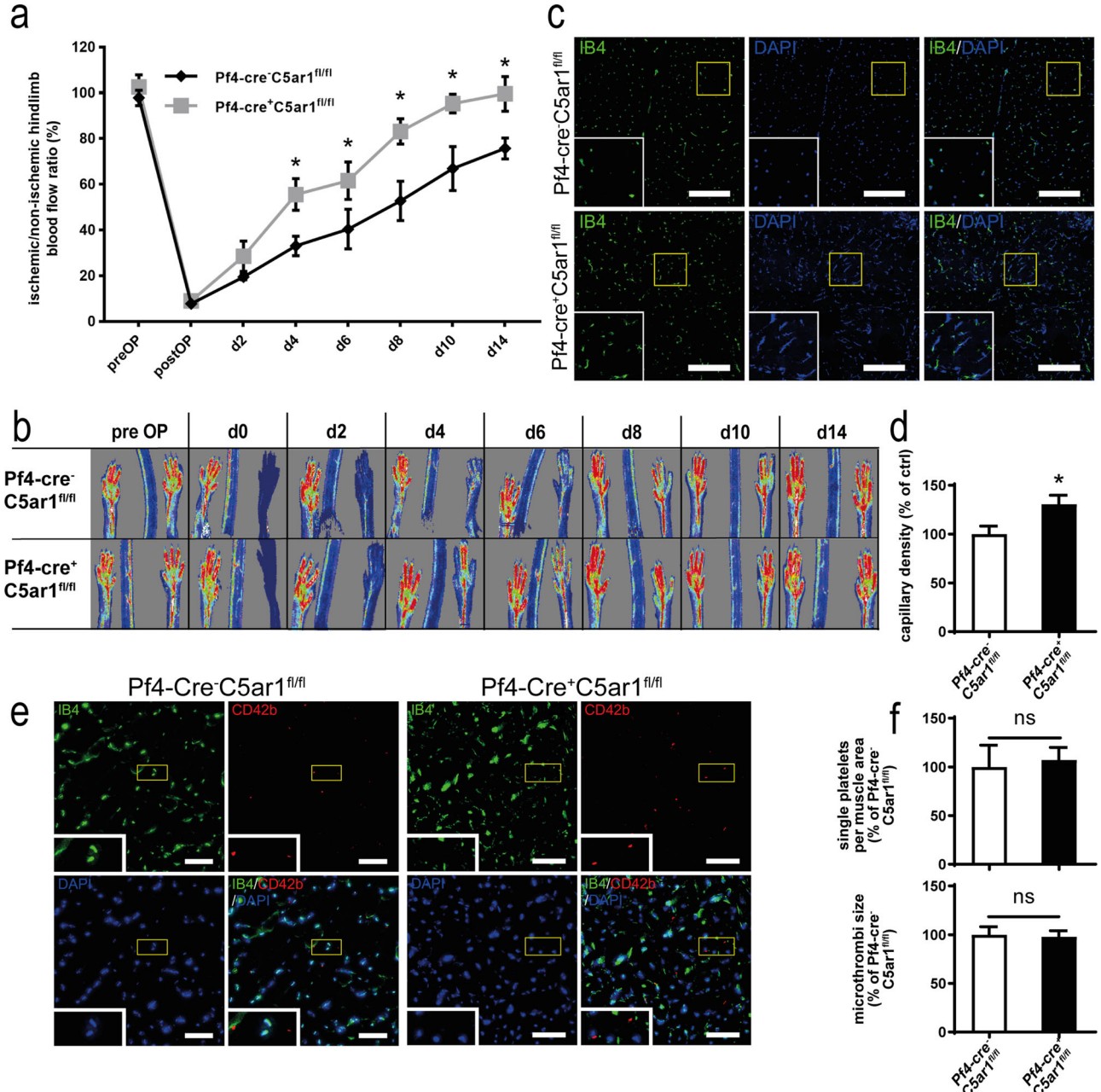

**Fig. 4 Platelet-specific C5a receptor deletion inhibits revascularization but does not alter platelet deposition in vivo.** *Pf4-cre+ C5ar1fl/fl* mice were generated as described in the "Methods" section. Hindlimb ischemia was induced, and *Pf4-cre− C5ar1fl/fl* mice served as controls. **a** Platelet-specific C5a receptor 1 knockout mice showed increased revascularization. Data are presented as the mean ± SEM (*n* = 5 animals per group) and are displayed as the percentage of the perfusion in the contralateral control limb. *\*p* < 0.05. **b** Representative LDI images of mouse hindlimbs after femoral artery ligation illustrate increased revascularization in *Pf4-cre+ C5ar1fl/fl* mice compared with littermate control animals. **c** Vessel density in the representative gastrocnemius muscle sections (vessel marker IB4 in green, nuclei in blue) of the ischemic limbs of *Pf4-cre+ C5ar1fl/fl* showed significantly higher vessel density than *Pf4-cre− C5ar1fl/fl* controls. Scale bars represent 200 μm. **d** Quantification of the IB4-positive area fraction in the muscle sections reveals higher vessel abundance in ischemic *Pf4-cre+ C5ar1fl/fl* hindlimbs at ×200 magnification. Data are shown as the mean ± SEM (*n* = 10 whole-muscle sections per group) and are displayed as the percentage of control. The IB4-positive area fraction in *Pf4-cre− C5ar1fl/fl* hindlimbs represents 100%. *\*p* < 0.05. **e** By staining for CD42b (red), platelet deposition was quantified in whole-muscle sections of *Pf4-cre+ C5ar1fl/fl* or of *Pf4-cre− C5ar1fl/fl* mice, as described in the "Methods" section. No difference in the number of platelets per area or microthrombi size was detected. Displayed are representative images for both genotypes, IB4 staining (green) depicts vascular structures; DAPI (blue) depicts nuclei. Scale bars represent 100 μm. **f** CD42b-positive single platelets were quantified by size using automated digital image analysis as described in the "Methods" part in whole-muscle sections. There was no significant difference between *Pf4-cre+ C5ar1fl/fl* and *Pf4-cre− C5ar1fl/fl* mice both with respect of the number of platelets per muscle area as well as the size of the platelets/microthrombi. Data are shown as the mean ± SEM (*n* = 6 whole-muscle sections per group) and are displayed as the percentage of control. The readings in *Pf4-cre− C5ar1fl/fl* hindlimbs represent 100% in both graphs. *\*p* < 0.05. Two-way ANOVA with Bonferroni's post hoc test in **a**. Two-sided Student's *t* test in **d**, **f**.

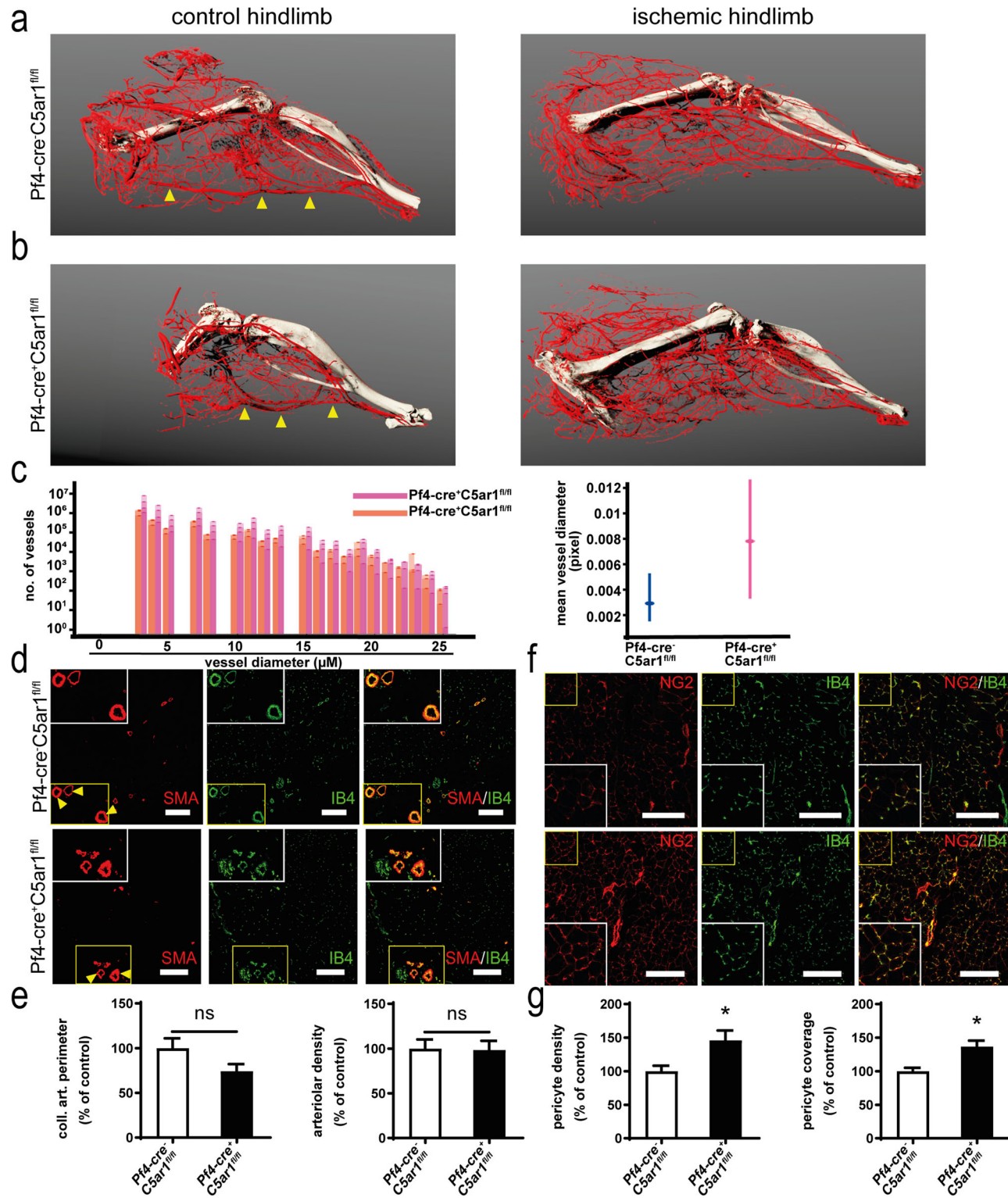

inhibitory effect on endothelial tube formation (Fig. 7a). This effect was C5aR1 specific, as the platelet supernatant from C5a-stimulated *C5ar1*−/− platelets had no effect (negative control; Fig. 7a). Addition of CXCL4 to C5a-conditioned *C5ar1*−/− supernatant restored the effect of C5a-conditioned WT supernatant and the supernatant of *C5ar1Cxcl4*-double-knockout mice yielded comparable results with *C5ar1*−/− supernatant (Fig. 7a). Moreover, endothelial migration was inhibited by C5a-stimulated WT platelet supernatant (Supplementary Fig. 24a), whereas proliferation was not affected (Supplementary Fig. 24b). Next, we

depleted CXCL4 from C5a-stimulated WT platelet supernatant using anti-CXCL4 antibody-coated Sepharose beads. Removal of CXCL4 from C5a-conditioned platelet supernatant resulted in significantly increased endothelial tube formation, suggesting that C5a-driven CXCL4 release is one of the central factors mediating the antiangiogenic role of platelets (Fig. 7b). In *C5ar1*−/− platelet supernatant, CXCL4 depletion had no effect (Fig. 7b). As C5a was shown to induce phosphorylation of PI3K, Akt, and PKC (Fig. 6n–q), we tested whether preincubation of platelets with the according kinase inhibitors followed by C5a stimulation could

**Fig. 5 C5aR1 deficiency on platelets induces increased collateral artery formation, capillarization, and pericyte coverage.** Platelet-specific C5aR1-deficient mice were subjected to hindlimb ischemia. To visualize collateral arteries, mice were perfused with a contrast agent at d9 after induction of ischemia and subjected to microCT analysis as described in the "Methods" section. **a** 3D reconstructions of arteries from *Pf4-cre⁻ C5ar1^fl/fl* mice showed collateral artery formation, while the main femoral artery (arrows) was no longer perfused. **a, b** A comparison of collateral artery formation in mice with C5aR1-deficient or -competent platelets shows increased collateralization in platelet-specific *C5ar1*-knockout mice. Images are representative of 3–4 mice, i.e., 3–4 vessel trees analyzed. **c** The size distribution of vessels within the ischemic hindlimbs of *Pf4-cre⁺ C5ar1^fl/fl* mice and *Pf4-cre⁻ C5ar1^fl/fl* mice was quantified within the microCT data in mouse hindlimbs proximal to the knee in the region of the adductor muscle (for details on methodology, please refer to the "Methods" section and Supplementary Fig. 8). *Pf4-cre⁺ C5ar1^fl/fl* mice displayed larger vessels in both the small vessel range as well as the medium-sized vessel range. Data are shown as overlaid single measurements of each of the 3–4 vessel trees analyzed and as mean ± 95% confidence interval of all vessels quantified within each size region (n = 3–4 animals, i.e., vessel trees per group). Data are stated in pixels/voxels and 1 voxel represents 21.6 µm. Furthermore, the mean vessel diameter shows a clear trend that vessels in the proximal part of the ischemic hindlimb are larger in *Pf4-cre⁺ C5ar1^fl/fl* mice compared to controls (right) going along with improved revascularization. **d** Characterization of arteries in the distal ischemic hindlimb gastrocnemius muscle of *Pf4-cre⁺ C5ar1^fl/fl* or *Pf4-cre⁻ C5ar1^fl/fl* mice reveal no significant differences in large artery size or abundance of arterioles (SMA in red, vessel marker IB4 in green). ×200 magnification, scale bars represent 200 µm, arrows mark large muscle arteries double positive for SMA and IB4. **e** Large distal muscle arteries were assessed by measuring the perimeter of the 5 largest arteries present in a whole-muscle section acquired by tile scanning at ×200 magnification (left). Furthermore, the area fraction in the muscle sections of SMA-positive vessels was quantified at ×200 magnification. Data are shown as the mean ± SEM (n = 10 whole-muscle sections per group) and are displayed as the percentage of control. The sum of artery perimeters of the five largest arteries per section or the area fraction of SMA staining in *Pf4-cre⁻ C5ar1^fl/fl* hindlimb muscle sections represent the 100% control. n.s. = no significant difference was observed. **f** Platelet-specific *C5ar1*-knockout mice displayed increased pericyte coverage of vessels in the ischemic muscle tissue at d14 after induction of ischemia. (pericyte marker NG2 in red, vessel marker IB4 in green). ×200 magnification, scale bars represent 200 µm. **g** Pericyte density was assessed by measuring the area fraction of NG2-positive staining in whole-muscle sections acquired by tile scanning at ×200 magnification (left). Furthermore, pericyte coverage was assessed by calculating the spatial colocalization of NG2 and IB4 staining. Data are shown as the mean ± SEM (n = 10 whole-muscle sections per group) and are displayed as the percentage of control. The area fraction of NG2 staining or the pericyte coverage in *Pf4-cre⁻ C5ar1^fl/fl* hindlimb muscle sections represent 100%. *p < 0.05 or <0.01. Two-sided Student's t test in **e**, **g**.

impede the inhibitory effect of C5a-conditioned platelet supernatant on endothelial tube formation. Indeed, we found no significant differences between tube formation with vehicle control-stimulated platelet supernatant and C5a-conditioned platelet supernatant from platelets preincubated with Akt, PI3K, and PKC kinase inhibitors (Fig. 7c).

To further assess the role of CXCL4 in platelet C5aR1-mediated inhibition of angiogenesis, we isolated primary murine lung endothelial cells from *Cxcr3⁻/⁻* mice. CXCR3 has been described as the endothelial receptor that mediates antiangiogenic effects of CXCL4[35]. The supernatant from C5a-stimulated WT platelets did not inhibit endothelial migration (Fig. 7d) and tube formation (Fig. 7e) in *Cxcr3⁻/⁻* cells compared with vehicle control-stimulated WT platelet supernatant, whereas this effect was present using WT endothelial cells (Fig. 7d). Moreover, we observed no increased endothelial tube formation of *Cxcr3⁻/⁻* endothelial cells using supernatant from C5a-conditioned versus vehicle control-stimulated *C5ar1⁻/⁻* platelets (Fig. 7e).

We continued to study the mechanisms, by which platelets control angiogenesis after C5a stimulation in vivo using a mouse model, which allowed us to selectively add C5aR1⁺ or C5aR1⁻ platelets. For this purpose, we used the in vivo Matrigel plug assay, in which vessel ingrowth into an extracellular matrix-like gel, which is subcutaneously injected into mice, is stimulated by endothelial growth factors[12]. Similar to ischemic hindlimb tissue (Fig. 1a), we observed complement activation upon induction of vessel growth, as verified by the deposition of C3b (Fig. 7f). Having confirmed the presence of complement activation, we considered the Matrigel plug assay a suitable model to further determine the functional role of platelets in vessel formation. Consistent with our findings of improved revascularization after induction of hindlimb ischemia as a consequence of C5aR1 deficiency (Fig. 3), Matrigel plugs in *C5ar1⁻/⁻* mice displayed a larger area of neovascularization after 7 days, indicating a higher level of angiogenic factor-induced vessel growth (Fig. 7g). To further delineate the cell-specific role of C5aR1 on platelets, we resuspended freshly isolated murine platelets in Matrigel supplemented with basic fibroblast growth factor (bFGF). WT platelets inhibited growth factor-induced

angiogenesis in *C5ar1⁻/⁻* mice, which could not be observed for *C5ar1⁻/⁻* platelets (Fig. 7h). Next, we injected Matrigel into *C5ar1⁻/⁻* mice and supplemented it with WT platelets or platelets isolated from *Cxcl4⁻/⁻* animals. Compared to *Cxcl4⁻/⁻* platelets, WT platelets inhibited growth factor-induced angiogenesis in *C5ar1⁻/⁻* mice (Fig. 7i). In another set of experiments, we systemically inhibited CXCL4 by intravenous injection of an anti-CXCL4 blocking antibody or control IgG as previously described[36]. Matrigel plugs from *C5ar1⁻/⁻* mice supplemented with WT platelets displayed an increased level of growth factor-induced angiogenesis when CXCL4 was inhibited compared to controls (*C5ar1⁻/⁻* mice supplemented with WT platelets + IgG injection; Fig. 7j). Finally, we examined whether the C5a–C5aR1 axis can be influenced pharmacologically in the context of vessel growth. Matrigel was supplemented with the C5aR1 antagonist PMX53 or the respective control peptide. C5aR1 inhibition resulted in an increase of growth factor-induced angiogenesis as compared to control (Fig. 7k, l).

In the hindlimb ischemia model, we detected CXCL4 deposition in muscle tissue 1 week after induction of ischemia and CXCL4 was deposited outside of vessels (Fig. 7m). Interestingly, we detected significantly lower levels of CXCL4 in homogenized hindlimb muscles of *C5ar1⁻/⁻* mice subjected to hindlimb ischemia than in those of WT littermate controls (Fig. 7o). These findings strongly support our hypothesis that C5a-induced CXCL4 is an important negative regulator of ischemia-induced revascularization in vivo. Analyzing histological sections, we found significantly less CXCL4 deposition in *C5ar1⁻/⁻* mice (Fig. 7m, n). When we specifically targeted C5aR1 with the C5aR1-specific antagonist PMX205 in vitro, C5a-induced CXCL4 secretion from platelets was completely abrogated (Fig. 7p). Moreover, using the C5aR1 antagonist PMX205 in vivo, we observed that revascularization was significantly increased during days 4–10, when C5aR1 was inhibited (Fig. 7q), whereas PMX205 had no effect on tail bleeding time (Supplementary Fig. 25).

In conclusion, we provide evidence for a functional crosstalk between cells of hemostasis with the innate immune system during tissue revascularization. More specifically, we demonstrate

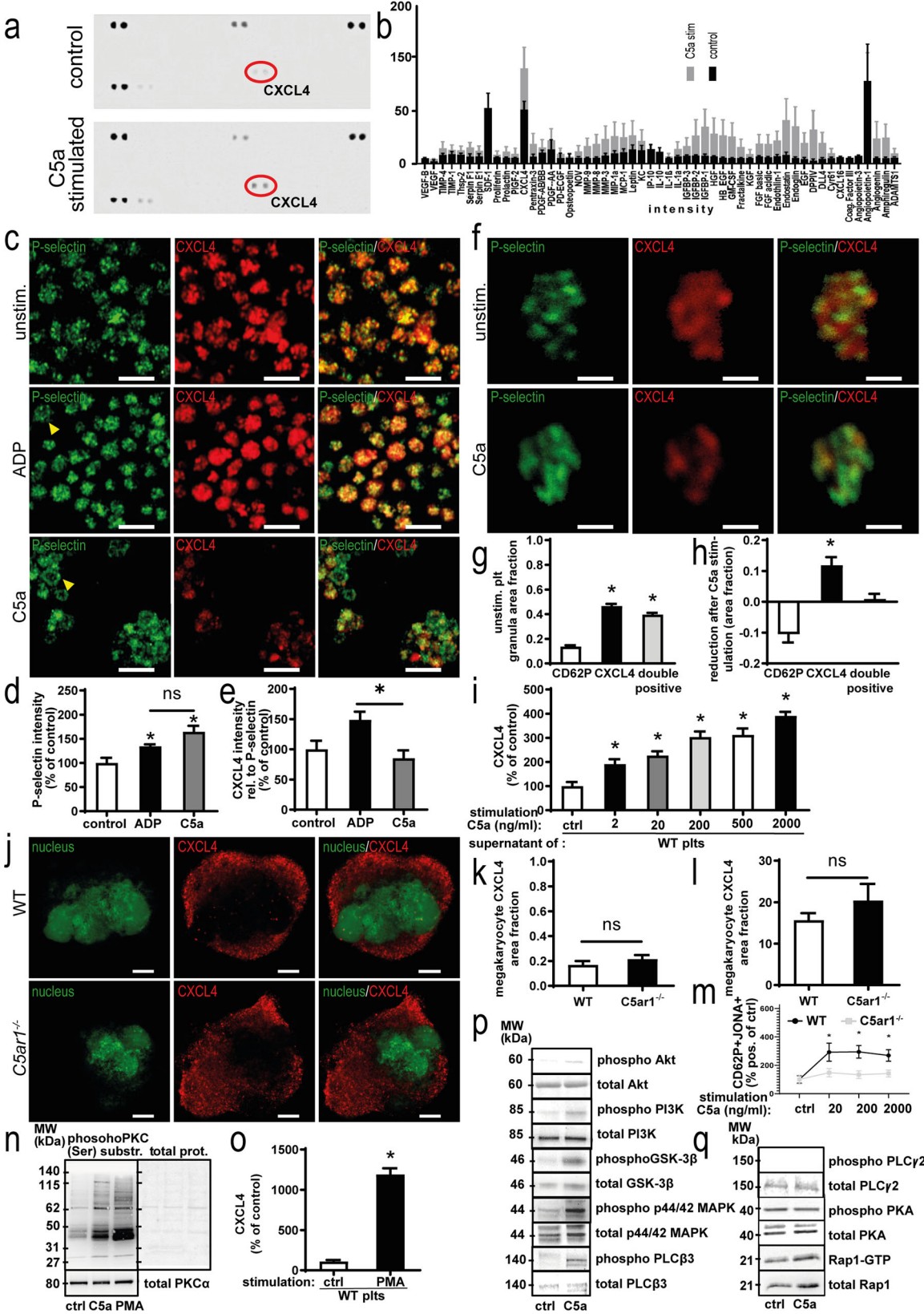

that platelet C5aR1 inhibits endothelial functions in vitro and decreases growth factor-induced angiogenesis and ischemia-induced revascularization in vivo. Mechanistically, we identify platelet-derived CXCL4 production and secretion downstream of C5a/C5aR1 axis activation as a secretory pathway, by which platelets control vessel formation.

## Discussion

Although both complement components and platelet-derived mediators contribute to the regulation of vessel formation[9–12,18,25–27,29,37–43], a functional intersection between platelets and the complement system in the context of revascularization has not been investigated so far. Here we characterized

**Fig. 6 Platelet C5aR1 mediates release of the paracrine effector CXCL4 in an Akt- and PKC-dependent fashion. a** Washed murine WT platelets carefully isolated with inhibitors of platelet activation were stimulated with C5a. The supernatant was analyzed by a membrane-based antibody array. Specific mediators, such as CXCL4 (red circles), were upregulated after stimulation with C5a (bottom) compared with the vehicle control (top). **b** Results of four array repeats were quantified. For most factors, C5a stimulation (gray) induced a slight upregulation compared to vehicle control (black). The strongest increase was observed for CXCL4. Data are shown as the mean ± SEM ($n = 4$ independent experiments) and are displayed as intensity values. $*p < 0.05$. Bonferroni's post hoc analysis was performed; results are displayed in Supplementary Fig. 9. **c** Single-cell staining revealed that murine platelets are activated upon C5a stimulation and release CXCL4. Upon stimulation with ADP and C5a, a ring-like staining pattern was observed, indicating platelet activation (arrows). However, C5a-stimulated platelets exhibit reduced CXCL4 content after stimulation with C5a, which cannot be observed to the same extent in the ADP-stimulated group. ×630 magnification, scale bars represent 5 µm. Images are representative of four independent experiments. **d** Quantification of P-selectin intensity in each platelet reveals upregulation, indicating platelet activation upon stimulation with ADP and C5a. However, no significant difference was detected between ADP and C5a. Data are shown as the mean ± SEM ($n = 5$ images analyzed) and are displayed as the percentage of control. The P-selectin intensity in vehicle control-stimulated WT platelets represents 100%. $*p < 0.05$ **e** Quantification of CXCL4 intensity in relation to P-selectin intensity in each platelet reveals relatively stronger secretion of CXCL4 upon C5a stimulation compared to ADP. Data are shown as the mean ± SEM ($n = 5$ images analyzed) and are displayed as the percentage of control. The CXCL4 intensity in relation to P-selectin intensity in vehicle control-stimulated WT platelets represents 100%. $*p < 0.05$. **f** Single platelets from mice were stimulated with C5a or vehicle control. Granules were quantified by calculating the area of predominantly green staining within the platelets (P-selectin-predominant), predominantly red areas (CXCL4-predominant) as well as overlayed yellow areas (P-selectin-CXCL4 double positive). ×630 magnification, scale bars represent 1 µm. Images are representative of >100 analyzed single platelets. **g** As described in Supplementary Fig. 10, granule area fractions were quantified. We observed a distinct distribution pattern of granules, with a similar amount of CXCL4-predominant as well as P-selectin-CXCL4 double-positive granules and a significantly lower number of P-selectin-predominant granules, which contain P-selectin over CXCL4. Data are shown as the mean ± SEM ($n = 112$ single platelets analyzed) and are displayed as the area fraction. $p$ (CXCL4 versus double positive) < 0.001, $p$ (CD62P versus double positive) = 0.0022. **h** Upon stimulation with C5a, analysis revealed a reduction in CXCL4-predominant granules, while the area fraction of double-positive granules remains unchanged, while the relative area fraction of P-selectin predominant granules increases, thus demonstrating C5a-induced secretion of a subset of α-granules, which contain CXCL4 over P-selectin. Data are shown as the mean ± SEM ($n = 166$ single platelets analyzed) and are displayed as the area fraction. $p$ (CD62P versus CXCL4) < 0.001. **i** Conventional ELISA confirmed a significant dose-dependent increase in CXCL4 secretion from human platelets after C5a stimulation. Maximum CXCL4 release is reached at a C5a concentration of 2000 ng/ml. Data are shown as the mean ± SEM ($n = 8$ independent experiments containing separate donors) and are displayed as the percentage of control. The CXCL4 protein level of vehicle-stimulated platelet supernatant represents 100%. $*p < 0.05$. **j** Murine megakaryocytes from WT versus $C5ar1^{-/-}$ mice were assessed for distribution of CXCL4-predominant granules (red). Displayed are representative images, nuclei are shown in green. ×630 magnification, scale bars represent 10 µm. **k** As described in Supplementary Fig. 10, granule area fractions were quantified in megakaryocytes from WT versus $C5ar1^{-/-}$ mice. No significant difference could be detected for CXCL4-predominant granules. Data are shown as the mean ± SEM ($n = 10–12$ single megakaryocytes analyzed) and are displayed as the area fraction. **l** RT-PCR analysis detected similar mRNA levels of CXCL4 in megakaryocytes from WT versus $C5ar1^{-/-}$ mice. Data are presented as the mean ± SEM ($n = 5$ independent experiments) and are shown as relative expression in relation to GAPDH. **m** Citrated whole blood from WT and $C5ar1^{-/-}$ mice was stimulated using different concentrations of C5a (20, 200, 2000 ng/ml) and vehicle control and assessed for platelet activation markers by flow cytometry. For the gating strategy, please refer to the "Methods" section. Activated platelets are detected as CD62PJONA++, whereas JONA detects activated GPIIbIIIa. C5a induced platelet activation in WT platelets but not in $C5ar1^{-/-}$ platelets. Data are shown as the mean ± SEM ($n = 4$ independent experiments) and are displayed as the percentage of control. The percentage gated CD62PJONA++ platelets in the vehicle-stimulated group represents 100%. $*p < 0.05$. **n** In order to uncover the signaling downstream of C5aR1 leading to CXCL4 secretion, lysates of WT platelets were generated after vehicle control or C5a stimulation and samples were probed at equal protein concentrations for phospho-proteins as well as non-phosphorylated controls. Platelet C5aR1 ligation induced reproducible PKC substrate phosphorylation (PKC phosphor Ser). As a control, we used the PKC activator PMA. PKC activation was quantified by dividing the phosphorylation signal by the total protein signal as well as unphosphorylated PKC α (Supplementary Fig. 23). Displayed are representative images of at least four independent experiments. **o** We also checked whether PKC activation also entails CXCL4 release by stimulating washed platelets with C5a and PMA. Both C5a and PMA induced significant CXCL4 release. Data are shown as the mean ± SEM ($n = 5–16$ independent experiments) and are displayed as the percentage of control of CXCL4 concentration in platelet supernatant measured by ELISA. The mean fluorescence intensity (MFI) of platelets in the vehicle-stimulated group represents 100%. $*p < 0.01$. **p** Furthermore, C5a induced an upregulation of phosphorylated Akt (Ser473), PI3K (Ser 47), GSK-3β (Ser 9), p44/42 MAPK (Thr202/Tyr204), PLCβ3 (Ser537). Displayed are representative images of at least four independent experiments. For quantification of phosphorylation, please refer to Supplementary Fig. 23. **q** Interestingly, C5a induced no regulation of phosphorylated PLCγ2 (Tyr1217) and PKA (α/β/γ catalytic subunit phospho T197). Activated Rap1 (Rap1-GTP) was analyzed using a Rap1 pulldown assay. No significantly altered amounts of Rap1-GTP could be detected. Displayed are representative images of at least four independent experiments. One-way ANOVA with Bonferroni's post hoc test in **b**, **d**, **e**, **g–i**, **m**. Two-sided Student's $t$ test in **k**, **l**, **o**.

a role for the anaphylatoxin receptor C5aR1 expressed on platelets in ischemic and revascularizing tissue. We found that (i) platelets modulate specific endothelial functions relevant for angiogenesis through C5aR1, (ii) platelet C5aR1 mediates an inhibitory effect on collateral artery formation in ischemia-induced revascularization and (iii) capillary formation and pericyte coverage, (iv) platelets release CXCL4 upon C5a stimulation, and (v) the platelet C5aR1-mediated effect on vessel formation is mediated by preferential CXCL4 release in vitro and in vivo.

C5a receptor 1 is expressed on immune cells, epithelial cells, and endothelial cells[8]. Interestingly, complement anaphylatoxin receptors have been reported to be expressed on platelets[21,24],

which are cells of the hemostatic system, and their expression correlates with platelet activation markers in atherosclerosis, a disease featuring vascular inflammation[22]. Platelets express a variety of complement-regulating proteins (reviewed in ref. [44]). In this study, we observed an increased presence of platelets in revascularizing tissue, and we detected the anaphylatoxin receptor C5aR1 on platelets within angiogenic tissue. Future studies should address whether platelets or platelet-derived factors can be useful as biomarkers to detect or characterize angiogenic tissue[45]. Importantly, we demonstrated the importance of platelet C5aR1 for endothelial functions in vitro and neovascularization in vivo using approaches of platelet reconstitution and pharmacological

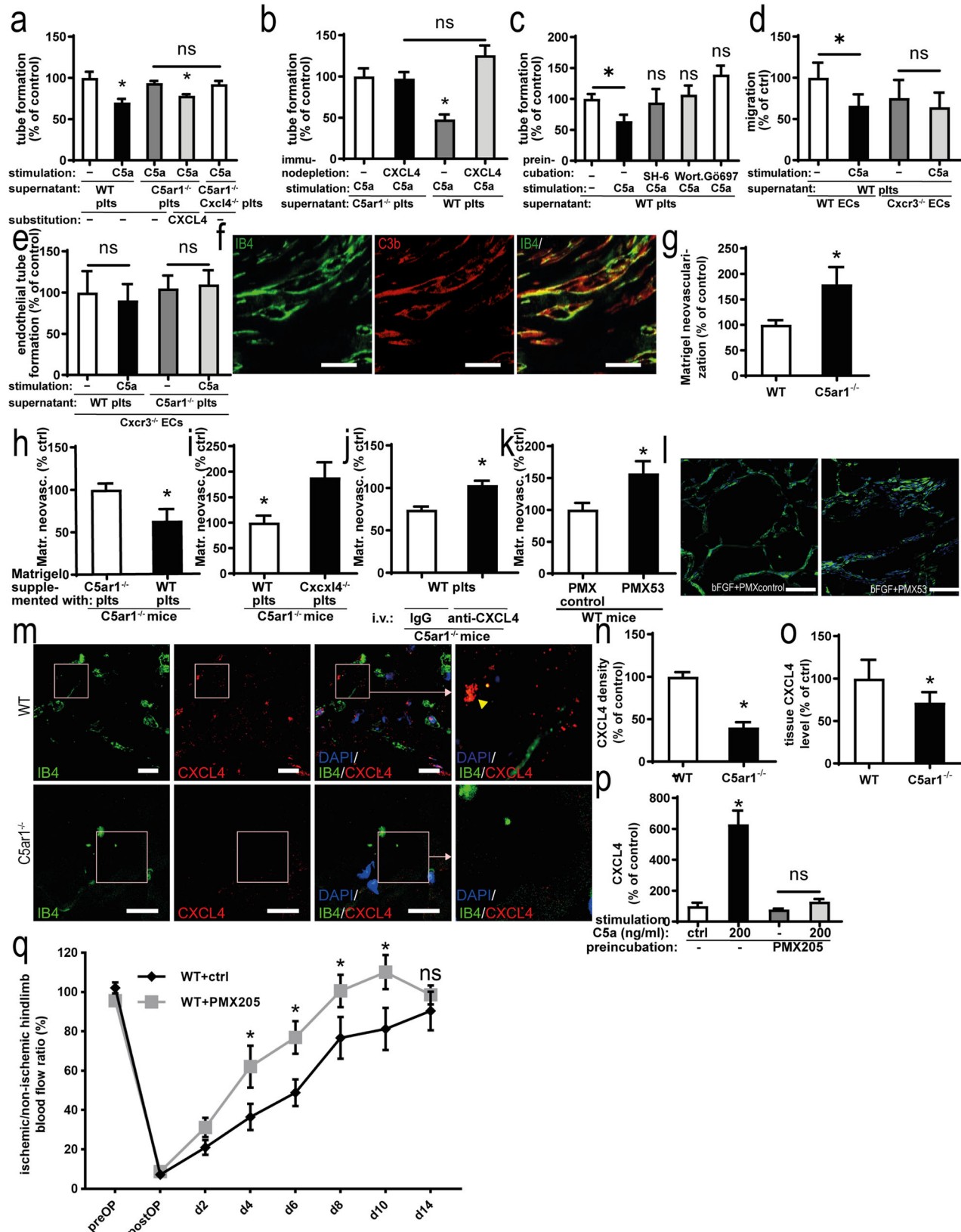

inhibition. Furthermore, we generated a platelet-specific C5aR1-deficient mouse model to specifically assess the platelet C5aR1-mediated effects.

The antiangiogenic relevance of platelets or platelet-derived mediators has been suggested for tumor angiogenesis. During the early stages of tumor growth, it was observed that platelet-derived

thrombospondin 1 serves as a negative regulator of angiogenesis[38]. Here we identified a complement-driven release mechanism for the platelet-derived chemokine CXCL4 as an inhibitor of vessel formation in vivo as well as endothelial functions in vitro. Italiano et al. and others showed that various proangiogenic and antiangiogenic factors are stored in distinct

**Fig. 7 Platelet C5aR1-induced inhibition of neovascularization is dependent on the secretion of CXCL4. a** MHEC-5T were coincubated with the C5a-stimulated supernatant of platelets isolated from WT, $C5ar1^{-/-}$, and $C5ar1^{-/-}Cxcl4^{-/-}$ mice. Furthermore, the supernatant of $C5ar1^{-/-}$ platelets was supplemented with CXL4 (2 μg/ml). C5a-stimulated WT platelet supernatant inhibited endothelial tube formation, which was not detectable in $C5ar1^{-/-}$ and $C5ar1^{-/-}Cxcl4^{-/-}$ platelets. Reconstitution with CXCL4 in the $C5ar1^{-/-}$ group led to a similar level of tube formation as in the WT group. Data are displayed as the mean ± SEM ($n = 5$ independent experiments). The total tube length after treatment with vehicle-stimulated platelet supernatant represents 100%. *$p < 0.05$. **b** CXCL4 was immunodepleted from C5a-stimulated WT as well as $C5ar1^{-/-}$ platelet supernatant, and the impact of this supernatant on endothelial tube formation was assessed. Data are shown as the mean ± SEM ($n = 5$ independent experiments). The total tube length in the group with control-depleted supernatant from C5a-stimulated $C5ar1^{-/-}$ platelets represents 100%. *$p < 0.05$. **c** Freshly isolated washed murine WT platelets were preincubated with vehicle control or kinase inhibitors SH-6, Wortmannin, and Gö976 before stimulation with C5a. The supernatant was then coincubated with MHEC-5T and endothelial tube formation was quantified. Preincubation of platelets with SH-6, Wortmannin, and Gö976 led to a level of tube formation not significantly different from vehicle control-stimulated WT platelet supernatant. Data are shown as the mean ± SEM ($n = 4$–8 independent experiments). The total tube length in the group with control-preincubated vehicle control-stimulated WT platelet supernatant represents 100%. *$p < 0.05$. **d** Primary murine lung endothelial cells were isolated from $Cxcr3^{-/-}$ mice (for endothelial cell quality control refer to Supplementary Fig. 3). After coincubation with C5a-stimulated platelet supernatant, WT endothelial cells displayed decreased endothelial migration in the scratched wound assay. $Cxcr3^{-/-}$ endothelial cells exhibited a significantly smaller decrease in migration compared with WT cells. Data are displayed as the mean ± SEM ($n = 6$ independent experiments). Migration in the WT endothelial cell group treated with vehicle-stimulated WT platelet supernatant subtracted represents 100%. *$p < 0.05$. **e** Similarly, the decrease in endothelial tube formation was significantly smaller in $Cxcr3^{-/-}$ endothelial cells following coincubation with C5a-stimulated platelet supernatant compared with control supernatant and comparable to the level $C5ar1^{-/-}$ platelet supernatant stimulated with C5a. Data are shown as the mean ± SEM ($n = 5$ independent experiments). The total tube length in the group treated with vehicle-stimulated WT platelet supernatant represents 100%. *$p < 0.05$. **f** Similar to the hindlimb ischemia model (Fig. 1a), complement activation (C3b, red) colocalized with vascular structures (IB4, green) within the Matrigel plug 7 days after implantation. ×400 magnification. Scale bars represent 100 μm. Displayed is a representative image of four independent experiments. **g** To study the paracrine effect of platelets after C5a stimulation in vivo, we used the Matrigel model to analyze vascularization after the addition of platelets and soluble mediators. Matrigel was supplemented with bFGF and injected into WT or $C5ar1^{-/-}$ mice. Quantification of neovascularization after 7 days yielded a significantly higher level of growth factor-induced angiogenesis in $C5ar1^{-/-}$ animals than in WT controls. Data are shown as the mean ± SEM ($n = 6$–8 Matrigels per group). The neovascularization area fraction within Matrigels from WT mice represents 100%. *$p < 0.05$. **h** Freshly isolated WT or $C5ar1^{-/-}$ platelets were resuspended in Matrigel, the Matrigel was implanted into $C5ar1^{-/-}$ animals, and vascularization was assessed after 7 days. Data are presented as the mean ± SEM ($n = 7$–8 Matrigels per group). The neovascularization area fraction in Matrigels from $C5ar1^{-/-}$ mice supplemented with $C5ar1^{-/-}$ platelets represents 100% expressed as the area fraction of nuclear staining. *$p < 0.05$. **i** WT platelets or $Cxcl4^{-/-}$ platelets were coinjected with Matrigel into $C5ar1^{-/-}$ animals. Injection of $Cxcl4^{-/-}$ platelets or vehicle control yielded similar levels of growth factor-induced vessel formation. However, injection of WT platelets significantly reversed the phenotype of increased neovascularization in $C5ar1^{-/-}$ mice. Data are shown as the mean ± SEM ($n = 6$ Matrigels per group). The neovascularization area fraction in Matrigels from $C5ar1^{-/-}$ mice re-transfused with vehicle control represents the 100% value expressed as the area fraction of nuclear staining. *$p < 0.05$, WT platelets versus $Cxcl4^{-/-}$ platelets. **j** WT platelets were coinjected with Matrigel into $C5ar1^{-/-}$ mice, and animals were treated systemically with an anti-CXCL4 antibody or IgG control. While the animals that received WT platelets and IgG control showed a reduced level of growth factor-induced angiogenesis, the animals that received WT platelets and anti-CXCL4 antibody did not exhibit significantly different vessel formation levels compared with control animals that did not receive platelet retransfusion. Data are shown as the mean ± SEM ($n = 7$–9 Matrigels per group). The neovascularization area fraction in Matrigels from WT mice without platelet injection and vehicle control treatment represents 100% expressed as the area fraction of nuclear staining. *$p < 0.001$, anti-CXCL4 antibody versus IgG control. **k** Matrigel was supplemented with bFGF and additionally PMX53 or PMX control and injected into WT mice. Quantification of neovascularization after 7 days yielded a significantly higher level of growth factor-induced angiogenesis in the PMX53 group. Data are shown as the mean ± SEM ($n = 7$ Matrigels per group). The neovascularization area fraction within Matrigels from WT mice supplemented with PMX control represents 100%. *$p < 0.05$. **l** Representative immunofluorescent stainings of Matrigel plug sections at ×400 magnification shows Matrigel neovascularization (IB4 in green, nuclei in blue) is increased after supplementation of Matrigel with PMX53. Scale bars represent 100 μm. Image is representative of at least four analyzed plugs. **m** Ischemic hindlimb muscle tissue from WT or $C5ar1^{-/-}$ mice was stained for the presence of CXCL4 deposition (red) 1 week after induction of ischemia. IB4 staining (green) depicts vascular structures, DAPI (blue) nuclei. ×630 magnification, scale bars represent 2 μm. **n** Quantification of CXCL4 abundance by measuring the area fraction of CXCL4-positive staining in whole-muscle sections acquired by tile scanning at ×400 magnification. Data are shown as the mean ± SEM ($n = 10$ whole-muscle sections per group) and are displayed as the percentage of control. The area fraction of CXCL4 staining in hindlimb muscle sections from WT mice represents 100%. *$p < 0.01$. **o** Ischemic muscle tissue from the hindlimb ischemia experiments was homogenized, and samples of equal protein content were probed for CXCL4 concentration using ELISA. Homogenates from $C5ar1^{-/-}$ mice yielded significantly lower CXCL4 levels compared with those from WT mice. Data are shown as the mean ± SEM ($n = 6$–8 muscles processed). The CXCL4 concentration in WT muscle homogenates measured by ELISA represents 100%. *$p < 0.05$. **p** Freshly isolated washed murine WT platelets were isolated and preincubated with PMX205 or vehicle control. C5a-induced CXCL4 secretion was quantified by ELISA. PMX205 inhibited C5a-induced CXCL4 secretion. Data are shown as the mean ± SEM ($n = 5$–25 independent experiments). The CXCL4 concentration in the supernatant of WT platelets stimulated with vehicle control represents 100%. *$p < 0.05$. **q** WT mice treated with the C5aR1 inhibitor PMX205 or control were subjected to hindlimb ischemia and analyzed after 2 weeks. We observed increased revascularization in PMX205-treated animals. Data are shown as the mean ± SEM ($n = 7$ animals per group) and are displayed as the percentage of the perfusion in the contralateral control limb. *$p < 0.05$ Two-sided Student's $t$ test in **g**–**k**, **n**, **o**. One-way ANOVA with Bonferroni's post hoc test in **a**–**e**, **p**. Two-way ANOVA with Bonferroni's post hoc test in **q**.

subpopulations of platelet α-granula and that these factors can be released differentially in response to specific stimuli[25,27]. However, our understanding of the mechanisms underlying this tailored secretion is poor. Angiostatin release from platelets, for example, follows different kinetics than VEGF secretion during thrombus formation[46]. Furthermore, proteinase-activated receptors 1 and 4 can counter-regulate endostatin and VEGF release

from human platelets[47]. Here we identified the C5a/C5aR1 axis not only as a potent inducer of platelet activation but also demonstrate that this pathway drives the release of CXCL4, an α-granule component, which only partially colocalizes with other α-granule components, such as P-selectin.

Platelets contain both proangiogenic and antiangiogenic factors[27]. Accordingly, platelets have also been described as

proangiogenic mediators in conditions, such as ovarian cancer[48]. In fact, platelet-derived products are used by clinicians to foster healing in orthopedic patients[48,49], partially due to their proangiogenic properties. Here we observed a specific inhibitory effect of platelet C5aR1 in ischemia-induced revascularization. Furthermore, platelets are important for the prevention of excessive hemorrhage from newly formed vessels[39]. The net effect that platelets have on vessel formation may depend on the surrounding microenvironment or other factors, which should be further characterized in future studies.

Complement activation in ischemia–reperfusion injury (IRI) is well characterized (reviewed in ref. [50]). However, this work shows complement deposition in hindlimb ischemia tissue without reperfusion. Reactive oxygen species (ROS) production is regarded as the main driver of complement activation in IRI by the classical pathway. Most likely, this mechanism also applies to the hindlimb ischemia model, where ROS production plays a critical role even in the absence of reperfusion[51]. Furthermore, the activated endothelium in the ischemic tissue may bind gC1qR and thereby activate the classical complement pathway[52,53].

The complement system is known to be involved in the regulation of angiogenesis[8,12,18]. An inhibitory role of C5aR1 on revascularization is in accordance with previous reports[10,12]. In an earlier study, we showed that C5aR1 inhibits angiogenesis through the secretion of soluble VEGF receptor 1 (sVEGFR1) from macrophages. Complement-mediated release of sVEGFR1 has been described to be clinically relevant in placental dysfunction[10]. How does this reconcile with our current finding that the complement-mediated inhibition of neovascularization is also dependent on platelets? Vessel formation and growth are ubiquitous mechanisms in organisms and require a finely balanced equilibrium of proangiogenic and antiangiogenic stimuli from different cells[54]. Thus, several cell types may utilize the complement system as a means to modulate angiogenesis and revascularization. The complement system is one of the ontogenetically oldest plasma protein systems and mediates crosstalk among various cell systems in immunity and tissue repair processes[1].

Other studies have reported a decreased level of angiogenesis in $C5ar1^{-/-}$ mice[9] and in mice deficient in complement factor 3[55]. These studies investigated the role of the complement system in laser-induced choroidal neovascularization, a model for age-related macular degeneration, in which the retinal pigment epithelium plays a unique pathophysiological role[56]. Given the versatility of the complement system, it is possible that it regulates angiogenesis in a context-dependent manner, such as stimulating angiogenesis in the eye but eliciting different effects in other tissues and experimental models.

In the present study, we identified a mechanism underlying the inhibitory role of platelet C5aR1 in revascularization. The anti-angiogenic factor CXCL4[57,58] is released from platelets upon stimulation with C5a and mediates this effect in a receptor-specific manner as verified by in vitro and in vivo approaches. At this point, we cannot rule out that other platelet factors besides CXCL4 released upon C5a stimulation mediate the observed revascularization-modulating effects. These factors will have to be identified and studied in future investigations.

CXCL4 is a pro-coagulant, which binds and neutralizes heparins and thus impacts not only on hemostasis but also acts as an inhibitor of angiogenesis. The functions of CXCL4 in vascular homeostasis are complex. In vivo studies using various transgenic mouse models demonstrated an important role for CXCL4 in thrombosis. Neutralization of CXCL4 identified its central role for the anticoagulant effect of heparins[59]. Others have shown in a primate model that physiologically relevant concentrations of CXCL4 stimulate the generation of activated protein C (APC), which suggests that CXCL4 plays a previously underappreciated role in the soluble coagulation cascade[60]. As one of the underlying mechanisms, it was suggested

that CXCL4 binds with relative high specificity and high affinity to thrombomodulin and protein C, interactions which may enhance the affinity of the thrombin–thrombomodulin complex for protein C, thereby promoting the generation of APC[61]. Interestingly, CXCL4 can alter the structure of fibrin, thereby contributing to the sealing of blood clots[62].

Previous findings identified C3a and C5a as activators of platelet activation, aggregation of gel-filtered human platelets as well as serotonin release[63,64]. For C5a, we detected increased platelet activation; however, we found no significant dense granule release from murine platelets. At this point, we cannot entirely explain why C5a induces a2b/b3 integrin activation but not aggregation. Further profound studies are needed to characterize this observation. It is well appreciated that C5 deficiency protects mice from lethal thrombosis. Mice lacking C5 experience milder thrombocytopenia, consumptive coagulopathy, and liver injury than C5-competent mice in a model of histone-induced liver injury[65]. We found that C5-deficient platelets secrete CXCL4 upon C5a stimulation. Interestingly, the absence of C5aR1 had no effect on in vivo hemostasis or bleeding time in our study suggesting an effect on revascularization, which is distinct from the hemostatic functions of activated platelets.

Strikingly, we observed that targeting CXCL4 was sufficient to block the effect of platelet C5aR1 on vessel formation. These findings suggest that C5aR1-transmitted effects can be blocked pharmacologically and that CXCL4 is the predominant angiogenic factor released upon C5a stimulation of platelets. However, at this point, we cannot entirely rule out that in addition to CXCL4 other platelet factors are released upon C5a stimulation and contribute to the observed revascularization-modulating effects.

The C5a receptor 1 is a seven transmembrane G protein-coupled receptor, which transmits its signals via the Gβγ subunit that induces PI3Kγ-dependent Akt phosphorylation and the Gαi subunit, leading to PKA activation[66]. Overall, C5aR1-dependent signaling is complex involving PI3K and Akt-dependent pathways, PKC and MAPKs, IκBα, NfκB, and sphingosin 1 phosphate (S1P)[67]. In T cells, it has been shown that C5aR1 transmits its signal via PI3K and Akt[68,69]. In neutrophils, C5a-induced PI3K signaling has been demonstrated[70], in macrophages PI3K, Akt, MEK1/2, and ERK1/2 signaling[71]. Furthermore, PLCβ has been implicated in C5aR1 signaling, particularly if PKC is involved[72]. Here we demonstrate a signaling pathway involving PI3K, Akt, PLCβ3, PKC, and Erk1/2. No C5a-dependent signal in platelets was detected for PLCγ2, PKA, and Rap-1. This is important, as the C3aR has been shown to signal via Rap1 in platelets, recently[23,24]. A PI3K- and Akt-dependent pathway leading to PKC activation has already been observed for platelets, such as in PAR-1-driven CXCL12 release[73].

We found an increased abundance of CXCL4 in ischemic muscle tissue of C5aR1-deficient mice. Platelets are the major source of CXLC4 in organisms[74]. In vitro, we identified CXCL4 as the major component released from platelets in response to C5a stimulation mediating the inhibitory effect on endothelial functions, such as tube formation and migration. This effect of CXCL4 is in accordance with previous findings[75]. In vivo, arteriogenesis or collateral artery outgrowth is considered the primary mechanism responsible for hindlimb revascularization after femoral artery ligation[76]. Indeed, we observed increased collateral artery formation in platelet-specific C5aR1-deficient mice associated with their phenotype of enhanced revascularization. We observed differences in hindlimb revascularization between the assessed genotypes (WT versus $C5ar1^{-/-}$ or $Pf4\text{-}cre^{+/-}\ C5ar1^{fl/fl}$) after the onset of ischemia. Although we have quantified the vascular network in nonischemic control limbs, we cannot, at this point, entirely rule out that platelet C5aR1 impacts also on the formation of pre-existent collaterals in the mouse hindlimb.

While in the distal hindlimb no differences in artery size or number could be detected, we noted differences in vessel density indicating an additional angiogenesis-related process. This is in accordance with recent reports attributing an important additional role to angiogenesis in hindlimb revascularization[77].

A key to understanding the phenotype of ameliorated revascularization in platelet-specific C5aR1-deficient mice might be the increased presence and an amplified coverage of vessels with pericytes, which we observed in platelet-specific C5aR1-deficient animals. Pericytes have been shown to express CXCR3, the receptor for CXCL4[78]. Exploring the effect of the C5aR1-CXCL4 axis on pericyte biology might be a very interesting avenue for future studies, as pericytes have also been implicated as mediators of tissue injury after acute ischemic events, such as myocardial infarction[79].

Our observations could be of clinical relevance because targeting angiogenesis has become an important therapeutic approach for treating cancer and vascular diseases, such as myocardial infarction[54,80]. Modulation of the complement system is recognized as a promising strategy in drug discovery[81]. The complement system has been demonstrated to be a promising therapeutic target for a growing number of clinical conditions including paroxysmal nocturnal hemoglobinuria (PNH)[82,83] and atypical hemolytic uremic syndrome[84]. The C5 blocking monoclonal antibody (mAb) eculizumab has been successfully used for PNH for >10 years and a specific C5aR1 antagonist (avacopan) has been filed for Food and Drug Administration approval for the treatment of antineutrophil cytoplasmic antibody-associated vasculitis. Translational studies showed that inhibition of C5a and the C5b–9 complex inhibits leukocyte and platelet activation during extracorporeal circulation[85]. Regarding the soluble coagulation system, C5a was suggested as a pharmacological target to treat patients with sepsis, as this approach countered the impairment of the homeostasis between coagulation and fibrinolysis[86]. Moreover, a large number of additional therapeutic modalities that target the complement system are currently in the pharmaceutical pipeline[87]. Finally, the complement system has been evaluated as a therapeutic approach at the preclinical level in a number of immunological and cardiovascular diseases, such as arthritis[88], sepsis[89], IRI[90], fetal injury in antiphospholipid syndrome[91], and abdominal aortic aneurysm[92]. Whether strategies of revascularization targeting the C5aR1 are beneficial in these clinical settings will have to be addressed in further studies. Here we successfully targeted C5aR1 in vivo using a C5aR1 antagonist resulting in increased vessel formation. Applying inhibitors of the complement system that inhibit the C5a/C5aR1 axis may be a promising approach to shift the platelet-mediated response in ischemic diseases such as coronary or peripheral artery disease toward increased revascularization.

In conclusion, our findings emphasize a role for platelets and the complement system in vessel formation and identify the C5aR1–CXCL4 axis as an intersection point between these three systems.

## Methods

**Mice**. All animal procedures were approved by the regional animal care and use committee of the District of Tübingen, Baden-Württemberg (Konrad-Adenauer-Straße 20, 72072 Tübingen, Germany). All animal experiments were performed in accordance with the German law guidelines of animal care. C57BL/6 (WT) mice were originally acquired from Jackson Laboratories but were bred in our own animal facility. Here the light cycle was 12 h, temperature 20–22 °C, humidity 40–60%. C5aR1-deficient mice (CD88, $C5ar1^{−/−}$) have been previously described elsewhere[93]; they were kindly provided by Dr. C. Gerard (Harvard Medical School, Boston, MA, USA) and bred in our animal facility on the C57BL/6 background. Furthermore, we used mice deficient for C5 (B6(Cg)Tg(Ins2-GP)zbz) bred in the animal facility of the University of Lübeck (Germany), which have been described previously[94]. Mice deficient in C5aR2 as well as tdTomato-C5aR2$^{fl/fl}$ knock-in mice were described previously[34]. Furthermore, C3aR-deficient mice (B6.129×1-

C3ar1tm1Raw) were used[95]. Mice deficient for the CXC chemokine receptor 3 (CD183, $Cxcr3^{−/−}$) were purchased from Jackson Laboratories and have been previously described elsewhere[96]. Mice deficient in CXCL4 ($Cxcl4^{−/−}$) were generated and kindly provided by Dr. P. von Hundelshausen (IPEK, Munich, Germany) and have been previously described elsewhere[97]. Briefly, $Cxcl4^{−/−}$ mice were generated from the offspring of CreERT2-$Cxcl4^{L1/L1}$ mice after the ubiquitous deletion of CXCL4 by tamoxifen. Mice double deficient for C5aR1 and CXCL4 ($C5ar1^{−/−}Cxcl4^{−/−}$) were bred in our own animal facility (on a C57BL/6 background). GFP-$C5ar1^{fl/fl}$ mice have been described previously[30]. Briefly, GFP-C5aR1-knock-in mice were generated by gene targeting. (Ac)GFP and an internal ribosomal entry site (IRES) were inserted adjacent to the coding exon of C5ar1. Simultaneously, the AcGFP IRES $C5ar1$ cassette was flanked with two loxP sites. We then crossed these mice with a $Pf4$-cre strain expressing Cre-recombinase under control of the platelet- and megakaryocyte-specific promoter[31] to generate platelet-specific $C5ar1$-knockout mice.

For characterization of this new strain, platelets, leukocytes, and red blood cells were counted using a Sysmex cytometer (Sysmex KX-21N, Görlitz, Germany).

**Genotyping of floxed GFP-$C5ar1^{fl/fl}$ $Pf4$-cre mice**. For genotyping, we used ear biopsies. DNA was extracted from the tissue using a Qiagen Blood and Tissue DNA Extraction Kit (Qiagen, Hilden, Germany), following the manufacturer's instructions. For amplification of the different fragments, we used the DreamTaq PCR Kit (Thermo Fisher Scientific, Waltham, MA, USA) and the following primers: $C5ar1$flox_primer_01F: 5′-TAGAGTTGAGACTCAGAAAGACGG-3′, $C5ar1$flox_primer_02R:5′-GTACACGAAGGATGGAATGGTG-3′, $C5ar1$flox_primer_03F: 5′-CTAGGCCACAGAATTGAAAGATCT-3′, and $C5ar1$flox_primer_04R: 5′-GT AGGTGGAAATTCTAGCATCATCC-3′; GFP_primer_01F: 5′-TAGAGTTGAGA CTCAGAAAGACGG-3′, GFP_primer_02R: 5′-GGGTGGACAGGTAGTGGTTA TC-3′; $Pf4$_cre_primer_01F: 5′-CCCATACAGCACACCTTTTG-3′, $Pf4$_cre_primer_02R: 5′-TGCACAGTCAGCAGGTT-3′, $Pf4$_cre_primer_03F: 5′-CAAATG TTGCTTGTCTGGTG-3′, and $Pf4$_cre_primer_04R: 5′-GTCAGTCGAGTG-CAC AGTTT-3′ (all from biomers.net, Ulm, Germany; see also Supplementary Data 1). PCR was run under the following conditions: 95 °C for 3 min, followed by 35 cycles at 95 °C for 30 s, 68 °C or 65 °C for 30 s, and 72 °C for 150 s, followed by 72 °C for 300 s. Then the samples were loaded onto a 1.5% TAE-agarose gel. The amplification products were detected by ethidium bromide staining.

**Mouse hindlimb ischemia model**. We used a previously described protocol to induce hindlimb ischemia[98]. Briefly, the femoral artery of mice aged 10–12 weeks was ligated immediately distal to the branch point of the caudal femoral artery and the epigastric artery. Tissue perfusion was assessed preoperatively, immediately post-ligation, and at 2, 4, 6, 8, 10, and 14 days after the surgical intervention, as previously described[99,100]. Blood flow in the hindlimb was analyzed using an infrared LDI (Moor Instruments, Axminster, UK) at 37 °C under medetomidine/midazolam (0.5/5 mg/kg) anesthesia. For some experiments, platelet-depleting serum (10 μl per dose in 200 μl phosphate-buffered saline (PBS), Accurate Chemical and Scientific, Westbury, USA) was applied principally as described previously[23,24] on days 1 and 5 by intraperitoneal (i.p.) injection. Successful platelet depletion was monitored in the peripheral blood of mice taken retro-orbitally (Supplementary Fig. 5).

For further experiments, we applied the C5aR1 antagonist PMX205 in the hindlimb ischemia model. As suggested by a recent publication on the pharmacokinetics of PMX205[101], we applied the substance subcutaneously every 24 h at a dose of 1 mg/kg.

Data were analyzed with the Moor LDI image processing software (Moor Instruments) and reported as the ratio of flow in the ischemic versus nonischemic hindlimb. Furthermore, after 4, 7, or 14 days, some mice were sacrificed and systemic perfusion with 15 ml PBS at 37 °C was performed to clear the vessels. Then the gastrocnemius muscles of the ischemic and nonischemic limbs were harvested, snap-frozen or perfused with 4% paraformaldehyde (PFA), and processed for histological analysis.

Images of whole-muscle sections were acquired by tile scanning with a Zeiss AxioObserver.Z1 microscope and Zen acquisition software. Vessels were stained by IB4, and vessel density was assessed as the fraction of IB4-positive structures per whole-muscle area. Arterial structures were visualized by staining for IB4 and SMA. Pericytes were assessed by staining for NG2. Collateral artery diameter was assessed by automatically measuring the perimeter of the five largest arterial structures per whole-muscle section. Arteriolar density was assessed as the area fraction of SMA-IB4 double-positive structures per whole-muscle area. Pericyte density was assessed as the area fraction of NG2-positive structures per whole-muscle area, and pericyte coverage was calculated as the spatial colocalization of NG2 and IB4 within whole-muscle sections. All quantifications were performed using the Image Pro Plus Software (Ver. 7.0, Media Cybernetics). A total of 10 sections, each 200 μm apart, were analyzed per muscle.

**Sample preparation for microCT**. For microCT analysis, 9 days after induction of ischemia, mice were injected with 50 μl heparin (i.p.). After 10 min, the mice were euthanized, the thorax was opened, and the aorta was exposed. A catheter was carefully inserted into the descending aorta and manually fixed with sutures, and

the liver incised several times to permit drainage of blood. Then mice were perfused with PBS (37 °C, 80 ml) containing heparin and nitroglycerin to remove the blood. μAngiofil® (Fumedica AG, Muri, AG, Switzerland) was prepared according to a previously published protocol[102] and then injected through the catheter in the aorta until leakage from the liver was observed. Thereafter, it was left to polymerize for 1 h at room temperature (RT). Legs were collected and immersion-fixed in 2% PFA–PBS for 48 h followed by fixation in 70% ethanol. For tomographic imaging, samples were immersed in paraffin (3 h) before they were covered by a thin layer of paraffin and let air-dry.

**Image acquisition by microCT.** For tomographic imaging, we shaped the samples minimally with a scalpel, wrapped the samples in X-ray transparent melamine resin foam (Basotect, SWILO GmbH, Sta. Maria, Switzerland) and mounted them in a standard sample holder inside a Bruker SkyScan1172 high-resolution micro-tomography machine (Bruker microCT, Kontich, Belgium).

The X-ray source was set to a voltage of 70 kV and a current of 142 μA, with a 0.5-mm Al filter in the beam path. For the relatively low-resolution scans we show here, we recorded a set of 315 projections of $820 \times 1224$ pixels at every 0.6° over a 180° sample rotation. Every projection was exposed for 1410 ms, and three projections were averaged to one to reduce image acquisition noise. To cover the entire leg, we performed an oversize scan with three subscans stacked along the long axis of the leg. This resulted in approximately 35 min of imaging time per sample and an isometric voxel size of 21.6 μm in the final data sets.

**MicroCT reconstruction and vessel visualization.** The projection images were then subsequently reconstructed into a 3D stack of 8 bit gray value PNG images with NRecon (Bruker, Version: 1.7.4.2). After reconstruction, we visualized the legs with MeVisLab (Version 3.1 (2018-06-26 Release), MeVis Medical Solutions AG, Bremen, Germany). We extracted the bone with a gray value threshold-based region growing algorithm with manually placed seed points inside the bone. The vessels were extracted with a "Vesselness" filter, which is calculated as a function of the Hessian matrix. Threshold-based segmentation of this Vesselness measure provided the vessels for a visualization with the MeVis Path Tracer, which is a completely refactored fork of the "ExposureRender" framework by Thomas Kroes[103]. For quantitative analysis of the vascular network within the hindlimbs, thresholding was performed. Then bones were filled by flooding and analysis was limited to "small" vessels in order to exclude bones. By skeletonization, vessel diameters could be measured at centerlines. A graphical representation of the imaging data reconstruction and analysis algorithm is represented in Supplementary Fig. 8.

**Quantitative real-time PCR (qPCR).** Total RNA was isolated from the hindlimb muscles of mice 1 week after the induction of hindlimb ischemia; total RNA was also isolated from the nonischemic contralateral control hindlimb muscles. The tissue was perfused with PBS and ground cryogenically, and RNA was isolated using TriFast (peqgold, VWR, Radnor, PA, USA) followed by purification with the RNeasy Mini Kit (Qiagen, Hilden, Germany) including DNase I treatment (Qiagen) according to the manufacturer's instructions. The RNA was quantified using a NanoDrop spectrophotometer (Thermo Fisher Scientific, Waltham, MA, USA), and RNA quality was evaluated by an Agilent Bioanalyzer 2100 (Agilent Technologies, Santa Clara, CA, USA). To generate cDNA, equal amounts of total RNA were converted into cDNA using the Transcriptor First Strand cDNA Synthesis Kit (Roche Diagnostics, Rotkreuz, Switzerland). The cDNA samples were subjected to qPCR amplification using the QuantiTect SYBR Green PCR Kit (Qiagen) and a LightCycler 480 (Roche). The primers used were as follows: *actin*, 5′-cgtgggccgccctaggcacca-3′ and 5′-ttggccttagggttcaggggg-3′; *Gapdh*, 5′-cccttcatt-gacctccaactacaatggt-3′ and 5′-gaggggccatccacacgtcttctg-3′; *CD42b*, 5′-tggcttcatcc-cacaacaag-3′ and 5′-tttctgaaggactggcacga-3′; CD41, 5′-tggcttcatcccacaacaag-3′ and 5′-tttctgaaggactggcacga-3′; and *C5ar1* 5′-tatagtcctgccctcgctcat-3′ and 5′-tcaccacttt-gagcgtcttgg-3′ (see also Supplementary Data 1). The following PCR cycling parameters were used: 95 °C for 15 min; 50 cycles of 15 s at 94 °C, 30 s at 52 °C, and 30 s at 72 °C.

Dissociation curve analysis was performed on all PCR products to ensure that specific PCR products were generated. PCR was performed in triplicate for every product, and the data were analyzed using the LightCycler software (v.1.5, Roche) with the Ct advanced relative quantification function.

For the detection of CXCL4 mRNA in megakaryocytes, RNA was isolated as described for hindlimb muscle tissue. Transcription into cDNA was performed using Random hexamer primer mix (Carl Roth GmbH, Karlsruhe, Germany), MMLV RT (Thermo Fisher Scientific), and RiboLock (Thermo Fisher Scientific). Equal amounts of cDNA were applied for qPCR amplification using Luna Universal qPCR Master Mix (New England Biolabs, Ipswich, MA) and 7900 HT Real-Time PCR System (Applied Biosystems). The following primers were used: cxcl4 5′-ttcagtggcaccctcttgac-3′ and 5′-atctccatcgcttctcttcgg-3′; gapdh 5′-gaccacagtccatgccatcac-3′ and 5′-ccgttcagctctgggatgac-3′. The following PCR cycling parameters were used: 95 °C for 10 min, 40 cycles of 15 s at 95 °C and 60 s at 60 °C, and finally 15 s at 95 °C, 15 s at 60 °C, and 15 s at 95 °C. The data were analyzed using the Delta-Delta-CT-Method.

**In vivo Matrigel plug assay.** The in vivo Matrigel plug assay was performed as previously described[12] with some modifications. Two aliquots of Matrigel (0.5 ml, Corning, Tewksbury, MA, USA) containing recombinant hirudin (22.4 U/ml, Merck, Darmstadt, Germany) supplemented with bFGF (150 ng/ml, PeproTech, Rocky Hill, NJ, USA) were subcutaneously injected into the mid-abdominal region of mice, one aliquot on each side. Hirudin was used as an anticoagulant because the classical anticoagulant heparin that is usually used in Matrigel plug assays has been shown to inhibit complement activation[104]. Furthermore, heparin is known to form complexes with CXCL4[59]. In some experiments, Matrigel was supplemented with freshly isolated murine platelets ($10^8$/ml Matrigel) from various knockout mice, in addition to bFGF, as indicated in the figure legends. In rescue experiments, the Matrigel injected into knockout animals was supplemented with WT or knockout platelets. Other Matrigel experiments involved intravenous injection of a blocking anti-CXCL4 antibody (10 μg/mouse, rat anti-mouse IgG2b, Clone 140910, R&D, Minneapolis, MN, USA) or control IgG2b (BioLegend, San Diego, CA, USA) as previously described[36]. Furthermore, in some experiments, Matrigel was supplemented with the C5aR1 antagonist PMX53 (AcF[OPdChaWR], Tocris Bioscience, R&D) or a control peptide (PMXcontrol; AcF[OPdChaAdR], Tocris) at 5 μg for each Matrigel plug (500 μl), which has been reported as a successful dose for local injection, previously[105].

After 7 days, mice were sacrificed, and the Matrigel plugs were fixed with 4% PFA, processed for histology (frozen sections), and stained with hematoxylin and eosin using standard staining protocols and reagents. Bright-field images were obtained with a Nikon Optishot-2 microscope equipped with a ×2 plan-apochromat (N.A. 0.08) objective lens and a digital sight DS-5M camera using the Nikon NIS elements BR software (v.3.2, Nikon Instruments, Tokyo, Japan) for image acquisition and analysis. For readout, the ratio of the vessel area to the Matrigel plug area was calculated. Furthermore, immunofluorescent staining using IB4 was performed to visualize plug neovascularization as described in immunofluorescence microscopy studies.

**Intravital microscopy and bleeding time.** Bleeding time experiments were carried out principally as described before[24]. Briefly, mice were anesthetized, and a 3-mm segment of the tail tip was removed with a scalpel. Tail bleeding was monitored by gentle absorption of the blood with filter paper at 20-s intervals without making contact to the wound site. Intravital microscopy and induction of platelet thrombus formation in vivo were carried out as described before[106].

**Human samples.** For experiments with human material, written informed consent was received from participants prior to inclusion in the study. The study was approved by the institutional ethics committee (270/2011BO1) and complies with the Declaration of Helsinki and the good clinical practice guidelines.

**Flow cytometry.** Most flow cytometry stainings were performed in citrated whole blood, which was drawn from mice as described below. Blood was diluted 1:5 using Tyrode's solution (pH 7.4, supplemented with 1 mM $CaCl_2$ and 1 mM $MgCl_2$). In some experiments, blood was stimulated with the following agonists for 10 min at 37 °C: ADP (5, 20, 100 μM, Chrono-log Corporation, Havertown, PA, USA); CRP (0.5, 2, 5 μg/ml; CambCol Laboratories, Cambridge, UK); C5a (2, 20, 200, 2000 ng/ml; R&D). After stimulation, blood was diluted 1:5 once again to stop activation and staining was performed at RT for 30 min using the following antibodies: anti-mouse CD88-APC (C5aR1, clone 20/70, Biolegend, #135808); rat anti-mouse Cd62P-FITC (P-selectin, clone Wug.E9, Emfret Analytics, Eibelstadt, Germany, #D200); rat anti-mouse JONA-PE (integrin aIIbß3(activ.form), clone JON/A, Emfret Analytics, # D200); anti-mouse GPVI-FITC (Rat IgG2a, clone JAQ1, Emfret Analytics, #M011-1); anti-mouse CD61-PE594 (Biolegend, #104321v); anti-mouse CD40L (IgG MR1, Biolegend, # 106512); anti-mouse CD45-BV605 (Biolegend, #103139); anti-mouse CD41-PacificBlue (Biolegend, #133932); anti-mouse CD63-PECy5.5 (clone NKI/C3, Novus Biologicals, Littleton, CO, USA, #NBP2-34694PECY55); anti-mouse GPII-bIIIa-FITC (anti-integrin αIIbβIII, clone Leo.F2, Emfret Analytics, #M025-1); anti-mouse CD49e-PE, (Biolegend, #103805); anti-mouse CD42b-PE (anti-GPIbα Rat IgG2b, clone Xia.G5, Emfret Analytics, #M040-2); anti-mouse integrin α5 chain-FITC (clone Tap.A12, Emfret Analytics, #M080-1); anti-mouse GPIX-FITC (CD42a Rat IgG1, clone Xia.B4, Emfret Analytics, #M051-1); anti-mouse CLEC-2-PE (CLEC1B, clone 17D9, Biolegend, # 146103). After staining, cells were fixed using freshly prepared 4% PFA solution and diluted using FACS buffer (PBS, 0.5% bovine serum albumin (BSA), 0.1% Na-Azide). Gating was performed by CD41 and CD45 as well as FSC and SSC at a logarithmic scale. Platelets were defined as $CD45^-CD41^+$ and differentiated from debris using FSC and SSC (see also Supplementary Fig. 26).

For the detection of C5aR2 expression in platelets, platelets were isolated from tdTomato-C5aR2 reporter mice and controls. The gating antibody CD41-PacificBlue was already added to the whole blood during platelet isolation. Acquisition was performed without prior fixation.

For detection of calcium signaling in platelets using flow cytometry, we used a previously published protocol[107]. Briefly, platelets were isolated using activation inhibitors (as described below), washed, quantified, and loaded with calcium-sensitive dyes fluo-4 and fura-red as well as a CD41-PacificBlue as gating antibody for 20 min at 37 °C. Platelets were then resuspended at a concentration of $10^4$/μl in

a buffer containing ionomycin (3 µg/ml), nigericin (2 µg/ml), and CCCP (10 µM). After addition of calcium, level was acquired for 30 s, then the stimulant was added.

In another experiment, platelets isolated from Matrigels were stimulated with ADP (0.1 mM, Chrono-log Corporation) for 30 min at 37 °C and stained with CD62p-FITC (rat anti-mouse P-selectin, clone Wug. E9) or an appropriate isotype control (rat IgG1, both from Emfret Analytics) for the assessment of platelet activation.

Most flow cytometry experiments were performed using a Beckman Coulter Cytoflex S 4-laser cytometer (Beckman Coulter, Krefeld, Germany) immediately after sample preparation and staining. Unless otherwise stated, specific mAb binding was expressed as the mean fluorescence intensity (MFI) of 25,000 events in the target gate, and data were analyzed using the CytExpert software (v.2.4, Beckman Coulter).

For the characterization of C5aR1 expression on immune cells in $Pf4$-$cre^-$ GFP-$C5ar1^{fl/fl}$ and $Pf4$-$cre^+$ GFP-$C5ar1^{fl/fl}$ mice, 200 µl of citrate whole blood were used per sample. Ten microliters of FcR Block (Miltenyi Biotec) was added to each sample followed by 50 µl of antibody dilution of pre-titrated antibodies. Antibodies were F4/80-BV785, CD11b-BV605, and CD14-APC (all BioLegend) and CD45 PE-Cy5.5 (eBioscience) and incubated for 20 min in the dark at RT. Samples were treated with ammonium chloride red blood cell lysis and washed with PBS. After spinning down again, cells were stained with Zombie NIR (dead cell stain, BioLegend, 1:600) for 15 min in the dark. Cells were washed and resuspended in FACS-Buffer for analysis. Analysis was performed on a LSRFortessa flow cytometer (BD Bioscience) with a violet (402 nm), blue (488 nm), yellow/green (561 nm), and red (640 nm) laser. Specific mAb binding was expressed as the MFI, and data were analyzed using the FloJo software (v.10, Tree Star, Ashland, OR, USA).

**ATP release**. Aggregation of platelets was estimated with light transmission aggregometry using a lumiaggregometer Model 700 (ChronoLog). Washed murine platelets were adjusted to a concentration of $250 \times 10^3$ platelets per µl in Tyrode buffer pH 7.4 and were preincubated with PBS or 20 ng/ml C5a for 10 min. After adjusting the measurement according to the manufacturer's protocol, platelets were activated with 0.25 µg/ml CRP or with 2.5 mU/ml thrombin for 10 min at 37 °C and a stirring speed of 1000 rpm. Analysis was performed using the aggrolink8 software (ChronoLog).

**Aggregometry**. Aggregation of platelets was estimated with light transmission aggregometry using a lumiaggregometer Model 700 (ChronoLog). Washed murine platelets were adjusted to a concentration of $250 \times 10^3$ platelets per µl in Tyrode buffer pH 7.4 and were preincubated with PBS or 20 ng/ml C5a for 10 min. After adjusting the measurement according to the manufacturer's protocol, platelets were activated with 0.25 µg/ml CRP, with 2.5 mU/ml thrombin, or further agonists for 10 min at 37 °C and a stirring speed of 1000 rpm. Analysis was performed using the aggrolink8 software (ChronoLog). For human samples, platelet-rich plasma (PRP) was generated as described above and used for aggregometry with platelet-poor plasma as a control; stimulation was performed using ADP and CRP as well as C5a at defined concentrations.

**Immunofluorescence microscopic studies**. From ischemic murine hindlimb muscle tissue, 8-µm-thick sections of the gastrocnemius muscle were processed for immunofluorescence staining. Matrigel sections were processed accordingly. For PFA-perfused tissue, we performed antibody retrieval as described previously[108]. Briefly, we rehydrated tissue in PBS and then heated the tissue to 70 °C in 1 mM EDTA (pH 8.00) three times. Snap-frozen tissue was fixed for 10 min in ice-cold acetone and blocked with 10% goat or donkey serum and 1% BSA for 30 min at RT. Subsequently, the sections were incubated with primary antibody in 1% blocking serum containing CaCl₂ and 0.2% Tween 20 overnight at 4 °C. Furthermore, when tertiary amplification was used, a biotin blocking kit (Thermo Fisher) was applied.

As primary antibodies, we used Isolectin GS-IB4 Alexa Fluor 594 (Invitrogen, Carlsbad, CA, USA, 1:100), rat anti-mouse anti-C3 antibody (clone 11H9, ab11862, Abcam, Milton, UK, 1:20), α-actin antibody (clone1A4, sc-32251, Santa Cruz Biotechnology, Dallas, TX, USA, 1:20), rabbit anti-mouse anti-NG2 chondroitin sulfate proteoglycan antibody (Merck, AB5320, 1:20), rabbit polyclonal anti-CXCL4 antibody (PAA172Mu01, Cloud Clone Corp., Houston, TX, USA, 1:20), goat anti-mouse C5aR1 antibody (CD88, Santa Cruz, clone P14, sc-3124, 1:200), and rat anti-mouse GPIb antibody (CD42b, clone POP-B, Emfret Analytics, 1:100). For visualization or secondary amplification, we incubated samples with donkey anti-rat highly cross-adsorbed IgG coupled with biotin (Thermo Fisher, 1:500), donkey anti-rabbit highly cross-adsorbed IgG Alexa Fluor 647 (Thermo Fisher, 1:100), donkey anti-goat Alexa Fluor 488 antibody (Invitrogen, 1:200), goat anti-rat Alexa Fluor 568 antibody (Invitrogen, 1:1000), goat anti-rat cross-adsorbed IgG Alexa Fluor 488 (Thermo Fisher, 1:500), goat anti-rabbit Alexa Fluor 488 antibody (Invitrogen, 1:500), and goat anti-rabbit IgG coupled with biotin (ab6720, Abcam, 1:200). If required, we performed tertiary amplification using Streptavidin Alexa 546 (Thermo Fisher, 1:200) in a buffer containing 100 mM Tris-HCl, pH 7.5, 150 mM NaCl, 0.3% Triton X-100, and 1% BSA. Finally, the sections were probed with DAPI (Sigma Aldrich, Taufkirchen, Germany, 1:3000) to visualize the nuclei.

The stained sections were mounted on glass slides, and image acquisition of muscle sections was performed as described earlier. Higher-magnification images were taken using a Zeiss LSM 800 confocal laser scanning microscope with the

Zeiss ZEN 2.3 (blue edition) software. Subsequent image analysis was performed with Image Pro Plus (Ver. 7.0).

For single-cell staining, we blocked samples with 10% donkey serum and used the following primary antibodies: mouse monoclonal P-selectin antibody (CTB201, sc-8419, Santa Cruz, 1:50), rat anti-mouse CD88 antibody (C5aR1, BioLegend, #135815 1:60), and rabbit polyclonal CXCL4 antibody (PAA172Mu01, Cloud Clone Corporation, Houston, TX, USA, 1:12). For visualization, the following secondary antibodies were used: donkey anti-mouse preadsorbed IgG Alexa Fluor 488 (ab150109, Abcam, 1:200), donkey anti-rabbit highly cross-adsorbed IgG Alexa Fluor 647 (Thermo Fisher, 1:100), and donkey anti-rat preadsorbed IgG Alexa Fluor 568 (ab175475, Abcam, 1:200). Images were acquired using a Zeiss LSM 800 confocal laser scanning microscope with Zeiss ZEN 2.3 (blue edition) software or an Olympus Fluoview 1000. Subsequent image analysis was performed with Image Pro Plus (Ver. 7.0). For analysis of single platelet granule composition, we took snapshots of single platelets from confocal immunofluorescence images, and granules were differentiated by distinct regions of interest in a fluorescence scatter plot using Image Pro Plus (Supplementary Fig. 10).

For the characterization of platelet abundance in ischemic hindlimb tissue, CD42b-positive signals were quantified according to size characteristics in an automated fashion as single platelets or microthrombi using Image Pro Plus (Ver. 7.0).

For the staining of megakaryocytes, we blocked samples with 5% donkey serum and used the following primary antibodies: purified rat anti-mouse CD62P (BD Pharmingen, San Jose, CA, USA, 1:50) and rabbit polyclonal CXCL4 antibody (PAA172Mu01, Cloud Clone Corporation, 1:12). The following secondary antibodies were used: Donkey F(ab')2 anti-rat IgG (H + L)-Alexa Fluor 647 (#712-606-153, Dianova, Hamburg, Germany, 1:250) and Donkey anti-rabbit IgG (H + L)-Cy3 (#711-165-152, Dianova, 1:250). Cell nuclei were stained using YOPRO (green signal). Images were acquired using a Zeiss LSM 800 confocal laser scanning microscope with Zeiss ZEN 2.3 software, and analysis was performed by the Image Pro Plus software as illustrated in Supplementary Fig. 10.

**Isolation of human and murine platelets and generation of platelet releasate**. Human and mouse platelets were isolated from human or mouse blood as previously described[109,110]. Briefly, human venous blood was donated by healthy volunteers. Blood was drawn from the antecubital vein into acid–citrate–dextrose (ACD) buffer and centrifuged at $430 \times g$ for 20 min. PRP was removed and added to HEPES-buffered Tyrode's solution (2.5 mM HEPES, 150 mM NaCl, 1 mM KCl, 2.5 mM NaHCO₃, 0.36 mM NaH₂PO₄, 5.5 mM glucose, and 1 mg/ml BSA, pH 6.5) and subsequently centrifuged at $900 \times g$ for 10 min. The resulting platelet pellet was resuspended in HEPES-buffered Tyrode's solution (pH 7.4, supplemented with 1 mM CaCl₂ and 1 mM MgCl₂). Platelets were then preincubated with a C5aR1 antagonist (10 µM; PMX53, AcF[OPdChaWR], Tocris) for 30 min at 37 °C or control peptide (AcF[OPdChaAdR]) as previously described[111]. Subsequently, platelets were coincubated with endothelial cells to assess the impact of platelet C5aR1 blockade on endothelial tube formation in vitro.

To isolate murine platelets, blood was drawn from the heart or the retro-orbital plexus, collected in ACD buffer, and centrifuged at $120 \times g$ for 20 min. PRP was resuspended in HEPES-buffered Tyrode's solution (pH 6.5) and then centrifuged at $2600 \times g$ for 10 min. The platelet pellet was carefully resuspended in HEPES-buffered Tyrode's solution (pH 7.4). HEPES-buffered Tyrode's solution was supplemented with 1 mM CaCl₂ and 1 mM MgCl₂. The platelet content was quantified using a Sysmex cytometer (Sysmex KX-21N, Görlitz, Germany) and was adjusted to the required concentration. Subsequently, either the platelets were stimulated, the supernatant was collected or lysed, and the platelets were used for single-cell staining or the platelets were used at specific concentrations for coincubation with endothelial cells, resuspended in Matrigel for injection into mice, or used in flow cytometric analyses. For some experiments, platelet isolation was performed using activation inhibitors prostacyclin (0.5 µM, sigma Aldrich) and apyrase (0.02 U/mL, Sigma Aldrich).

As agonists for stimulation of murine as well as human platelets, we used CRP (CambCol) at specified concentrations, ADP (Chrono-log Corporation), C5a (R&D), thrombin (Roche), TRAP-6 (TRAP, Diasys, Holzheim, Germany), thromboxane A2 (U46619, TxA2, R&D), phorbol 12-myristate 13-acetate (Abcam) at 100 ng/ml (approximately 160 nM), and the C5aR2 agonist P32 (used at 1 µM), which has been described previously[112]. For some experiment, platelets were preincubated with the C5aR1 antagonist PMX205 (used at 15 µM for 30 min at 37 °C, Tocris) or control peptide. Subsequently, platelets were stimulated with C5a.

If not otherwise stated, stimulation with C5a (R&D) was performed at a concentration of 20 nM for 10 min at 37 °C as previously described for other cell types[12]. Following C5a stimulation, the supernatant was collected by centrifugation at $14,000 \times g$ at 4 °C and analyzed or used to stimulate endothelial cells. The pellet was lysed using RIPA buffer supplemented with protease/phosphatase inhibitor (Halt, Thermo Fisher) and analyzed by western blotting.

For single platelet staining, 5 µM ADP was used along with C5a for 5 min at RT or in other experiments for 10 min at 37 °C. After stimulation, platelets were fixed using PFA. Resting, C5a-stimulated, or ADP-stimulated fixed platelets were immobilized on poly-L-lysine-coated coverslips and stained using the antibodies and staining conditions specified above.

For lysate preparation, $100 \times 10^6$ platelets were lysed using RIPA buffer and protease inhibitor (Halt, Thermo Fisher) in a volume of 100 μl. In other experiments, the supernatant of C5a-treated platelets was immunodepleted of CXCL4 following a protocol that was previously described[12]. Briefly, supernatants were incubated overnight at 4 °C with 100 μg/ml anti-CXCL4 antibody (rat IgG2b, MAB 595, R&D) or IgG2b control (BD Biosciences, San Jose, CA, USA). Then washed protein G-coupled Sepharose beads (4 Fast Flow, GE Healthcare, Little Chalfont, UK) were added to the supernatant and incubated for 2 h at 4 °C, and then immune complexes were removed by centrifugation at $14,000 \times g$ for 5 min. The resulting CXCL4-depleted supernatant was subsequently incubated with endothelial cells.

**Enzyme-linked immunosorbent assay.** Platelet releasates were analyzed by ELISA using a mouse PF4/CXCL4 Quantikine ELISA Kit (R&D), a membrane-based antibody array (Proteome Profiler Mouse Angiogenesis Array Kit, ARY015, R&D), a Serotonin ELISA Kit (BA E-8900, LDN, Nordhorn, Germany), a Hexosaminidase B (HEXb) ELISA Kit (SEA637Mu, Cloud Clone Corporation), a Mouse VEGF ELISA Kit (ab209882, Abcam), a Thrombospondin 1 ELISA Kit (THBS1, ABIN6574175, Antibodies-Online, Aachen, Germany), an endostatin COL18A1/ES ELISA Kit (Mouse collagen type XVIII α 1 Endostatin ELISA Kit, MBS701673, MyBioSource.com, San Diego, USA), and a TIMP-1 ELISA Kit (Mouse TIMP-1 Quantikine ELISA, MTM100, R&D). For human samples, we used a human CXCL4/PF4 Quantikine ELISA Kit (R&D), a human thrombospondin-1 Quantikine ELISA Kit (R&D), a human PDGF BB ELISA Kit (ab100624, Abcam), and a human Endostatin ELISA Kit (RayBiotech, Peachtree Corners, GA, USA).

For ex vivo analysis, tissue homogenates were prepared from ischemic and nonischemic gastrocnemius muscles by cryo-grinding the tissue with a mortar and pestle after exposure to liquid nitrogen. The pulverized tissue was then resuspended in PBS supplemented with protease inhibitor (Roche) and subjected to sonication twice at maximum intensity while keeping the sample on ice. Debris was removed by centrifugation at $5000 \times g$ for 5 min, and the supernatant was harvested. Samples were adjusted to equal protein concentrations and probed for CXCL4 content using another ELISA Kit (SEA172Mu, Cloud Clone Corp.) according to the manufacturer's instructions.

**Western blot analysis.** Platelet lysates were separated by sodium dodecyl sulfate (SDS)-polyacrylamide gel electrophoresis using a 12% gel and subjected to western blot. Whole-protein detection was performed, and nitrocellulose membranes were incubated with primary rat IgG2b CD88-specific antibody (10/92, Hycult Biotech, Uden, Netherlands). We used goat anti-rat 800 CW (LI-COR, Bad Homburg, Germany) as a secondary antibody and REVERT total protein stain (LI-COR) for normalization. Membranes were scanned with the Odyssey Infrared Imaging System (LI-COR) and analyzed.

For other experiments, platelets were lysed in RIPA buffer (150 mM NaCl, 50 mM TRIS, 0.1% SDS, 0.5% sodium deoxycholate, 1% Triton-X 100, and complete protease inhibitor cocktail (PI)) and boiled in SDS sample buffer for 10 min at 95 °C. Next, samples were separated by SDS-polyacrylamide gel electrophoresis and transferred to polyvinylidene difluoride membranes using the iBlot2 system (Thermo Fisher). Transfer success and total protein content was quantified by Ponceau staining. Membranes were then blocked with 5% skim milk or 5% BSA (for the detection of phosphorylated antigens) in Tris-buffered saline 0.05% Tween-20 and incubated with the following primary antibodies: anti-mouse C3aR (Santa Cruz Biotechnologies; clone D12; 2 μg/ml); thrombospondin 1 (rabbit polyclonal Anti-Thrombospondin-1 Antibody, Proteintech, Chicago, IL, USA, #1 18304-1-AP, 1:2000); phospho PLCγ2 (Tyr1217, Cell Signaling Technology, #3871S, 1:1000), total PLCγ2 (Cell Signaling Technology, Danvers, USA, #3872S, 1:1000); phospho PKA (PKA α/β/γ catalytic subunit phospho T197, Abcam, #ab75991, 1:3000), total PKA (α/β catalytic subunits, Abcam, #ab216572, 1:500); phospho Akt (Ser473, Cell Signaling Technology, #9271S, 1:1000), total Akt (Rabbit monoclonal Akt pan C67E7, Cell Signaling Technology, # 4691S, 1:1000); phospho PI3K (Phospho PI3 Kinase p85 Tyr458/p55 Tyr199, Cell Signaling Technology, #4228S, 1:1000), total PI3K (rabbit monoclonal PI3 Kinase p85 19H8, Cell Signaling Technology, #4257S, 1:1000); phospho GSK-3β (Ser9, Cell Signaling Technology, #9336S, 1:1000), total GSK-3β (rabbit monoclonal anti GSK-3β 27C10, Cell Signaling Technology, #9315S, 1:1000); phospho p44/42 MAPK (Erk1/2 Thr202/Tyr204, Cell Signaling Technology, #9101S, 1:1000), total p44/42 MAPK (Erk1/2, Cell Signaling Technology, #9102S, 1:1000); phospho PLCβ3 (Ser537, Cell Signaling Technology, #29021S, 1:1000), total PLCβ3 (rabbit monoclonal PLCβ3 D9D6S, Cell Signaling Technology, #14247, 1:1000); phospho PKC (Ser) substrate (Cell Signaling Technology, #2261S, 1:1000), total PKCα (Cell Signaling Technology, #2056S, 1:1000).

As loading controls, we used anti-α-Tubulin (Sigma Aldrich, #T5168, diluted 1:4000) and anti-β-actin (Abcam, #8226, 1:1000) and incubated at 4 °C overnight. As secondary horseradish peroxidase (HRP)-conjugated antibodies, we used HRP-conjugated goat-anti-rabbit IgG H&L (Abcam, ab205718) as well as HRP-conjugated goat-anti-mouse IgG H&L (Abcam, ab205719). Antibodies were incubated for 1 h at RT, and membranes were washed thrice with Tris-buffered saline 0.05% Tween-20. Membranes were developed using the LS ECL PRIME WESTERN BLOTTING SYSTEM (GE Healthcare) and scanned with a

Chemiluminescence Imaging System (Biorad, Feldkirchen, Germany) and analyzed.

**Rap1 activation and pulldown assay.** For detection and pulldown of activated Rap1 (GTP-Rap1), a commercially available kit (affinity precipitation assay) was used and performed according to the manufacturer's recommendations (Rap1 Activation Assay Kit, product number 17-321, EMD Millipore, USA)[113].

**Isolation of primary MLECs.** Primary MLECs were isolated according to a previously published protocol[114]. Briefly, lungs were removed from C57Bl/6 or $Cxcr3^{-/-}$ mice, washed in Dulbecco's modified Eagle's medium (DMEM, Gibco, Thermo Fisher) containing 10% fetal bovine serum (FBS, Gibco, Thermo Fisher) + penicillin–streptomycin (P/S, Thermo Fisher), and minced into 1–2 mm³ pieces. The tissue was digested with collagenase type I (2 mg/ml, Gibco, Thermo Fisher) in DMEM containing 0.1% BSA for 3–4 h at 37 °C with occasional shaking, and the resulting suspension was passed through a 70-μm cell strainer. After pelleting, the cells were resuspended in DMEM/F12 (Gibco, Thermo Fisher) containing 20% FBS, P/S, and 100 μg/ml endothelial cell growth supplement (ECGS, Corning) and seeded in T25 culture flasks coated with 0.4% gelatin (Sigma-Aldrich). After incubation overnight, floating and loosely adherent cells were removed by washing with PBS, and new growth medium + ECGS was added. Cells were grown until near confluence, with fresh growth medium + ECGS added daily. Anti-rat Dyna-beads (Invitrogen) were preincubated with rat anti-mouse CD102 antibodies (Clone 3C4, BD) overnight in PBS containing 0.1% BSA and 2 mM EDTA and washed twice with DMEM containing 0.1% BSA. The beads were added to cell culture flasks containing DMEM with 0.1% BSA and incubated for 10 min. Flasks were then washed with PBS to remove unbound Dynabeads. After trypsinization, bead-bound cells were isolated by washing 4× with DMEM 0.1% BSA on a magnet to remove bead-free cells and then seeded into fresh T25 culture flasks coated with 0.4% gelatin. A second purification step was performed when the cells were confluent, and the cells were then transferred to T75 flasks and harvested when near confluence. Cell purity was verified by flow cytometric analysis for the expression of CD102, CD31, and CD144 (Supplementary Fig. 2). All endothelial cell assays using MLECs were performed within 2 weeks of the flow cytometric quality control. Additionally, the murine endothelial cell line MHEC-5T[115] was used and HUVECs were purchased from PromoCell (Heidelberg, Germany) and cultured in DMEM containing 10% FBS or Endothelial Cell Growth Medium (PromoCell). HUVECs were generally used at low passages and grown on culture dishes pre-coated with 0.2% gelatin (Sigma-Aldrich).

**Platelet adhesion assay to endothelial cells and fibrinogen.** Platelet adhesion to endothelial cells was determined by quantifying the number of platelets per endothelial cell area in a static adhesion assay. Briefly, MHEC-5T (150,000/well) were plated onto gelatin-coated coverslips in a 24-well-plate and incubated for 14 h in full medium or 10 h in full medium followed by 6 h under hypoxic conditions (1% oxygen) in a low-glucose DMEM (Gibco). Isolated platelets (5 million/well) were labeled using CD41 antibody (X488, Emfret Analytics) and incubated with the endothelial monolayer for 30 min at 37 °C. Endothelial cells were washed twice with PBS, fixed (1% PFA), and stained with DAPI. Finally, coverslips were placed on slides and imaged using a digital microscope (BZ-9000, Keyence, Osaka, Japan). Images were merged by the BZ-Analyzer software; platelet count was normalized for endothelial area, both were quantified by ImageJ and Image Pro Plus 7.

For the assessment of static adhesion to fibrinogen, wells of a 96-well plates were coated with fibrinogen (#341576, Merck Millipore) at 2 mg/ml at 4 °C overnight. Murine WT or $C5ar1^{-/-}$ platelets were isolated as described and stimulated with ADP (10 μM) and C5a (20 ng/ml) or vehicle control. The 96-well plate was washed with prewarmed 0.9% NaCl and platelets ($1 \times 10^6$/well) were added to the wells and left to adhere for 1 h at RT. Plates were washed twice with 0.9% NaCl, and pictures were taken with a Visiscope inverted microscope (IT404). A particle analysis was performed using the ImageJ software.

**Cell proliferation assay, scratched wound assay, tube-formation assay, and spheroid tube-formation assay.** Endothelial cell proliferation was determined by measuring the total number of adherent cells at different time points. Briefly, MHEC-5T ($5 \times 10^4$ cells/well) or MLECs ($3 \times 10^4$ cells/well) were plated onto 96-well plates and incubated for 6, 18, or 25 h in full medium; washed platelets were resuspended in medium ($5 \times 10^6$ cells/well) or vehicle control. Then the cells were washed once with PBS and quantified. In some experiments, cell proliferation was also assessed using a BrdU-ELISA Kit (Roche) according to the manufacturer's instructions. Readouts were taken after 18 h.

A scratched wound assay was performed as previously described[110]. Briefly, MHEC-5T ($2 \times 10^5$ cells/well) or MLECs ($1 \times 10^5$ cells/well) were plated onto 24-well plates and incubated for 24 h. Subsequently, the confluent cell monolayer was wounded with a plastic pipette tip, generating a 1-mm-wide gap. Cells were then incubated with washed platelets resuspended in medium ($2 \times 10^7$ cells/well) or vehicle control. In other experiments, cells were coincubated with the supernatant of platelets generated as described earlier. As a measure of endothelial migration, the area repopulated with MHEC-5T or MLECs after 16 or 28 h was quantified by phase-contrast microscopy.

For the in vitro tube-formation assay, MHEC-5T, HUVECs, or MLECs ($3 \times 10^4$ cells/well) were plated onto Matrigel-coated 48-well tissue culture plates in endothelial culture medium containing 1% FBS or on ibidi slides (ibidi, Planegg, Germany) with medium containing 2% FBS. Cells were coincubated with washed platelets ($2 \times 10^7$ cells/well), vehicle control, or platelet supernatant. After 6.5, 12, or 24 h, depending on the cell type, tube formation was imaged by phase-contrast microscopy. Furthermore, preincubation of platelets with antagonists, such as PMX53 (30 min at 37 °C, Tocris), was performed as described in the figure legends. Using Axiovision software (v.3.2, Zeiss), the total tube length in multiple high-power fields per well was assessed and quantified as the average total tube length per well.

For the spheroid tube-formation assay, endothelial cell spheroids of defined cell numbers were generated as previously described[116]. In brief, HUVECs were suspended in culture medium containing 0.2% (wt/vol) carboxymethylcellulose (Sigma-Aldrich) and seeded in round-bottom 96-well plates to form a single spheroid per well (400 cells/spheroid). The spheroids were then embedded into rat collagen I (BD), and the spheroid-containing collagen was rapidly transferred into warm 24-well plates and allowed to polymerize for 30 min. Then 100 µl of endothelial cell medium supplemented with platelet supernatant or vehicle control was added to the wells. After 24 h, pictures were acquired using an Axiovert 100 microscope and a Plan-NEOFLUAR 10x objective. In vitro capillary sprouting was quantified by measuring the cumulative length of sprouts per spheroid using the Axiovision software. The mean cumulative sprout length per spheroid was calculated after the evaluation of 10–15 spheroids/condition.

**Embryonic hindbrain angiogenesis model.** At day E11.5, the hindbrains of murine WT and $C5ar1^{-/-}$ fetuses were successfully prepared, isolated, and stained as whole mounts using a previously published protocol[117]. Briefly, isolated hindbrains were fixed in 4% PFA for 2 h at RT and washed three times. Afterwards, they were blocked using 0.1% Triton-X (Sigma-Aldrich) and 10% normal donkey serum (Sigma-Aldrich) in PBS at 4 °C overnight and then incubated with Alexa-488-conjugated *Griffonia simplicifolia* IB4 (1:200, Thermo Fisher) for 3 h at RT. The ventricular plexus was analyzed using a Zeiss LSM5 EXCITER confocal laser scanning microscope (Zeiss) with a ×10 objective. 3D confocal images were reconstructed with ImageJ (NIH). Analysis of the vascular networks was performed using a previously described method[118] and AngioTool software (National Cancer Center, Rockville, MD, USA) as recently described[119].

**Data presentation and statistics.** The results are expressed as the mean ± SEM. All statistical analyses were performed using GraphPad Prism 9. Comparisons between two groups were performed using Student's $t$ test. Comparisons between more than two groups were conducted using analysis of variance (ANOVA), and comparisons between two groups with several time points were performed using two-way ANOVA. Bonferroni's post hoc test was performed for all ANOVA analyses. $p < 0.05$ was considered statistically significant.

**Reporting summary.** Further information on research design is available in the Nature Research Reporting Summary linked to this article.

## Data availability

All relevant data are available upon request from the corresponding author. Source data are provided with this paper.

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

## Acknowledgements

We thank Sarah Gekeler, Birgit Fehrenbacher, Jacob von Esebeck, and Anke Constantz for excellent technical assistance. We also thank Jennifer Axnick and Eckhard Lammert for technical assistance with the hindbrain staining. This work was supported by the Volkswagen Foundation (Lichtenberg program), the German Heart Foundation, the Wilhelm Sander Foundation (to H.F.L), the KFO 274 – Platelets – basic mechanisms and clinical implications, and the DZHK (German Research Centre for Cardiovascular Research), partner site Hamburg/Lübeck/Kiel (STO Projekt F280404). M.G. and H.F.L. are members of the SFB/Transregio 240 funded by the German Research Council (Deutsche Forschungsgemeinschaft, DFG). T.C. was supported by the ERC (END-HOMRET). H.N. is supported by the Clinician Scientist Program of the DZHK (German Research Centre for Cardiovascular Research), partner site Hamburg/Lübeck/Kiel. H.F.L., J.E. and H.N. are supported by the ERA (PerMed JTC2020).

## Author contributions

H.N. and H.F.L. designed the research study. H.N., M.S., L.B., F.E., J.P., M. Mezger, O.B., E.C., D.H., R.H., and H.F.L. conducted the experiments. H.N., F.E., J.P., M. Meusel, O.B., E.C., H.F.L., A.N., C.M.K., R.M.-S., and K.K. analyzed the data. D.S., O.B., R.F., A.V., P.H., and J.K. provided reagents. H.N., R.S., F.E., J.P., D.S., O.B., R.F., B.P., E.C., J.E., I.E., T.C., P.H., J.K., M.G., and H.F.L. wrote the manuscript.

## Funding

## Competing interests

The authors declare no competing interests.
