## [Peer Review File · Nature Communications]

Reviewers' comments:

Reviewer #1 (Remarks to the Author):

The manuscript by Nording et al is an interesting and reasonably solid paper that is attempting to address the link of the complement system to angiogenesis. There were a number of items that I hope the authors will find helpful.

1) A significant finding of the paper is the demonstration of the increase in the angiogenic response in the platelet-specific deletion of C5aR1 mice. The mouse strain used were C57/Bl6 which are well known to demonstrate a robust angiogenic response to hind-limb ischemia and thus anything that shows an improvement over excellent/robust is of somewhat limited value. Performing the studies in strains that have a known impairment in their angiogenic response would be preferable. Even a study of a less specific pharmacologic inhibitor would be helpful for this report.

2) How different are the platelets from the C5aR1 knock out mice compared to the wild type? This is relevant because the authors used a targeted candidate approach to establish the link between the C5aCR1 and CXCL4 with the latter validated from an array of candidates that have known properties. It follows from that approach, that a factor found is expected to demonstrate the known antiangiogenic properties. I do not think there is a right – wrong answer here and of course the difference will depend on the method used to assess for differences but if a large number of differences are found then the specificity of the finding would be less certain. Another approach would be to compliment the data in Fig 6a with a look at secreted angiogenic factors.

3) Do the platelets from the C5aCR1 KO mice act differently in regards to their deposition? This relates to a bigger question of how and where the platelets adhere and deposit within this model and how that relates to the early response difference based on the genetic deletion.

4) Responses based on therapy or genetic deletion in mice following HLI that are present at 3 or 4 days post HLI, which this study shows, are often considered as from differences in arteriogenesis in pre-existing collaterals, as opposed to angiogenesis.

Reviewer #2 (Remarks to the Author):

The authors study the role of platelets on tissue revascularization in a murine hindlimb ischemia model and provide evidence that complement C5a receptor 1 (C5aR1) deficiency specifically on platelets augments limb neorevascularization over 14 days. Mechanistic analysis associates the absence of platelet C5aR1 to increased platelet release of CXCL4 which in turn inhibits endothelial cell function by binding to EC expressed CXCR3. They provide some evidence the effect is relevant in human endothelial cells and platelets using C5aR1 blockade in vitro.

The finding that platelets express C5aR1 and that its absence from platelets alters neovascularization are novel. While statistically significant, the biological significance of the findings is unclear as the effects are modest (approximately 20-30%). In fact, the absence of endothelial CXCR3 (Figure 6) seems to have a much more robust effect than the absence of platelet C5aR1.

Major critical comments

1.) Significance: It is unclear in what clinical/pathophysiologic circumstance this mechanism would be relevant. The authors mention diseases including aortic aneurysm, heart attack, or anti-phospholipid disease but these processes tend to be sudden and catastrophic requiring immediate revascularization to preserve tissue function. Given that the authors showed no change in embryologic vascularization due to C5aR1 deficiency and only see an effect in an acute permanent ischemia model over 14 days, it seems unlikely that C5a/C5aR1 blockade would provide a benefit before the acute event occurred. Further, their biochemical readout for increased neovascularization is in augmented capillaries, is not quantifiable using their analytic approaches, and did not seem to impact the number of larger arterioles, which is what would be needed in this model of large vessel ischemic disease.

2.) Potential role of C5aR2 needs to be addressed. Evidence from multiple groups including co-author J Kohl indicate that C5aR2 binds C5a and can, in some cell types, transmit an activating or inhibitory signal (latter potential involving b-arrestin among other mechanisms). If C5aR2 is present on the platelets (the authors need to test this), absence of C5aR1 could increase C5aR2 signaling, which in turn could account for their findings (more signaling via C5aR2 inhibits CXCR4 release as opposed to less C5aR1 signaling inhibiting CXCR4 release). This needs to be addressed experimentally.

3.) Effect of C5aR1 deficiency on platelet aggregation in the ischemic limb: The authors show that complement activates in the ischemic limb and that platelets aggregate as well, but they do not show whether there are differences in platelet aggregation between WT and C5aR1-deficient conditions. If there are greater numbers of WT platelets forming microthrombi, the finding of neo-vascularization may not be due to any intrinsic signaling of the C5aR1 on platelets but rather via a mechanism of anaphylatoxin-induced platelet trafficking.

4.) Incomplete limb revascularization time course: The authors provide a time course up to 14d post-ischemia looking at revascularization in figures 3 and 4 but the curve of the lines suggests that the WT group will eventually meet the C5aR1 deficient group within a few days/week. If the blood flow curves stayed separate in the long term, this would be a much more interesting and robust finding. If not, this would further weaken the significance of the phenotype. The authors need to extend the finding to beyond 14 days to determine this.

5) Role of C5. The authors need to test C5 deficient mice and controls (available from Jax as Hc1 and Hc0). Understanding whether C5 deficiency leads to similar phenotype as C5aR1 deficiency will add insight regardless of outcome

6) Incomplete data linking CXCL4 and C5aR1 (in addition to the C5aR2 issue above):

-a. In figure 6 the authors show C5a induces platelet-derived CXCL4 and that this effect requires platelet expressed C5aR1. They need to provide biochemical signaling mechanisms through which absence of C5aR1 induces the platelet to increase the release of CXCL4 (related to the comment regarding C5aR2 above).

b. It is noteworthy that supplemental fig 6 shows robust baseline CXCL4 secretion suggesting other mediators. The fold increase upon adding C5a is about 35% over this high spontaneous production, consistent with a modulating but not biologically robust effect of C5a/C5aR1.

-c. In fig 6, the authors show that a) WT (but not C5aR1 KO) platelets stimulated with C5a reduce endothelial tube formation and b) immunodepletion of CXCL4 from the supernatant can augment tube formation. The need to "close the loop" by testing whether CXCL4 immunodepleted C5aR1KO-platelet supernatants behave similarly or that by adding back CXCL4 to C5aR1KO-platelet supernatants they can replicate the findings of the WT conditions.

-d. The data using CXCR3 KO endothelial cells (the putative receptor responding to CXCL4) is similarly incomplete. Figures 6h and i are firstly a bit confusing and misleading because they shift the axis to show it as % inhibition when all the previous figures showed % formation. While there is less inhibition in CXCR3 KO endothelial cells vs WT the authors need to repeat the experiment using C5aR1 KO-platelets to provide test whether the C5aR1-CXCL4 axis is the primary driver of the finding they are describing.

-e. the authors show that with increasing amounts of C5a they can induce cultured platelets to release CXCL4 (Fig 6b) but in figure 6e the authors conclude that CXCL4 release is C5a-C5aR1 specific by showing that C5a stimulation induces a lower CXCL4:P-selectin ratio by IF compared to ADP stimulated platelets. However, they disregard the possibility that C5a more potently augments P-selectin expression than ADP (which is actually suggested by their data in figure 6d). This need further experimental evidence to make the appropriate conclusions.

Minor issues related to complement

1. What is the activating pathway of complement in this system which differs from ischemia reperfusion?

2. Interactive effects of C5aR1 and C3aR1. The authors note in the introduction that C3aR1 on

platelets has been identified as an important mediator of platelet aggregation. Are there compensatory differences in C3aR1 levels in these C5aR1-deficient platelets that may be contributing to their findings?

3. Do platelets express complement regulators and if so which ones?

Reviewer #3 (Remarks to the Author):

The role of platelets in regulating angiogenesis is an important topic given the large array of pro- and anti-angiogenic molecules stored in alpha-granules and released during platelet activation. The concept of differential release of pro vs anti-angiogenic factors based on specific agonists and/or specific granule compartments has been under study for quite some time by several investigators in the field. In this manuscript the authors report 2 complementary sets of data. The first set reports a role for C5a and its receptor on platelets as potential negative regulators of neovascularization using the well-established hind limb ischemia (HLI) model in mice. In these studies (basically figures 1-5) they convincingly show that deletion of C5aR1 globally and targeted to platelets leads to more rapid recovery of blood flow after induction of HLI, and that this was associated with increased capillary density, increased collateralization and increased pericyte/smooth muscle cell investment, detected using immunofluorescence microscopy and micro-CT imaging. They then explored mechanisms and found that co-incubation of platelets from C5aR1 null mice with cultured endothelial cells modestly enhanced EC migration in a scratch wound assay, tube formation in matrigel and outgrowth from EC spheres. Incorporation of platelets into matrigel plugs in vivo decreased vascular invasion and vascularization. These effects were shown to require platelet chemokine CXCL4 (platelet factor 4) and were consistent with in vitro data showing that C5a induced platelet secretion of CXCL4. Although many of the effects were modest in scale (20% improvement in blood flow in the HLI model; 40% changes in migration and tube formation) the data are convincing and novel and will be of interest to those in the field.

The manuscript, however, has several major weaknesses related to potential novel mechanisms. Most importantly, characterization of C5aR1-induced platelet activation is superficial and incomplete. More troubling is the lack of convincing evidence supporting the most novel conclusion of the manuscript; i.e. that C5aR1 "preferentially" releases CXCL4.

Major issues:

1. To show convincingly that C5a induces "preferential" release of CXCL4 it would be necessary to show careful dose responses comparing C5a to other agonists and comparing CXCL4 release to release of other well characterized pro- and anti-angiogenic factors. Figure 6a and supplemental Fig6 are key to this issue and are not at all convincing. Most importantly, is the 33% increase in CXCL4 shown in the dot blot statistically significant? A Bonferroni correction for multiple variables is indicated. How many replicates were performed? The blots appear in some areas to be overexposed, therefore it is not clear that the quantification was done in the linear range. Oddly, the most well characterized platelet anti-angiogenic factor, thrombospondin-1 was not included on the blot? The ELISA validation data in Figure 6B is suggestive, but not convincing. A time course and a dose response showing saturation are needed.

2. The data showing platelet activation by C5a are not very rigorous. The paper would be strengthened greatly by showing dose responses, time courses and functional data (e.g. aggregometry). Since C5aR1 is a GPCR and much is known about GPCR signaling in platelets, it would be helpful to discuss what is known about specific G proteins coupled to C5aR1 in other cells and how that might relate to platelet signaling by C5a.

3. The in vivo matrigel experiments are very problematic. Fixed platelets would probably be a better control than buffer for the platelet incorporation studies. It was not clear how many platelets were added to the matrigel and it does not seem likely that CD41 antigen detected 7 days after implantation would be of donor origin given the short "life span" of platelets and the likelihood that platelets would be activated in the context of being embedded in matrigel. Also, what do the authors mean by "retransfusion"; the methods are not clear as to how many platelets were

transfused or whether native platelets were depleted prior to transfusion.

4. The colocalization studies of CD42b and C5AR1 shown in Figure 1d are confusing. These were done at 14d post injury, so the platelets are more likely representative of circulating platelets in the neovasculature, and not representative of platelet involved in the initial injury response. Also, there is some evidence that CD42 can be expressed by EC in some circumstances. This could impact the mRNA data in Figv1C.

Minor issues:

1. How were the microCT data quantified?
2. Page 4, line 82. It is probably not correct to use the word "recently" in describing papers from 2006-2010.
3. Page 6, line 116: What is the evidence that the complement deposition is "excessive".

Reviewers' comments:

Reviewer #1 (Remarks to the Author):

The manuscript by Nording et al is an interesting and reasonably solid paper that is attempting to address the link of the complement system to angiogenesis. There were a number of items that I hope the authors will find helpful.

We thank this reviewer for his/ her interest in our work and for the valuable suggestions for improvement.

1) A significant finding of the paper is the demonstration of the increase in the angiogenic response in the platelet-specific deletion of C5aR1 mice. The mouse strain used were C57/Bl6 which are well known to demonstrate a robust angiogenic response to hind-limb ischemia and thus anything that shows an improvement over excellent/robust is of somewhat limited value. Performing the studies in strains that have a known impairment in their angiogenic response would be preferable. Even a study of a less specific pharmacologic inhibitor would be helpful for this report.

We see the reviewers point and followed his/ her suggestion using a pharmacological inhibitor (PMX53) to further scrutinize the effect of increased angiogenesis upon C5aR1 inhibition (see figure 7k, l, and 7p of the revised manuscript). In accordance with our previous findings, inhibition of C5aR1 resulted in significantly increased vessel formation. In addition, we carried out hindlimb ischemia experiments applying a pharmacological method of platelet depletion (figure 3e, f). Under platelet depletion, no significant difference in revascularization in between WT and C5ar1^{-/-} could be observed any more further corroborating the importance of platelet C5aR1 in ischemic tissue revascularization.

2) How different are the platelets from the C5aR1 knock out mice compared to the wild type? This is relevant because the authors used a targeted candidate approach to establish the link between the C5aCR1 and CXCL4 with the latter validated from an array of candidates that have known properties. It follows from that approach, that a factor found is expected to demonstrate the known antiangiogenic properties. I do not think there is a right – wrong answer here and of course the difference will depend on

the method used to assess for differences but if a large number of differences are found then the specificity of the finding would be less certain.

These are very interesting and relevant questions. In the revised version of our manuscript, we first tested adhesion and secretion potential of WT vs. C5ar1^{-/-} platelets. There was no significant difference between knockout and WT platelets regarding these platelet functions (figure 2m, n and 6j, k and supplemental figure 11 of the revised manuscript). In addition, granule content was not different (supplemental figure 18a-k of the revised manuscript). Furthermore, various adhesion receptors were not differentially expressed on WT vs. C5ar1^{-/-} platelets and the aggregation behaviour was similar between both groups (supplemental figure 15a-m and supplemental figure 12a, b of the revised manuscript). Moreover, there was no significant difference regarding platelet activation receptor upregulation upon stimulation between WT and C5ar1^{-/-} platelets (supplemental figure 12a-l of the revised manuscript). As requested by reviewer 2, we furthermore tested the level of C3aR protein and found no difference in expression between WT and C5ar1^{-/-} platelets (supplemental figure 14 of the revised manuscript). Thus, all evaluated parameters showed no relevant difference between WT and C5ar1^{-/-} platelets.

Another approach would be to compliment the data in Fig 6a with a look at secreted angiogenic factors.

In the revised version of our manuscript, we further improved our experimental protocol for experiments of figure 6a, b (previously supplemental figure 6). We performed platelet isolation using activation inhibitors resulting in lower baseline activation and carried out more repeats of the experiment. We found that baseline CXCL4 secretion is low and appears to be more specific in comparison to other factors. Furthermore, the increase in secretion is specific in comparison to other important platelet released factors (for example SDF-1). Moreover, we carried out a Bonferroni correction for multiple variables, which showed a statistically significant difference of CXCL4 in comparison to many other factors (supplemental figure 8 of the revised manuscript). Thrombospondin-1, which is not included in this blot, as well as further important mediators such as VEGF and endostatin were separately analyzed by ELISA. As depicted in supplemental figure 16a-c of the revised manuscript, there is no relevant thrombospondin-1, VEGF or endostatin secretion upon C5a stimulation.

3) Do the platelets from the C5aCR1 KO mice act differently in regards to their deposition? This relates to a bigger question of how and where the platelets adhere and deposit within this model and how that relates to the early response difference based on the genetic deletion.

This is an important question, which we have addressed experimentally. On the one hand, there was no significant difference in platelet adhesion to endothelial cells *in vitro* between WT and C5ar1^{-/-} platelets (figure 2m, n of the revised manuscript). Importantly, there was no difference in platelet deposition *in vivo* in the ischemic hindlimb tissue between platelet-specific C5ar1-deficient mice and controls (figure 4e, f of the revised manuscript).

4) Responses based on therapy or genetic deletion in mice following HLI that are present at 3 or 4 days post HLI, which this study shows, are often considered as from differences in arteriogenesis in pre-existing collaterals, as opposed to angiogenesis.

The reviewer addresses an important aspect because pre-existing collaterals have been shown to be a vital prerequisite for efficient revascularization of ischemic limbs. In this regard, we found no impact of C5aR1 on embryonal vessel formation (supplemental figure 4). To further scrutinize our findings, we carried out additional microCT assessments. Here, we found no differences in the vessel diameters of non-ischemic hindlimbs from platelet-specific C5aR1-deficient mice and controls (supplemental figure 7c of the revised manuscript). To better address this important point, we consider it in detail in the discussion part of our manuscript.

Reviewer #2 (Remarks to the Author):

The authors study the role of platelets on tissue revascularization in a murine hindlimb ischemia model and provide evidence that complement C5a receptor 1 (C5aR1) deficiency specifically on platelets augments limb neovascularization over 14 days. Mechanistic analysis associates the absence of platelet C5aR1 to increased platelet release of CXCL4 which in turn inhibits endothelial cell function by binding to EC expressed CXCR3. They provide some evidence the effect is relevant in human endothelial cells and platelets using C5aR1 blockade in vitro.

The finding that platelets express C5aR1 and that its absence from platelets alters neovascularization are novel. While statistically significant, the biological significance of the findings is unclear as the effects are modest (approximately 20-30%). In fact, the absence of endothelial CXCR3 (Figure 6) seems to have a much more robust effect than the absence of platelet C5aR1.

We thank the reviewer for his/her close reading of our manuscript, for the interest in our novel findings and for the valuable suggestions to further improve our manuscript. We agree with the reviewer that absence of CXCR3 is followed by a more robust effect, which may be explained by the fact that other antiangiogenic ligands besides CXCL4 can bind to CXCR3 (e.g. reviewed in Van Raemdonck, 2015, Cytokine GrowthFR). We are convinced, however, that CXCL4 represents a relevant factor in this system, because it is abundant in ischemic hindlimb tissue.

Major critical comments

1.) Significance: It is unclear in what clinical/pathophysiologic circumstance this mechanism would be relevant. The authors mention diseases including aortic aneurysm, heart attack, or anti-phospholipid disease but these processes tend to be sudden and catastrophic requiring immediate revascularization to preserve tissue function. Given that the authors showed no change in embryologic vascularization due to C5aR1 deficiency and only see an effect in an acute permanent ischemia model over 14 days, it seems unlikely that C5a/C5aR1 blockade would provide a benefit before the acute event occurred. Further, their biochemical readout for increased neovascularization is in augmented capillaries, is not quantifiable using their analytic

approaches, and did not seem to impact the number of larger arterioles, which is what would be needed in this model of large vessel ischemic disease.

We demonstrate the clinical/ pathophysiological relevance in our hindlimb ischemia experiments, a model that is generally accepted as an *in vivo* model for occlusive peripheral artery disease (Limbourg et al. 2009). The second point of the reviewer refers to the discussion on page 23, line 17. There, we just wanted to elude to the role of complement-targeted therapeutics in general in the mentioned diseases, not the C5aR1 or revascularization in specific. Further studies, however, could be interesting to address the role of the C5aR1 in these clinical settings. To clarify this point, we adjusted the discussion accordingly. Experiments using the *in vivo* model of embryonic angiogenesis were carried out to exclude that a difference in ischemic angiogenesis (for instance using the hindlimb ischemia model) is caused by a pre-existing difference between WT and C5ar1^{-/-} mice rather than a difference caused by ischemia induced formation of new vessels. To address the reviewer's concerns regarding an impact on larger arterioles, we carried out additional microCT assessments. Here, we found that in platelet-specific C5ar1-deficient mice, larger arterioles are more abundant after induction of hindlimb ischemia resulting in improved revascularization compared to controls (figure 5c in the revised manuscript). Furthermore, we carried out *in vivo* Matrigel experiments using a C5aR1 antagonist (PMX53). In accordance with our previous findings, inhibition of C5aR1 resulted in significantly increased vessel formation (figure 7k, l of the revised manuscript).

2.) Potential role of C5aR2 needs to be addressed. Evidence from multiple groups including co-author J Kohl indicate that C5aR2 binds C5a and can, in some cell types, transmit an activating or inhibitory signal (latter potential involving b-arrestin among other mechanisms). If C5aR2 is present on the platelets (the authors need to test this), absence of C5aR1 could increase C5aR2 signaling, which in turn could account for their findings (more signaling via C5aR2 inhibits CXCR4 release as opposed to less C5aR1 signaling inhibiting CXCR4 release). This needs to be addressed experimentally.

We followed the reviewers request and analyzed expression of C5aR2 on platelets. As depicted in supplemental figure 20a-e of the revised manuscript, C5aR2 is not expressed on platelets using tdTomato-C5aR2 knockin mice and C5aR2-specific ligation using the C5aR2 agonist P32 did not induce platelet secretion.

3.) Effect of C5aR1 deficiency on platelet aggregation in the ischemic limb: The authors show that complement activates in the ischemic limb and that platelets aggregate as well, but they do not show whether there are differences in platelet aggregation between WT and C5aR1-deficient conditions. If there are greater numbers of WT platelets forming microthrombi, the finding of neo-vascularization may not be due to any intrinsic signaling of the C5aR1 on platelets but rather via a mechanism of anaphylatoxin-induced platelet trafficking.

This is an important question, which we have addressed experimentally. On the one hand, there was no significant difference in platelet adhesion to endothelial cells *in vitro* between WT and C5ar1^{-/-} platelets (figure 2m, n of the revised manuscript). Importantly, there was no difference in platelet deposition *in vivo* in the ischemic hindlimb tissue between platelet-specific C5aR1-deficient mice and controls (figure 4e, f of the revised manuscript). Furthermore, various adhesion receptors were not differentially expressed on WT vs. C5ar1^{-/-} platelets and the aggregation behaviour was similar between both groups (supplemental figure 15a-m and supplemental figure 12a, b of the revised manuscript). Moreover, there was no significant difference regarding platelet activation receptor upregulation upon stimulation between WT and C5ar1^{-/-} platelets (supplemental figure 12c-l).

4.) Incomplete limb revascularization time course: The authors provide a time course up to 14d post-ischemia looking at revascularization in figures 3 and 4 but the curve of the lines suggests that the WT group will eventually meet the C5aR1 deficient group within a few days/week. If the blood flow curves stayed separate in the long term, this would be a much more interesting and robust finding. If not, this would further weaken the significance of the phenotype. The authors need to extend the finding to beyond 14 days to determine this.

The experiment was designed to judge the impact of C5aR1 in ischemia-induced revascularization. In our experimental setting, blood flow ratio was already back to 100 % after 14 days in the knockout group, which we consider a reasonable timepoint to stop the experiment. In line with this view, the local authorities of the Animal and Care and Use Committee approved the experimental setting (termination at day 14) with the blood flow ratio reaching 100%. It will be difficult to convince the authorities to conduct experiments beyond the 14 d time point with the argument that the two curves may meet when the experiment is

extended. We agree that we cannot predict the course of the experiment at later time points. However, we already show significant differences between the two groups at different time points between d4 and d14.

5) Role of C5. The authors need to test C5 deficient mice and controls (available from Jax as Hc1 and Hc0). Understanding whether C5 deficiency leads to similar phenotype as C5aR1 deficiency will add insight regardless of outcome

To address this important point, we analyzed the expression of C5aR1 on C5^{-/-} platelets, the secretion of CXCL4 upon C5a-stimulation and activation marker upregulation in response to stimulation in WT and C5^{-/-} platelets. There was a slight but not statistically significant increase in C5aR1 expression on platelets (supplemental figure 19a of the revised manuscript). CXCL4 secretion from C5^{-/-} platelets occurred only at higher C5a concentrations compared to WT (supplemental figure 19b of the revised manuscript) and activation of C5^{-/-} platelets was similar to WT platelets, even though C5^{-/-} platelets displayed a significantly higher baseline activation level (supplemental figure 19c-d of the revised manuscript).

6) Incomplete data linking CXCL4 and C5aR1 (in addition to the C5aR2 issue above):

-a. In figure 6 the authors show C5a induces platelet-derived CXCL4 and that this effect requires platelet expressed C5aR1. They need to provide biochemical signaling mechanisms through which absence of C5aR1 induces the platelet to increase the release of CXCL4 (related to the comment regarding C5aR2 above).

We thank the reviewer for this helpful comment. In the revised manuscript, we analyzed several potential pathways, which could connect binding of C5a via C5aR1 with CXCL4 secretion. For instance, we tested phosphorylation of AKT, PI3K, GSK-3 β , p44/42 MAPK, PLC β 3, PLC γ 2, PKA and RAP1. Interestingly, we found that the signal seems to be conducted via the G $\beta\gamma$ subunit of C5aR1 and not G α i as we found C5a-dependent phosphorylation of PI3K and Akt but not PKA (new figure 6l-n and supplemental figure 21 of the revised manuscript). PKC activation seems to be central for C5a-induced CXCL4 secretion as we found consistent C5a-dependent PKC phosphorylation and could also show that PKC activation induces CXCL4 secretion (new figure 6l, o of the revised manuscript). We discussed these results accordingly in the discussion part of the revised manuscript.

b. It is noteworthy that supplemental fig 6 shows robust baseline CXCL4 secretion suggesting other mediators. The fold increase upon adding C5a is about 35% over this high spontaneous production, consistent with a modulating but not biologically robust effect of C5a/C5aR1.

In the revised version of our manuscript, we further improved our experimental protocol for experiments of figure 6a, b of the revised manuscript (previously supplemental figure 6). We performed platelet isolation using activation inhibitors resulting in lower baseline activation and carried out more repeats of the experiment. We found that baseline CXCL4 secretion assessed in this way is low and appears to be more specific in comparison to other factors. Furthermore, we used a platelet-specific C5ar1-deficient mouse model in the hindlimb ischemia assay and observed a significant difference compared to controls for vessel formation.

-c. In fig 6, the authors show that a) WT (but not C5aR1 KO) platelets stimulated with C5a reduce endothelial tube formation and b) immunodepletion of CXCL4 from the supernatant can augment tube formation. The need to “close the loop” by testing whether CXCL4 immunodepleted C5aR1KO-platelet supernatants behave similarly or that by adding back CXCL4 to C5aR1KO-platelet supernatants they can replicate the findings of the WT conditions.

We followed the reviewer's constructive suggestions and carried out additional experiments. Adding back CXCL4 to C5a-conditioned C5ar1^{-/-} platelets supernatant restored the effect of WT C5a-conditioned platelets supernatant (figure 7a of the revised manuscript). We found no significant difference between C5a-conditioned C5ar1^{-/-} supernatant as well as C5a-conditioned C5ar1Cxcl4-double-knockout platelets (figure 7a of the revised manuscript). Removal of C5a by immunodepletion significantly decreased tube formation in WT but not in C5ar1^{-/-} platelet supernatant (figure 7b of the revised manuscript).

-d. The data using CXCR3 KO endothelial cells (the putative receptor responding to CXCL4) is similarly incomplete. Figures 6h and i are firstly a bit confusing and misleading because they shift the axis to show it as % inhibition when all the previous figures showed % formation. While there is less inhibition in CXCR3 KO endothelial cells vs WT the authors need to repeat the experiment using C5aR1 KO-platelets to

provide test whether the C5aR1-CXCL4 axis is the primary driver of the finding they are describing.

We followed the reviewer's suggestion and showed the data as % formation (figure 7e of the revised manuscript). Furthermore, we carried out additional experiments using Cxcr3-knockout endothelial cells. In accordance with our hypothesis, there was no difference between control or C5a stimulated platelet supernatant using both WT and C5ar1^{-/-} platelets (figure 7e of the revised manuscript).

-e. the authors show that with increasing amounts of C5a they can induce cultured platelets to release CXCL4 (Fig 6b) but in figure 6e the authors conclude that CXCL4 release is C5a-C5aR1 specific by showing that C5a stimulation induces a lower CXCL4:P-selectin ratio by IF compared to ADP stimulated platelets. However, they disregard the possibility that C5a more potently augments P-selectin expression than ADP (which is actually suggested by their data in figure 6d). This need further experimental evidence to make the appropriate conclusions.

We agree with the reviewer that figure 6f of the revised manuscript (previously figure 6d) indicates that both C5a and ADP show rather similar increases in P-selectin induction (there was no statistical difference between both groups). We followed the reviewer's advice and carried out additional experiments depicted in figure 6j, supplemental figure 11a as well as supplemental figure 12c of the revised manuscript. Indeed, ADP also induces CXCL4 secretion from platelets. The level of CXCL4 secretion induced by 200 ng/ml C5a is comparable to that of 25 μ M ADP (supplemental figure 11a of the revised manuscript). C5a induced a low degree of platelet activation (figure 6h of the revised manuscript). However, 20 μ M ADP induce a much stronger platelet activation than 200 ng/ml C5a (supplemental figure 12c of the revised manuscript). For the convenience of the reviewer, we summarized these 3 subfigures into figure 1 for Reviewer 2.

Minor issues related to complement

1. What is the activating pathway of complement in this system which differs from ischemia reperfusion?

Complement activation in ischemia-reperfusion injury (IRI) is well-characterized (as reviewed by (Noris and Remuzzi 2013)). However, to the best of our knowledge, we are the first to show complement deposition in hindlimb ischemia without reperfusion. As reactive oxygen species (ROS) production is regarded as the primary complement activation mechanism activating primarily the classical pathway, this is likely to also apply to the hindlimb ischemia model as also without reperfusion, ROS production is known to play an important role (Carlomosti et al. 2017). Furthermore, the activated endothelium in the ischemic tissue may

bind gC1qR and further activate the classical complement pathway (Ghebrehiwet et al. 2006; Yin et al. 2007).

2. Interactive effects of C5aR1 and C3aR1. The authors note in the introduction that C3aR1 on platelets has been identified as an important mediator of platelet aggregation. Are there compensatory differences in C3aR1 levels in these C5aR1-deficient platelets that may be contributing to their findings?

To address this point, we tested the level of C3aR protein and found no difference in expression between WT and C5ar1^{-/-} platelets (supplemental figure 19 of the revised manuscript). Thus, all evaluated parameters showed no relevant difference between WT and knockout platelets.

3. Do platelets express complement regulators and if so which ones?

Platelets express a variety of complement regulating proteins reviewed in Seminas in Immunology (Nording and Langer 2018). Absence or impairment of these regulatory proteins is associated with platelet dysfunction, alterations in platelet activation or thrombocytopenia. Casein kinase released from platelets was found to regulate C3 activation by phosphorylation. CD55 and CD59 are important complement regulatory proteins. Lack of these proteins has been linked to enhanced generation of prothrombotic microparticles and reduced regulation of tissue factor by the tissue factor pathway inhibitor. Platelet alpha-granules contain regulators of complement activation, such as C1 inhibitor. Polyphosphates and the C1 inhibitor serpin colocalize on the surface of platelets and are likely to interact in complement modulation, as polyphosphates have been shown to regulate the activity of C1 inhibitors. Furthermore, Factor H is a protein, which is particularly abundant on platelets and can be released by platelets.

We supplemented the discussion part of our manuscript and state that complement modulating and inhibiting proteins are associated with platelets.

References

Carlomosti, Fabrizio, Marco D'Agostino, Sara Beji, Alessio Torcinaro, Roberto Rizzi, Germana Zaccagnini, Biagina Maimone, et al. 2017. "Oxidative Stress-Induced MiR-200c Disrupts the Regulatory Loop Among

SIRT1, FOXO1, and ENOS.” *Antioxidants & Redox Signaling* 27 (6): 328–44.
<https://doi.org/10.1089/ars.2016.6643>.

Ghebrehwet, Berhane, Claudia CebadaMora, Lee Tantral, Jolyon Jesty, and Ellinor I B Peerschke. 2006. “GC1qR/P33 Serves as a Molecular Bridge between the Complement and Contact Activation Systems and Is an Important Catalyst in Inflammation.” In *Advances in Experimental Medicine and Biology*, 586:95–105.
<https://doi.org/10.1007/0-387-34134-X-7>.

Limbourg, Anne, Thomas Korff, L. Christian Napp, Wolfgang Schaper, Helmut Drexler, and Florian P. Limbourg. 2009. “Evaluation of Postnatal Arteriogenesis and Angiogenesis in a Mouse Model of Hind-Limb Ischemia.” *Nature Protocols* 4 (12): 1737–48. <https://doi.org/10.1038/nprot.2009.185>.

Nording, Henry, and Harald F Langer. 2018. “Complement Links Platelets to Innate Immunity.” *Seminars in Immunology* 37 (December 2017): 43–52. <https://doi.org/10.1016/j.smim.2018.01.003>.

Noris, Marina, and Giuseppe Remuzzi. 2013. “Overview of Complement Activation and Regulation.” In *Seminars in Nephrology*, 33:479–92. Elsevier.

Yin, Wei, Berhane Ghebrehwet, Babette Weksler, and Ellinor I. Peerschke. 2007. “Classical Pathway Complement Activation on Human Endothelial Cells.” *Molecular Immunology* 44 (9): 2228–34.
<https://doi.org/10.1016/j.molimm.2006.11.012>.

Reviewer #3 (Remarks to the Author):

The role of platelets in regulating angiogenesis is an important topic given the large array of pro- and anti-angiogenic molecules stored in alpha-granules and released during platelet activation. The concept of differential release of pro vs anti-angiogenic factors based on specific agonists and/or specific granule compartments has been under study for quite some time by several investigators in the field. In this manuscript the authors report 2 complementary sets of data. The first set reports a role for C5a and its receptor on platelets as potential negative regulators of neovascularization using the well-established hindlimb ischemia (HLI) model in mice. In these studies (basically figures 1-5) they convincingly show that deletion of C5aR1 globally and targeted to platelets leads to more rapid recovery of blood flow after induction of HLI, and that this was associated with increased capillary density, increased collateralization and increased pericyte/smooth muscle cell investment, detected using immunofluorescence microscopy and micro-CT imaging. They then explored mechanisms and found that co-incubation of platelets from C5aR1 null mice with cultured endothelial cells modestly enhanced EC migration in a scratch wound assay, tube formation in matrigel and outgrowth from EC spheres. Incorporation of platelets into matrigel plugs in vivo decreased vascular invasion and vascularization. These effects were shown to require platelet chemokine CXCL4 (platelet factor 4) and were consistent with in vitro data showing that C5a induced platelet secretion of CXCL4. Although many of the effects were modest in scale (20% improvement in blood flow in the HLI model; 40% changes in migration and tube formation) the data are convincing and novel and will be of interest to those in the field.

The manuscript, however, has several major weaknesses related to potential novel mechanisms. Most importantly, characterization of C5aR1-induced platelet activation is superficial and incomplete. More troubling is the lack of convincing evidence supporting the most novel conclusion of the manuscript; i.e. that C5aR1 "preferentially" releases CXCL4.

We thank the reviewer for his/ her close reading of our manuscript, for the interest in the novelty of our work as well as the uncovered mechanisms and for the valuable suggestions for improvement. The constructive suggestions from the reviewer led us to perform additional

experiments and substantively revise our manuscript. We believe, these new data both affirm and extend the significance of our conclusions.

Major issues

1. To show convincingly that C5a induces "preferential" release of CXCL4 it would be necessary to show careful dose responses comparing C5a to other agonists and comparing CXCL4 release to release of other well characterized pro- and anti-angiogenic factors. Figure 6a and supplemental Fig6 are key to this issue and are not at all convincing. Most importantly, is the 33% increase in CXCL4 shown in the dot blot statistically significant? A Bonferonni corection for multiple variables is indicated. How many replicates were performed? The blots appear in some areas to be overexposed, therefore it is not clear that the quantification was done in the linear range. Oddly, the most well characterized platelet anti-angiogenic factor, thrombospondin-1 was not included on the blot? The ELISA validation data in Figure 6B is suggestive, but not convincing. A time course and a dose response showing saturation are needed.

In the revised version of our manuscript, we further improved our experimental protocol for experiments of figure 6a, b of the revised manuscript (previously supplemental figure 6). We performed platelet isolation using activation inhibitors resulting in lower baseline activation and carried out more repeats of the experiment. We found that baseline CXCL4 secretion is low and appears to be more specific in comparison to other factors. Furthermore, the increase in secretion is specific in comparison to other important platelet released factors (for example SDF-1). Moreover, we carried out a Bonferroni correction for multiple variables, which showed a statistically significant difference of CXCL4 in comparison to many other factors (supplemental figure 8 of the revised manuscript). Thrombospondin-1, which is not included in this blot, as well as further important mediators such as VEGF and endostatin were separately analyzed by ELISA. As depicted in supplemental figure 16a-c of the revised manuscript, there is no relevant thrombospondin-1, VEGF or endostatin secretion upon C5a stimulation.

Moreover, we carried out additional experiments as requested by the reviewer analyzing the dose response of CXCL4 to C5a showing saturation (new figure 6c of the revised manuscript). We also show that C5a induces a short-term calcium signal in platelets in figure 7g of the revised manuscript.

Furthermore, we carried out time course and dose response experiments using different temperatures to validate data of figure 6B of the initial manuscript. As depicted in supplemental figure 9 and 10, we detected a dose and time dependent release of CXCL4 upon C5a stimulation and found that optimal CXCL4 secretion is obtained with 10 minutes of stimulation at 37°C with a concentration of 200 ng/ml C5a.

2. The data showing platelet activation by C5a are not very rigorous. The paper would be strengthened greatly by showing dose responses, time courses and functional data (e.g. aggregometry). Since C5aR1 is a GPCR and much is known about GPCR signaling in platelets, it would be helpful to discuss what is known about specific G proteins coupled to C5AR1 in other cells and how that might relate to platelet signaling by C5a.

We thank the reviewer for this helpful question. In the revised manuscript, we analyzed several potential pathways, which could connect binding of C5a via C5aR1 with CXCL4 secretion. For instance, we tested phosphorylation of AKT, PI3K, GSK-3 β , p44/42 MAPK, PLC β 3, PLC γ 2, PKA and RAP1. Interestingly, we found that the C5a signal seems to be conducted via the G $\beta\gamma$ subunit of C5aR1 and not G α_i as we found C5a-dependent phosphorylation of PI3K and Akt but not PKA (new figure 6l-n of the revised manuscript). PKC activation seems to be central for C5a-induced CXCL4 secretion as we found consistent C5a-dependent PKC phosphorylation and could also show that PKC activation induces CXCL4 secretion (new figure 6l, o of the revised manuscript). We discussed these results accordingly in the discussion part of the revised manuscript.

As requested, we carried out time course and dose response experiments (new supplemental figure 9 and 10 and figure 6c of the revised manuscript). To analyze functional aspects of platelets upon C5a stimulation, we measured ATP release and *in vitro* aggregation (new supplemental figure 17 and 13 of the revised manuscript). There was no increased ATP release upon C5a stimulation and additional CRP or thrombin stimulation (supplemental figure 17). Similarly, aggregation was not significantly increased by C5a (supplemental figure 13).

3. The *in vivo* matrigel experiments are very problematic. Fixed platelets would probably be a better control than buffer for the platelet incorporation studies. It was

not clear how many platelet were added to the matrigel and it does not seem likely that CD41 antigen detected 7 days after implantation would be of donor origin given the short "life span" of platelets and the likelihood that platelets would be activated in the context of being embedded in matrigel. Also, what do the authors mean by "retransfusion"; the methods are not clear as to how many platelet were transfused or whether native platelets were depleted prior to transfusion.

We followed the reviewer's advice and removed all figures with experiments, for which platelets were added to the Matrigel and compared to vehicle control. Furthermore, the wording was corrected and the expression "retransfusion" was replaced by „supplementation“

4. The colocalization studies of CD42b and C5AR1 shown in Figure 1d are confusing. These were done at 14d post injury, so the platelets are more likely representative of circulating platelets in the neovasculature, and not representative of platelet involved in the initial injury response. Also, there is some evidence that CD42 can be expressed by EC in some circumstances. This could impact the mRNA data in Fig1C.

The main point of this figure was to show that C5aR1 is expressed and present on platelets in the ischemic hindlimb *in vivo*. To avoid detection of platelets in the circulation, mice were perfused extensively with PBS to get rid of platelets present within the blood vessel. To further address the reviewer's concern, we analyzed mice at day 4 post induction of ischemia. At this additional time point, C5aR1 expression could be detected on platelets as well (revised figure 1f of the revised manuscript). In order to address the point raised concerning the qPCR, we have analyzed a further platelet mRNA marker CD41 (revised figure 1d of the revised manuscript).

Minor issues:1. How were the microCT data quantified?

The new supplemental figure 7 of the revised manuscript describes the method for quantification of microCT data.

2. Page 4, line 82. It is probably not correct to use the word "recently" in describing papers from 2006-2010.

We corrected accordingly.

3. Page 6, line 116: What is the evidence that the complement deposition is "excessive".

We changed the wording.

REVIEWER COMMENTS

Reviewer #1 (Remarks to the Author):

I appreciate many of the responses and I do appreciate that the authors have made significant revisions toward their overall goal of linking complement to angiogenesis. I am not certain the authors completely addressed my first concern; though clearly they made efforts. The genetic background of the mice limits the ability to perform a -- "we made the response better" approach. The reference in Fig 7 k, l, p are more ex-vivo than in-vivo (though I agree there is a gray zone); they for certain are not hind-limb ischemia. Please provide additional detail on the platelet deleting approach. Could the authors not have used this approach in mice with known "impaired" perfusion recovery -- either in Balb/c mice or the same background strain but with diabetes or hyperlipidemia?

Reviewer #2 (Remarks to the Author):

The authors have responded well and addressed the majority of the questions/comments raised. There remain a few important issues that require additional clarification.

1. Several conclusions made by Nording et al have already been published by members of the same group and others.

- a. platelet activation and degranulation by C5a complement
- b. PF4 is released upon platelet degranulation
- c. PF4 is an anti-angiogenesis factor

<https://rupress.org.eresources.mssm.edu/jem/article-lookup/doi/10.1084/jem.158.2.603>
<https://www.jimmunol.org.eresources.mssm.edu/content/126/4/1506.long>
<https://www.thieme-connect.com/products/ejournals/abstract/10.1055/s-0038-1676349>
https://pubmed.ncbi.nlm.nih.gov/29802205/?from_term=langer+hF&from_pos=3
<https://pubmed.ncbi.nlm.nih.gov/15282661/>
<https://pubmed.ncbi.nlm.nih.gov/7657827/>
[https://ajp.amjpathol.org/article/S0002-9440\(10\)61133-9/fulltext](https://ajp.amjpathol.org/article/S0002-9440(10)61133-9/fulltext)
<https://link.springer.com/article/10.1007/BF00509775>
<https://www.nature.com/articles/srep42714#Bib1>
<https://pubmed.ncbi.nlm.nih.gov/15282661/>
<https://pubmed.ncbi.nlm.nih.gov/12038989/>

Thus, the novelty of this work mostly lies in how the authors nicely linked the phenomena together. The text should reflect this

2. Preferential release of PF4 by C5a (response to comment #6e)

As platelets do not have nuclei and both CD62 and PF4 are found in alpha granules, the authors need to clarify how platelet activation and degranulation (without altering degranulation markers, Fig 6E, I) could preferentially result in increased PF4 but not CD62 release. There are multiple possibilities potentially including but not limited to a) subsets of alpha granules that preferentially contain PF4 over CD62 and are triggered to release in response to C5a, b) C5aR alters megakaryocyte production of PF4 preferentially such that platelets contain more PF4 from development. Aggregation does not provide insight into this.

Importantly, the issue is NOT whether C5a triggers specifically PF4 as opposed to serotonin or beta-hexosaminidase (Fig 6j) and NOT whether C5a as opposed to other platelet stimulators (ADP, thrombin) triggers PF4. Any agent that triggers degranulation will lead to PF4 release (Fig S11a).

3. C5aR does not effect platelet function (response to comment #3)

The authors should address the possibility that C5aR affects platelet homeostasis that in turn impacts neovascularization. PF4 release is a consequence of degranulation and degranulation leads to hemostasis. If both PF4 and hemostasis negatively affects neovascularization, it may be difficult to set them apart. The authors need to address this issue ideally experimentally. Note that in a previous publication by the same group (Circulation 2018), the absence of C3aR prolonged bleeding times in mice. Are defects in hemostasis accounted for in the neovascularization phenotype of C5aR^{-/-} platelets? Does the absence of C5aR also affect bleeding times? Is coagulation/hemostasis altered by the absence of platelet C5aR to account for the phenotype? Is the ischemic injury protection in mice deficient in platelet C5aR mitigated by heparin? While all of these experiments need not be performed the issue needs to be addressed.

The authors then need to attempt to discuss/clarify the dual role of PF4 in the model. PF4 is a pro-coagulant (binds and neutralizes local heparins to permit primary hemostasis) and angiogenesis inhibitor. The role of PF4 is complex and needs to be contextualized:

<https://doi.org/10.1182/blood-2003-11-3994>
<https://pubmed.ncbi.nlm.nih.gov/12609838/>
<https://pubmed.ncbi.nlm.nih.gov/9395524/>
<https://www.jbc.org/content/282/1/710.full>

4. Fig 12, platelet aggregation assays were done in the absence of C5a. ADP, TRAP, CRP responses were thus not affected in C5aR^{-/-} platelets. Does the addition of C5a to threshold concentration of ADP increases platelet aggregation? The authors need to perform platelet aggregation studies with threshold ADP, CRP (not triggering aggregation) in the presence of C5a

Reviewer #3 (Remarks to the Author):

In my initial review I was concerned mainly about two issues:

1. The biological effect in the HLI model was modest, and of unclear translational significance
2. The characterization of C5aR1-induced platelet activation was superficial and incomplete with lack of convincing evidence supporting the most novel conclusion in the MS; i.e. that C5aR1 "preferentially" releases CXCL4.

The authors have added much new data (Figure 6) and several supplementary figures, but the data are still not at all consistent or convincing. The level of variance in the platelet release assays is unacceptably high, making conclusions difficult and in many cases the data are not consistent with the known biology of platelet; for example, seeing granule release in the absence of an aggregation response. Perhaps these issues could be resolved with a small number of straightforward studies using human rather than mouse platelets. Mouse platelets can be difficult to study *ex vivo* as methods of blood collection can greatly influence function.

Response to reviewers

Reviewer #1 (Remarks to the Author):

I appreciate many of the responses and I do appreciate that the authors have made significant revisions toward their overall goal of linking complement to angiogenesis. I am not certain the authors completely addressed my first concern; though clearly they made efforts. The genetic background of the mice limits the ability to perform a -- "we made the response better" approach. The reference in Fig 7 k, l, p are more ex-vivo than in-vivo (though I agree there is a gray zone); they for certain are not hind-limb ischemia. Please provide additional detail on the platelet depleting approach. Could the authors not have used this approach in mice with known "impaired" perfusion recovery -- either in Balb/c mice or the same background strain but with diabetes or hyperlipidemia?

We thank this reviewer for the positive feedback and that s/he appreciates our significant revisions. Unfortunately, we did not have the possibility to carry out the experiments in Balb/c mice or C57BL/6 mice suffering from diabetes or hyperlipidemia. However, to address the reviewer's point, we performed the experiments initially suggested by the reviewer. More precisely, we refer to the comment using a pharmacologic inhibitor to target C5aR1. In the revised manuscript, we show that the C5aR1 inhibitor PMX-205 markedly increased neovascularization in the hind limb ischemia model (new Figure 7q).

Furthermore, we provide additional experimental details on platelet depletion. We demonstrate successful platelet depletion in the circulation of mice (new supplemental figure 5). Also, we outline this part in detail in the materials and methods and the discussion section.

Reviewer #2 (Remarks to the Author):

The authors have responded well and addressed the majority of the questions/comments raised. There remain a few important issues that require additional clarification.

We thank the reviewer that s/he appreciates our improvements and the additional experiments that we have performed in response to the comments. Below, you will find our response to the remaining and additionally raised concerns.

1. Several conclusions made by Nording et.al have already been published by members of the same group and others.

a. platelet activation and degranulation by C5a complement

b. PF4 is released upon platelet degranulation

c. PF4 is an anti-angiogenesis factor

Our work is of major scientific relevance, as it uncovers several novel aspects of platelet biology. We identify a new mechanism, by which the anaphylatoxin C5a tailors the release of an anti-angiogenic factor from distinct platelet granules, which is in contrast to known release mechanisms induced for example by ADP. Furthermore, we demonstrate for the first time the significant pathophysiological relevance of this observation using different in vivo models of ischemia and growth factor-induced angiogenesis.

We agree that the publications kindly provided by the reviewer contribute to the overall topic, however they clearly cover different aspects. Below, we briefly summarize such aspects to distinguish the results obtained in these studies from the findings of our manuscript.

<https://rupress-org.eresources.mssm.edu/jem/article-lookup/doi/10.1084/jem.158.2.603>

Polley MJ, Nachman RL. Human platelet activation by C3a and C3a des-arg. J. Exp. Med. 1983;158(2):603–15.

This rather early work is on C3a but not C5a. Furthermore, it does not really provide any mechanistic insight on how anaphylatoxins regulate platelet functions relevant for angiogenesis. It shows that complement can be activated by thrombin and that complement activation occurs on the platelet surface. Our experiments are, however, all in the absence of thrombin and cover different settings such as in vivo revascularization.

<https://www.jimmunol.org.eresources.mssm.edu/content/126/4/1506.long>

Meuer S, Ecker U, Hadding U, Bitter-Suermann D. Platelet-serotonin release by C3a and C5a: two independent pathways of activation. *J. Immunol.* 1981;126(4):1506–1509.

This manuscript focuses on the secretion of serotonin, which is a dense granule component of platelets. The work hints at two independent receptors for C5a and C3a. However, all this early work is performed in guinea pig platelets. In our work, we demonstrate a role of C5a for the release of a factor from specific subgranules from platelets. Most platelet angiogenic factors are stored in alpha- granules. Therefore, the work by Meuer et al. is of limited relevance for understanding the role of platelets in neovascularization.

<https://www.thieme-connect.com/products/ejournals/abstract/10.1055/s-0038-1676349>

Sauter R, Sauter M, Obrich M, et al. Anaphylatoxin Receptor C3aR Contributes to Platelet Function, Thrombus Formation and In Vivo Haemostasis. *Thromb. Haemost.* 2019;119(01):179–182.

https://pubmed.ncbi.nlm.nih.gov/29802205/?from_term=langier+hF&from_pos=3

Sauter RJ, Sauter M, Reis ES, et al. Functional Relevance of the Anaphylatoxin Receptor C3aR for Platelet Function and Arterial Thrombus Formation Marks an Intersection Point Between Innate Immunity and Thrombosis. *Circulation.* 2018;138(16):1720–1735.

These manuscripts from our own lab have their focus on the role of platelet C3aR for primary hemostasis, not vessel formation. Furthermore, we investigate C5aR1 instead of C3aR. The interesting finding is that C5aR1 does not impact on some platelet functions, for which C3a is important, such as aggregation, clot formation and bleeding cessation. Instead, it is relevant for angiogenesis. Anaphylatoxin receptors on platelets seem to fulfil specific biological roles.

<https://pubmed.ncbi.nlm.nih.gov/15282661/>

Bikfalvi A. Platelet Factor 4: An Inhibitor of Angiogenesis. *Semin. Thromb. Hemost.* 2004;30(03):379–385.

This review depicts that PF4/CXCL4 is a known antiangiogenic agent. Nonetheless, our work goes beyond this knowledge. The important novel finding is that we describe for the first time a mechanism, by which CXCL4 is released in spatiotemporal proximity to growing vessel from platelets, its main reservoir in the body, in vivo.

<https://pubmed.ncbi.nlm.nih.gov/7657827/>

Rinder CS, Rinder HM, Smith BR, et al. Blockade of C5a and C5b-9 generation inhibits leukocyte and platelet activation during extracorporeal circulation. *J. Clin. Invest.* 1995;96(3):1564–1572.

In this early study, the authors report that a C5-blocking antibody administered to whole blood induced a reduction of platelet activation after extracorporeal circulation. Furthermore, the authors describe an impact of C5a and C5b-9 on platelet-leukocyte aggregate formation. This is a descriptive study reporting a relationship of C5a and platelet-leukocyte aggregate formation, which is not at all the topic of our project. On the contrary, our study delineates a specific receptor-dependent mechanism, by which C5a induces specific responses in platelets (secretion of content relevant for angiogenesis with little aggregation response).

[https://ajp.amjpathol.org/article/S0002-9440\(10\)61133-9/fulltext](https://ajp.amjpathol.org/article/S0002-9440(10)61133-9/fulltext)

Laudes IJ, Chu JC, Sikranth S, et al. Anti-C5a Ameliorates Coagulation/Fibrinolytic Protein Changes in a Rat Model of Sepsis. *Am. J. Pathol.* 2002;160(5):1867–1875.

This investigation focuses on the effect of C5a-blockade on i.e. survival in a cecal ligation and puncture model in rats. The authors could show that C5a had an effect on hemostatic functions. However, this is primarily mediated by the plasmatic coagulation system and not platelets themselves. Platelet count was reduced, which is to be expected when the plasmatic coagulation system is activated. Furthermore, the most likely explanation for the finding presented in the paper is a leukocyte-mediated effect and not primarily a platelet-mediated effect. In our work, however, we present, for the first time, data from a platelet-specific transgenic mouse model presenting with a phenotype of improved revascularization (C5aR1^{fl/fl}Pf4cre⁺ mice). This rules out that the observed platelet-response is primarily triggered by leukocyte-platelet interactions rather than complement-platelet crosstalk.

<https://link.springer.com/article/10.1007/BF00509775>

Grossklaus C, Damerau B, Lemgo E, Vogt W. Induction of platelet aggregation by the complement-derived peptides C3a and C5a. *Naunyn. Schmiedebergs. Arch. Pharmacol.* 1976;295(1):71–76.

In this very early study, the effect of C3a and C5a, on platelet aggregation in guinea-pig platelets was investigated. Importantly, the authors demonstrate species-specific differences of platelets from i.e. cats in comparison with guinea-pig platelets towards C3a and C5a. Previously, we have already been able to replicate the finding that C3a induces aggregation in mouse platelets. In this work, we did not find an effect of C5a on mouse or human platelet aggregation. Therefore, the work by Grossklaus et al. does not pre-empt any of the findings presented here.

<https://www.nature.com/articles/srep42714#Bib1>

Mizuno T, Yoshioka K, Mizuno M, et al. Complement component 5 promotes lethal thrombosis. *Sci. Rep.* 2017;7(1):42714.

This work illustrates that C5-deficient mice are protected from lethal thrombosis induced by histones and have milder thrombocytopenia, consumptive coagulopathy, and liver injury with embolism and lower PLA production than C5-sufficient mice. This strengthens the idea of a crosstalk of the complement system as well as the plasmatic coagulation system. Primarily, thrombosis is a function of the plasmatic coagulation system. Accordingly, the paper does not demonstrate that the observed effect is dependent on a platelet-complement crosstalk, or that a specific complement-dependent platelet response is required.

Here, we show a specific receptor-dependent response of platelets to the anaphylatoxin C5a, which holds biological relevance for revascularization in vivo.

<https://pubmed.ncbi.nlm.nih.gov/12038989/>

Myler HA, West JL. Heparanase and Platelet Factor-4 Induce Smooth Muscle Cell Proliferation and Migration via bFGF Release from the ECM. *J. Biochem.* 2002;131(6):913–922.

This work demonstrates that CXCL4 may not only transmit antiangiogenic effects but also induces proangiogenic SMC proliferation. Nevertheless, this work focuses on the role of CXCL4 in vitro.

It is uncontested that platelets provide the main reservoir for CXCL4 in the body. However, so far, no mechanism has been described for the release of CXCL4 from platelets in the context of ischemia-induced revascularization.

Together, the work we present here describes a novel secretion mechanism of CXCL4 from platelets in vivo, which does not induce full-blown platelet activation, complete degranulation and – ultimately – platelet resolution, but relies on secretion of specific granule subsets with only moderate general platelet activation. Such a mechanism on the level of platelets in revascularization is entirely novel and has not been described, previously.

Thus, the novelty of this work mostly lies in how the authors nicely linked the phenomena together. The text should reflect this

Above, we have delineated that there are specific novel findings of this work not only linking together already known phenomena but providing new insights for the understanding of how angiogenesis is regulated by platelet-derived factors via complement C5a.

In the revised version of the manuscript, we included the references suggested by the reviewer and discussed them accordingly.

2. Preferential release of PF4 by C5a (response to comment #6e)

As platelets do not have nuclei and both CD62 and PF4 are found in alpha granules, the authors need to clarify how platelet activation and degranulation (without altering degranulation markers, Fig 6E, I) could preferentially result in increased PF4 but not CD62 release. There are multiple possibilities potentially including but not limited to a) subsets of alpha granules that preferentially contain PF4 over CD62 and are trigger to release in response to C5a, b) C5aR alters megakaryocyte production of PF4 preferentially such that platelets contain more PF4 from development. Aggregation does not provide insight into this.

Importantly, the issue is NOT whether C5a triggers specifically PF4 as opposed to serotonin or beta-hexosaminidase (Fig 6j) and NOT whether C5a as opposed to other platelet stimulators (ADP, thrombin) triggers PF4. Any agent that triggers degranulation will lead to PF4 release (Fig S11a).

We agree with the reviewer that this is an important question, which we addressed in the following additional experiments. First, when we analysed subsets of granules in platelets, we found that alpha-granules exist, which contain more PF4 than P-Selectin (new figure 6 f, g). When we stimulated platelets with C5a, we observed a stronger release of CXCL4-containing granules in comparison to P-Selectin-containing granules (new figure 6h). As suggested, we also analysed megakaryocytes for differences in granule contents and CXCL4 mRNA. Using megakaryocytes from C5ar1^{-/-} mice, we observed no difference in CXCL4 area fraction (new figure 6 j, k). Furthermore, there was no difference in CXCL4 mRNA between WT and C5ar1^{-/-} megakaryocytes as analysed by real time PCR (new figure 6 l). These data strongly suggest C5aR1 activation does not alter megakaryocyte production of PF4. The data, which the reviewer considered to be of minor importance, i.e. the release of CXCL4 as opposed to serotonin or beta-hexosaminidase and CXCL4 release by C5a as opposed to other platelet stimulators in murine platelets were removed from the revised manuscript.

Moreover, using human platelets, we observed that C5a stimulation of platelets dose-dependently induced the release of CXCL-4 (new figure 6 i) but not of other angiogenesis-modulating factors such as thrombospondin-1, endostatin or PDGF (new supplemental figure 18), supporting the view of a specific C5aR1 effect, at least at the applied doses. Given the large number of identified angiogenesis-modulating factors stored in platelets, we now point to other paracrine mechanisms in the discussion section that have to be evaluated in future investigations.

3. C5aR does not effect platelet function (response to comment #3)

The authors should address the possibility that C5aR affects platelet homeostasis that in turn impacts neovascularization. PF4 release is a consequence of degranulation and degranulation leads to hemostasis. If both PF4 and hemostasis negatively affects neovascularization, it may be difficult to set them apart. The authors need to address this issue ideally experimentally. Note that

in a previous publication by the same group (Circulation 2018), the absence of C3aR prolonged bleeding times in mice. Are defects in hemostasis accounted for in the neovascularization phenotype of C5aR^{-/-} platelets? Does the absence of C5aR also affect bleeding times? Is coagulation/hemostasis altered by the absence of platelet C5aR to account for the phenotype? Is the ischemic injury protection in mice deficient in platelet C5aR mitigated by heparin? While all of these experiments need not be performed the issue needs to be addressed.

The authors then need to attempt to discuss/clarify the dual role of PF4 in the model. PF4 is a pro-coagulant (binds and neutralizes local heparins to permit primary hemostasis) and angiogenesis inhibitor. The role of PF4 is complex and needs to be contextualized:

<https://doi.org/10.1182/blood-2003-11-3994>

<https://pubmed.ncbi.nlm.nih.gov/12609838/>

<https://pubmed.ncbi.nlm.nih.gov/9395524>

<https://www.jbc.org/content/282/1/710.full>

As suggested by the reviewer, we introduced a paragraph into the discussion section of the revised manuscript addressing the complex role of PF4. Further, we included the references kindly provided by the reviewer in the revised manuscript and discussed these findings in the context of our results.

The separation of effects conferred by the influence on hemostasis from direct effects of C5a binding to platelet C5aR is an important point additionally raised by the reviewer. Our previous project on the impact of platelet C3aR activation on thrombus formation is not directly connected to the topic of this manuscript. C3a and C5a are two distinct anaphylatoxins that exert their effects through different GPCRs. Platelet mediated thrombosis is a function of C3a previously uncovered by our lab, which is distinct from effects of C5a. Further, C3a is not part of the cascade that drives the formation of the C5 convertase and the activation of the terminal pathway leading to C5 cleavage and generation of C5a.

To address the reviewers concern, we carried out additional in vivo experiments to determine the role of C5aR for coagulation/ hemostasis. After ferric-chloride-induced injury in C5aR1^{-/-} mice, we observed no significant difference in thrombus formation time in the absence of C5aR1 when compared to WT mice. Also, the angiogenic response of C5aR1^{-/-} mice vs. WT was not significantly different, when heparin was applied (see figure 1 for reviewer 2). Furthermore, C5aR1 deficiency had no significant impact on in vivo bleeding time or time to clot formation (supplemental figure 14c, d). Finally, when we applied a pharmacological inhibitor of C5aR1 (PMX205), we found that mice treated with this inhibitor showed significantly more ischemia-induced neovascularization in the hind limb ischemia model (new figure 7 q), whereas the bleeding time was not significantly altered by PMX205 (new supplemental figure 25).

4. Fig 12, platelet aggregation assays were done in the absence of C5a. ADP, TRAP, CRP responses were thus not affected in C5aR^{-/-} platelets. Does the addition of C5a to threshold

concentration of ADP increases platelet aggregation? The authors need to perform platelet aggregation studies with threshold ADP, CRP (not triggering aggregation) in the presence of C5a

We followed the reviewer's advice and carried out further aggregation studies. As a result, we identified the threshold for ADP- and CRP-induced aggregation in human platelets, which was 0.1 μM for ADP (with 0.01 μM or 0.1 μM no aggregation could be induced, but with 1 μM) and 0.1 mg/ml for CRP (with 0.01 mg/ml or 0.1 mg/ml no aggregation could be induced, but with 1 $\mu\text{g/ml}$, data not shown). When we added increasing concentrations of C5a (20ng/ml, 200 ng/ml, 2000ng/ml), we observed no significant increase in platelet aggregation (figure 2 for reviewer 2).

Reviewer #3 (Remarks to the Author):

In my initial review I was concerned mainly about two issues:

1. The biological effect in the HLI model was modest, and of unclear translational significance

As the mouse strain used in this project was C57/Bl6 well known to demonstrate a robust angiogenic response to hind-limb ischemia, our observed effect, which is an improvement over a pre-existing strong response to ischemia, may indeed be of biological relevance. To further address the issue of translational significance, we performed additional in vivo experiments, in which we specifically targeted C5aR1. Using the C5aR1-specific inhibitor PMX205, we observed in the hind limb ischemia model that revascularization was significantly increased in response to C5aR1-targeting (new Figure 7 q).

2. The characterization of C5aR1-induced platelet activation was superficial and incomplete with lack of convincing evidence supporting the most novel conclusion in the MS; i.e. that C5aR1 "preferentially" releases CXCL4.

The authors have added much new data (Figure 6) and several supplementary figures, but the data are still not at all consistent or convincing. The level of variance in the platelet release assays is unacceptably high, making conclusions difficult and in many cases the data are not consistent with the known biology of platelet; for example, seeing granule release in the absence of an aggregation response. Perhaps these issues could be resolved with a small number of straightforward studies using human rather than mouse platelets. Mouse platelets can be difficult to study ex vivo as methods of blood collection can greatly influence function.

Human platelets generally respond in a much more consistent manner than do mouse platelets, probably because of the difficulty in getting blood from mice in a state that does not at least partially activate them. Thus a series of simple studies examining the response of washed human platelets to graded doses of C5a looking at aggregation (or alpha2b/beta3 integrin activation) and granule releases assessed by P-selectin exposure, ADP release, and release of 2 or 3 angiogenic regulators (e.g. CXCL4, TSP1) would be very helpful. Ideally, showing that platelet activation was blocked by an inhibitory antibody to C5aR1 should also be done.

All experiments of the first round of revision were conceptualized as suggested by this reviewer to address his/ her concerns. Firstly, we ruled out baseline platelet inhibition using activation inhibitors, Secondly, we increased the number of experiments to increase the power and make the data more consistent. Thirdly, we analysed the dose response of CXCL4 to C5a and its saturation. Also, we analysed further factors to make our statements on C5a-specificity more reliable, performed additional time course experiments, tested further functional aspects of platelet activation by C5a stimulation such

as ATP release or in vitro aggregation. Finally, we analysed signal transduction pathways of C5aR1 in platelets.

In the 2nd revision, we carried out additional experiments as suggested by the reviewer. Using human platelets, we observed that C5a stimulation resulted in a dose-dependent release of CXCL-4 corroborating our murine data (new figure 6 i). Importantly, other angiogenesis-modulating factors such as thrombospondin-1, endostatin or PDGF were not significantly affected by C5a stimulation (new supplemental figure 18) strongly suggesting C5aR1-specificity. In the revised manuscript, we added data obtained with human platelets to partially replace data on murine platelets. Given the large number of identified angiogenesis-modulating factors stored in platelets (26, 27, 38), we point out in the discussion section that other paracrine mechanisms have to be evaluated in future investigations.

Furthermore, we looked at the platelet aggregation response upon C5a stimulation in human platelets. First, we identified the threshold for ADP- and CRP-induced aggregation in human platelets, which was 0.1 μ M for ADP (with 0.01 μ M or 0.1 μ M no aggregation could be induced, but with 1 μ M) and 0.1 mg/ml for CRP (with 0.01 mg/ml or 0.1 mg/ml no aggregation could be induced, but with 1 μ g/ml, figure 1 for reviewer 3). When we added increasing concentrations of C5a (20ng/ml, 200 ng/ml, 2000ng/ml,), we observed no significant increase in platelet aggregation (figure 1 for reviewer 3).

Then, we looked at alpha2b/beta3 integrin activation (as measured by PAC-1 binding to platelets and binding of fibrinogen) and observed activation of platelets by increasing doses of C5a (figure 2 a, b for reviewer 3). P-Selectin exposure, however, was not significantly changed (figure 2 c for reviewer 3). As requested, we also inhibited platelet C5aR1. Figure 3 for reviewer 3 shows, that blockade of C5aR1 resulted in reduced binding of fibrinogen to platelets.

a

b

Figure 1 for reviewer 2: Effect of heparin on hindlimb revascularization of WT and *C5ar1*^{-/-} mice
 (a) WT and *C5ar1*^{-/-} mice were treated with heparin at a dose of 500 U/kg over 14 days and hindlimb ischemia was induced. Mice were measured using laser doppler fluximetry over two weeks. Under influence of heparin, no significant difference in hindlimb revascularization between WT and *C5ar1*^{-/-} mice could be observed any more. Data are shown as the mean±SEM (n=6 animals per group) and are displayed as % of the perfusion in the contralateral control limb. *p<0.05. (b) Tail bleeding times of mice were assessed as described in the Methods section using heparin at the dose of the hindlimb ischemia experiment in (a). Heparin treatment induced a strong increase in time to complete bleeding cessation. Data are shown as the mean±SEM (n=7) and are displayed as % of control. The tail bleeding time in WT mice treated with vehicle control represents 100%. Two-way ANOVA with Bonferroni's post hoc test in (a). One-way ANOVA with Bonferroni's post hoc test in (b).

Figure 2 for reviewer 2: Effect of C5a on aggregation of human platelets

PRP was generated from human CPDA whole blood by centrifugation. The effect of C5a on aggregation was assessed. As there was no effect of C5a alone (data not shown), we tested threshold concentration of CRP (a) or ADP (b), which did not induce aggregation on their own. Additional treatment with C5a did not impact on the aggregation threshold in response to CRP or ADP as assessed by light transmission aggregometry. Shown is one representative experiment out of 3.

Figure 1 for reviewer 3: Effect of C5a on aggregation of human platelets

PRP was generated from human CPDA whole blood by centrifugation. The effect of C5a on aggregation was assessed. As there was no effect of C5a alone (data not shown), we tested threshold concentration of CRP (a) or ADP (b), which did not induce aggregation on their own. Additional treatment with C5a did not impact on the aggregation threshold in response to CRP or ADP as assessed by light transmission aggregometry. Shown is one representative experiment out of 3.

Figure 2 for reviewer 3: Effect of C5a on activation of human platelets

Platelets were isolated from citrated whole blood of healthy donors. Subsequently, platelets were stimulated using different concentrations of C5a. (a) C5a induced a significant increase of fibrinogen binding of platelets. (b) We also found that C5a stimulation induced significant upregulation of activated GPIIb/IIIa (PAC-1, an antibody binding selectively to activated GPIIb/IIIa) thus confirming this result. (c) However, no significant upregulation of CD62P was induced by C5a. Data are shown as the mean \pm SEM (n=4) and are displayed as % of control. The mean MFI or the percent binding of platelets in the vehicle-stimulated group represents 100%. 1-way ANOVA with Bonferroni post-hoc correction in (a) – (c).

Figure 3 for reviewer 3: Effect of C5aR1 blockade on C5a-induced activation of human platelets
 Platelets were isolated from citrated whole blood of healthy donors. Subsequently, platelets were preincubated with C5aR1 antagonist PMX205 or the control peptide PMXctrl and then stimulated using C5a. (a) After PMX205 preincubation, C5a did not induce a significant increase of fibrinogen binding. Data are shown as the mean \pm SEM (n=3-4) and are displayed as % of control. The mean MFI or the percent binding of platelets in the vehicle-stimulated group represents 100%. T-test.

REVIEWERS' COMMENTS

Reviewer #1 (Remarks to the Author):

I am satisfied with the response of the authors.

Reviewer #2 (Remarks to the Author):

the authors have nicely addressed all concerns and altered the manuscript appropriately

Reviewer #3 (Remarks to the Author):

The authors have done important additional experiments that adequately address the two major remaining concerns I had after the first revision. Most importantly, they performed a series of studies on human platelets that confirmed the mouse work and added supporting data to the most interesting observation of the MS; i.e. that C5a seems to cause preferential release of a granule compartment that is rich in PF4 but low in P-selectin and other angiogenic regulators (TSP). The concept of different granule subsets with differential release response to agonists has been raised by others (Italiano group), so this new work adds specific new information that advances the field. I remain somewhat troubled by the data that shows C5a inducing $\alpha 2b/\beta 3$ integrin activation (PAC1 binding and fibrinogen binding), but not aggregation. This should be commented on in the discussion.

Response to reviewers

Reviewer #1 (Remarks to the Author):

I am satisfied with the response of the authors.

We thank the reviewer for his/ her positive feedback.

Reviewer #2 (Remarks to the Author):

the authors have nicely addressed all concerns and altered the manuscript appropriately

We thank the reviewer for his/ her positive feedback.

Reviewer #3 (Remarks to the Author):

The authors have done important additional experiments that adequately address the two major remaining concerns I had after the first revision. Most importantly, they performed a series of studies on human platelets that confirmed the mouse work and added supporting data to the most interesting observation of the MS; i.e. that C5a seems to cause preferential release of a granule compartment that is rich in PF4 but low in P-selectin and other angiogenic regulators (TSP). The concept of different granule subsets with differential release response to agonists has been raised by others (Italiano group), so this new work adds specific new information that advances the field. I remain somewhat troubled by the data that shows C5a inducing $\alpha 2b/\beta 3$ integrin activation (PAC1 binding and fibrinogen binding), but not aggregation. This should be commented on in the discussion.

We fully agree with the reviewer and amended the discussion part accordingly. "At this point, we cannot entirely explain, why C5a induces $\alpha 2b/\beta 3$ integrin activation, but not aggregation. Further profound studies are needed to characterize this observation."